# Imitation Learning from Imperfection: Theoretical Justifications and Algorithms

**Ziniu Li**[*1,2]**, Tian Xu**[*3,4]**, Zeyu Qin**[†5]**, Yang Yu**[‡3,4]**, and Zhi-Quan Luo**[‡1,2]

[1]The Chinese University of Hong Kong, Shenzhen
[2]Shenzhen Research Institute of Big Data
[3]National Key Laboratory for Novel Software Technology, Nanjing University
[4]Polixir.ai
[5]Hong Kong University of Science and Technology

## Abstract

Imitation learning (IL) algorithms excel in acquiring high-quality policies from expert data for sequential decision-making tasks. But, their effectiveness is hampered when faced with limited expert data. To tackle this challenge, a novel framework called (offline) IL with supplementary data has been proposed [25, 61], which enhances learning by incorporating an additional yet imperfect dataset obtained inexpensively from sub-optimal policies. Nonetheless, learning becomes challenging due to the potential inclusion of out-of-expert-distribution samples. In this work, we propose a mathematical formalization of this framework, uncovering its limitations. Our theoretical analysis reveals that a naive approach—applying the behavioral cloning (BC) algorithm concept to the combined set of expert and supplementary data—may fall short of vanilla BC, which solely relies on expert data. This deficiency arises due to the distribution shift between the two data sources. To address this issue, we propose a new importance-sampling-based technique for selecting data within the expert distribution. We prove that the proposed method eliminates the gap of the naive approach, highlighting its efficacy when handling imperfect data. Empirical studies demonstrate that our method outperforms previous state-of-the-art methods in tasks including robotic locomotion control, Atari video games, and image classification.[1] Overall, our work underscores the potential of improving IL by leveraging diverse data sources through effective data selection.

## 1 Introduction

Imitation learning (IL) [3, 34] is an essential technique in artificial intelligence, allowing machines to enhance their performance by imitating expert behaviors. This technique has showcased remarkable achievements across various domains such as robotics [27], self-driving cars [35], and language models [55, 8]. Among the IL approaches, behavioral cloning (BC) [40] stands out as a popular method. In particular, BC leverages temporally-correlated state-action pairs extracted from expert demonstrations, employing them as training samples to learn a mapping from states to actions via maximum likelihood estimation. In this manner, IL expands upon the traditional supervised learning framework, enabling the acquisition of sequential decision-making capabilities.

The quantity of expert trajectories plays a crucial role in achieving satisfactory performance. Previous studies have shown that BC works well when the dataset contains a large number of expert-level

---

*: Equal contribution. Author ordering is determined by coin flip. Emails: `ziniuli@link.cuhk.edu.cn` and `xut@lamda.nju.edu.cn`.

†: Email: `zeyu.qin@connect.ust.hk`.

‡: Corresponding authors. Emails: `yuy@nju.edu.cn` and `luozq@cuhk.edu.cn`.

[1]The code is available at [https://github.com/liziniu/ISWBC](https://github.com/liziniu/ISWBC).

37th Conference on Neural Information Processing Systems (NeurIPS 2023).

trajectories [48]. However, the well-known compounding errors issue [44] renders any offline IL algorithm, including BC, ineffective when the number of expert trajectories is small [42, 60]. Furthermore, collecting more trajectories from the expert is costly and impractical in many domains, such as robotics and healthcare.

To overcome the challenge of scarce expert data, we propose to use an additional yet imperfect dataset to supplement the expert data; see Figure 1 for illustration. In particular, this additional dataset can be cheaply obtained by executing sub-optimal policies.[2] However, the incorporation of supplementary dataset introduces a distribution shift issue due to the presence of out-of-expert-distribution trajectories.[3] The distribution shift issue may hamper the model's performance in utilizing the supplementary data, as we will argue later.

We realize that a number of empirical successes have been reported in this direction [25, 24, 61, 32]. Most algorithms rely on a discriminator to distinguish between expert-style and sub-optimal samples, followed by optimization of a weighted BC objective to learn a good policy. For example, DemoDICE [25] uses a regularized state-action distribution matching objective to train the discriminator, while DWBC [61] employs a cooperative training framework for the policy and discriminator. Despite the empirical successes in certain scenarios, there remains a notable absence of systematic theoretical studies, particularly in terms of *imitation gap* (i.e., performance difference between the expert and learner), which may hinder deep understanding and impede future algorithmic advances.

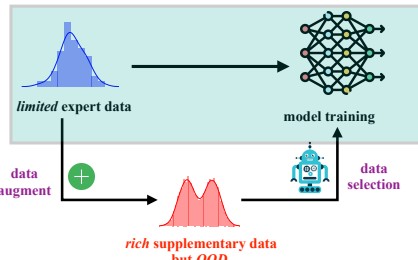

Figure 1: Compared with the conventional IL framework (shown in cyan), the *supplementary data* helps address the expert data scarcity issue, and the *data selection* technique helps address the distribution shift issue in model training.

Table 1: The theoretical guarantees of three methods: (1) BC, which relies solely on expert data; (2) NBCU, which directly uses the union of expert data and supplementary data without selection; and (3) ISW-BC, a new method that employs importance sampling for data selection. Compared with BC, NBCU suffers a non-vanishing error due to the distribution shift between two datasets while ISW-BC does not.

|  | Imitation Gap |
|---|---|
| BC | $\mathcal{O}(\frac{|\mathcal{S}|H^2}{N_\mathrm{E}})$ |
| NBCU | $\widetilde{\mathcal{O}}((1-\eta)(V(\pi^\mathrm{E}) - V(\pi^\beta)) + \frac{|\mathcal{S}|H^2}{N_\mathrm{E}+N_\mathrm{S}})$ |
| ISW-BC | $\mathcal{O}(\frac{|\mathcal{S}|H^2}{N_\mathrm{E}+N_\mathrm{S}/\mu})$ |

We aim to bridge the gap between theory and practice in the (offline) IL with supplementary data framework by developing effective algorithms and providing rigorous theoretical guarantees. To the best of our knowledge, only [9] provided imitation gap bounds for a model-based adversarial imitation learning approach in a similar problem. However, our focus lies on the widely used and simpler BC and its variants, which are model-free in nature. Our contributions are summarized below.

- We develop a formal mathematical framework of the IL with supplementary data framework and conduct corresponding theoretical analysis. Our findings highlight the impact of the distribution shift between expert and supplementary data. Our results are summarized in Table 1.[4]

  In particular, our analysis shows that naively applying BC on the union of expert and supplementary data (referred to as the NBCU method in this paper) has a non-vanishing error term in the imitation gap bound (see the second row of Table 1). This means that the direct use of additional data might yield inferior performance compared with solely using the expert data with BC. This necessitates the development of more advanced algorithms.

---

[2]For instance, web texts and images are easily available for language and vision models, respectively; robotics can benefit from previously collected datasets.

[3]This issue is separate from the *intrinsic* distribution shift problem that IL already faces, where the training and evaluation distributions differ [44]. It is important to note that these are distinct issues.

[4]We briefly explain notations in Table 1: $|\mathcal{S}|$ indicates the state space size, $H$ is the horizon length, $N_\mathrm{E}$ means the expert data size, and $N_\mathrm{S}$ refers to the supplementary data size. In addition, we define $\eta = N_\mathrm{E}/(N_\mathrm{E} + N_\mathrm{S})$, and $V(\pi^\mathrm{E}) - V(\pi^\beta)$ as the performance gap between expert policy $V^\mathrm{E}$ and supplementary data collection policy $\pi^\beta$. For ISW-BC's bound, $\mu > 0$ represents the state-action distribution density ratio bound between these two policies (to be detailed in Theorem 3).

- To address the distribution shift issue, we propose a new approach called ISW-BC, which uses the importance sampling technique to select data within the expert distribution. In contrast to prior methods [25, 61], ISW-BC re-weights data in an unbiased way. We develop a new imitation gap bound for ISW-BC (see the last row of Table 1), revealing that it not only eliminates the gap exhibited by the naive approach but also offers a superior guarantee over BC.

- Our theoretical analysis has been validated through extensive experiments, including robotic locomotion control, Atari video games, and image classification. The results affirm the superiority of ISW-BC over existing methods, thus demonstrating the potential of our method in addressing the distribution shift issue in IL with supplementary data.

## 2 Related Work

We review briefly relevant studies in the main text and provide a detailed discussion in Appendix B.

**Behavioral Cloning.** Behavioral cloning (BC) is a popular algorithm in the offline setting, where the learner cannot interact with the environment. According to the learning theory in [42], only using an expert dataset, BC has an imitation gap of $\mathcal{O}(|\mathcal{S}|H^2/N_{\mathrm{E}})$, where $|\mathcal{S}|$ is the state space size, $H$ is the planning horizon, and $N_{\mathrm{E}}$ is the number of expert trajectories. Our work investigates the use of a supplementary dataset to enhance the dependence on the data size.

**Adversarial Imitation Learning.** In contrast to BC, adversarial imitation learning (AIL) methods, such as GAIL [21], perform imitation through state-action distribution matching. It has been demonstrated both empirically and theoretically that AIL methods do not suffer from the compounding errors issue when the expert data is limited [21, 16, 26, 59, 60, 28, 63]. Under mild conditions, [62] provided a horizon-free bound of $\mathcal{O}(\min\{1, \sqrt{|\mathcal{S}|/N_{\mathrm{E}}}\})$, which is much better than BC in terms of dependence on $H$. However, AIL methods work naturally in the online setting (i.e., the interaction is allowed), which is not directly applicable in the offline setting that we study in this paper. Although the proposed method has a discriminator and a policy like AIL, our discriminator and policy are not designed to compete with each other adversarially, as we will explain in detail later.

**Imitation Learning with Supplementary Data.** Our theoretical study is motivated by recent empirical successes in IL with supplementary data [25, 61, 32, 24]. Compared with [25, 61], a related setting, learning from observation, is studied in [32, 24]. In this setting, expert actions are absent, and only expert states are observed. The importance sampling technique used in our method for addressing distribution shift is also studied in (semi-)supervised learning [49, 11, 30, 15]. Our contribution is to show this technique is also effective in the imitation learning, where data has a Markovian structure.

## 3 Preliminaries

**Markov Decision Process.** In this paper, we consider the episodic Markov decision process (MDP) framework [41]. An MDP is defined by the tuple $\mathcal{M} = (\mathcal{S}, \mathcal{A}, \mathcal{P}, r, H, \rho)$, where $\mathcal{S}$ and $\mathcal{A}$ are the state and action space, respectively. $H$ is the maximum length of a trajectory, and $\rho$ is the initial state distribution. The non-stationary transition function is specified by $\mathcal{P} = \{P_1, \cdots, P_H\}$, where $P_h(s_{h+1}|s_h, a_h)$ determines the probability of transiting to state $s_{h+1}$ given the current state $s_h$ and action $a_h$ in time step $h$, for $h \in [H]$. Here the symbol $[x]$ means the set of integers from 1 to $x$. Similarly, the reward function $r = \{r_1, \cdots, r_H\}$ specifies the reward received at each time step, where $r_h : \mathcal{S} \times \mathcal{A} \to [0, 1]$ for $h \in [H]$. A policy in an MDP is a function that maps each state to a probability distribution over actions. We consider time-dependent policies $\pi_h : \mathcal{S} \to \Delta(\mathcal{A})$, where $\Delta(\mathcal{A})$ is the probability simplex. The policy at each time step $h$ is denoted as $\pi_h$, and we use $\pi$ to denote the collection of time-dependent policies $\{\pi_h\}_{h=1}^{H}$ when the context is clear.

We measure the quality of a policy $\pi$ by the policy value (i.e., environment-specific long-term return): $V(\pi) = \mathbb{E}\left[\sum_{h=1}^{H} r_h(s_h, a_h) \mid s_1 \sim \rho; a_h \sim \pi_h(\cdot|s_h), s_{h+1} \sim P_h(\cdot|s_h, a_h), \forall h \in [H]\right]$. To facilitate later analysis, we need to introduce the state-action distribution $d_h^\pi(s, a) = \mathbb{P}(s_h = s, a_h = a|\pi)$. We use the convention that $d^\pi$ is the collection of time-dependent state-action distributions.

**Imitation Learning.** Imitation learning (IL) aims to learn a policy that mimics an expert policy based on expert demonstrations. In this paper, we assume that there exists a good expert policy $\pi^{\mathrm{E}}$ that generates a dataset $\mathcal{D}^{\mathrm{E}}$ consisting of $N_{\mathrm{E}}$ trajectories of length $H$.

$$\mathcal{D}^{\mathrm{E}} = \left\{ \mathrm{tr} = (s_1, a_1, s_2, a_2, \cdots, s_H, a_H) ; s_1 \sim \rho, a_h \sim \pi_h^{\mathrm{E}}(\cdot|s_h), s_{h+1} \sim P_h(\cdot|s_h, a_h), \forall h \in [H] \right\}.$$

The learner aims to imitate the expert using the expert dataset $\mathcal{D}^{\mathrm{E}}$. The quality of the imitation is measured by the *imitation gap*, defined as $\mathbb{E}\left[V(\pi^{\mathrm{E}}) - V(\pi)\right]$, where the expectation is taken over the randomness of data collection. It is worth noting that in the training phase, the IL learner does *not* have access to reward information. A good learner should closely mimic the expert, resulting in a small imitation gap. We assume that the expert policy is deterministic, a common assumption in the literature [42, 43, 60], and applicable to tasks such as MuJoCo locomotion control.

**Behavioral Cloning.** Behavioral cloning (BC) is a popular imitation learning algorithm that aims to learn a policy from an expert dataset $\mathcal{D}^{\mathrm{E}}$ via supervised learning. Specifically, BC seeks to find a policy $\pi^{\mathrm{BC}}$ that maximizes the log-likelihood of the expert actions in the dataset:

$$\pi^{\mathrm{BC}} \in \arg\max_{\pi} \sum_{h=1}^{H} \sum_{(s,a) \in \mathcal{S} \times \mathcal{A}} \widehat{d_h^{\mathrm{E}}}(s,a) \log \pi_h(a|s), \tag{1}$$

where $\widehat{d_h^{\mathrm{E}}}(s,a)$ is the empirical state-action distribution in the expert dataset. By the maximum likelihood estimation (MLE), BC can make good decisions by duplicating expert actions on states visited in $\mathcal{D}^{\mathrm{E}}$. However, BC may take sub-optimal actions on non-visited states, resulting in compounding errors and a large imitation gap. This issue is significant when the expert data is limited.

## 4  Imitation Learning with Supplementary Data

To address the challenge of limited availability of expert data, we consider an IL with a supplementary dataset framework. We assume that a supplementary yet imperfect dataset $\mathcal{D}^{\mathrm{S}} = \{ \mathrm{tr} = (s_1, a_1, s_2, a_2, \cdots, s_H, a_H) \}$ is collected by a behavior policy $\pi^{\beta}$. A naive approach is to perform MLE on the *union* of the expert and supplementary dataset $\mathcal{D}^{\mathrm{U}} = \mathcal{D}^{\mathrm{E}} \cup \mathcal{D}^{\mathrm{S}}$:

$$\pi^{\mathrm{NBCU}} \in \arg\max_{\pi} \sum_{h=1}^{H} \sum_{(s,a)} \widehat{d_h^{\mathrm{U}}}(s,a) \log \pi_h(a|s), \tag{2}$$

where $\widehat{d_h^{\mathrm{U}}}(s,a)$ is the empirical state-action distribution in $\mathcal{D}^{\mathrm{U}}$. We refer to this approach as NBCU (naive BC with the union dataset). NBCU treats these two datasets equally and is brittle to distribution shift, as we will demonstrate later. For theoretical analysis purpose, we assume expert data represents a $\eta \in [0, 1]$ fraction of the total union data.

**Assumption 1.** *The expert dataset $\mathcal{D}^{\mathrm{E}}$ and supplementary dataset $\mathcal{D}^{\mathrm{S}}$ are collected in the following way: each time, we roll-out a behavior policy $\pi^{\beta}$ with probability $1 - \eta$ and the expert policy with probability $\eta$. Such an experiment is independent and identically conducted $N_{\mathrm{tot}}$ times.*

Under Assumption 1, we slightly overload our notations: we use $N_{\mathrm{E}}$ to denote the *expected* number of expert trajectories, which is given by $N_{\mathrm{E}} = \eta N_{\mathrm{tot}}$, and $N_{\mathrm{S}}$ to denote the *expected* number of supplementary trajectories, which is given by $N_{\mathrm{S}} = (1 - \eta) N_{\mathrm{tot}}$. Note that the probabilistic sampling procedure does not change the nature of our theoretical insights. In practice, one may collect a fixed number of expert and supplementary trajectories, respectively.

To establish a common ground, we begin by specifying the policy representations. Here, we adopt tabular representations, which assume that the parameterized value functions can take any possible form. Specifically, we define $\pi_h(a|s; \theta) = \langle \phi(s,a), \theta \rangle$, where $\phi(s,a) \in \mathbb{R}^d$ is the feature representation and $\theta \in \mathbb{R}^d$ is the parameter to optimize. In tabular representations, we use one-hot features for $\phi(s,a)$. For a discussion on general function approximation schemes, please refer to Appendix E.

**Imitation Gap of BC.** In order to evaluate the usefulness of the supplementary dataset, we use BC with only the expert dataset as a baseline. The analysis of this approach has been done in the conventional IL set-up in [42], and we re-state their results in our setting.

**Theorem 1.** *Under Assumption 1, if we apply BC only on the expert dataset, we have that $\mathbb{E}\left[V(\pi^{\mathrm{E}}) - V(\pi^{\mathrm{BC}})\right] = \mathcal{O}(\frac{|\mathcal{S}|H^2}{N_{\mathrm{E}}})$, where the expectation is taken over the randomness in the dataset collection (same as other expectations).*

Proofs of Theorem 1 and other theoretical results are deferred to the Appendix. The proof of Theorem 1 builds on [42], with the main difference being that the number of expert trajectories is a random variable in our set-up. We handle this difficulty by using Lemma 3 in the Appendix. The quadratic dependence on the planning horizon $H$ indicates the compounding errors issue of BC.

If the expert data is limited (i.e., $N_E$ is small), the performance gap can be large, suggesting poor performance of BC in this scenario.

**Imitation Gap of NBCU.** Guarantees of naively using the supplementary data are presented below.

**Theorem 2.** *Under Assumption 1, if we apply BC on the union dataset, we have*
$$\mathbb{E}\left[V(\pi^E) - V(\pi^{NBCU})\right] = \mathcal{O}((1-\eta)(V(\pi^E) - V(\pi^\beta)) + \frac{|\mathcal{S}|H^2\log(N_{tot})}{N_{tot}}).$$

**Remark 1.** *In case when the behavior policy is inferior to the expert policy, we have $V(\pi^E) - V(\pi^\beta) > 0$. In this case, even if $N_{tot}$ is large enough to make the second term negligible, there is still a non-vanishing gap of $V(\pi^E) - V(\pi^\beta)$ due to the behavior policy's potential to collect non-expert actions. In other words, the recovered policy may select wrong actions even on expert states, leading to the sub-optimal performance of NBCU.*

*It is worthy to note that this non-vanishing error term can also be interpreted from the viewpoint of the distribution shift. Specifically, using the analysis in [62], we can show that*
$$V(\pi^E) - V(\pi^\beta) = \mathcal{O}(H\varepsilon_d) = \mathcal{O}(H^2\varepsilon_\pi),$$

*where $\varepsilon_d = \max_h \mathrm{TV}(d_h^{\pi^E}, d_h^{\pi^\beta})$ is the state-action distribution total variation (TV) distance and $\varepsilon_\pi = \max_h \max_s \mathrm{TV}(\pi_h^E(\cdot|s), \pi_h^\beta(\cdot|s))$ is the policy distribution TV distance. Hence, we can also view Theorem 2 in the context of underline{state-action} or underline{policy} distribution shifts.*

*Note also that we do not claim that NBCU is always worse than BC. Instead, Theorem 2 implies that if the distribution shift between two data datasets is small, NBCU could be better than BC by leveraging more data, which we will also show in experiments later.*

The next proposition establishes the inevitability of the gap $V(\pi^E) - V(\pi^\beta)$ in the worst case.

**Proposition 1.** *Under Assumption 1, there exists an MDP $\mathcal{M}$, an expert policy $\pi^E$ and a behavior policy $\pi^\beta$, such that we have $\mathbb{E}\left[V(\pi^E) - V(\pi^{NBCU})\right] = \Omega((1-\eta)(V(\pi^E) - V(\pi^\beta)))$.*

The hard instance in Proposition 1 builds on the following idea: NBCU considers all action labels in the union dataset equally important. Therefore, we can build an instance where the expert $\pi$ selects a good action with an one-step reward of 1, while the behavior policy $\pi^\beta$ chooses a bad action with an one-step reward of 0. The noise introduced by $\pi^\beta$ results in incorrect learning goals, causing NBCU to make a mistake with probability $1 - \eta$, which is the fraction of the noise in the union dataset. By a carefully designed transition probability, we can obtain the expected bound in Proposition 1.

## 5 Addressing Distribution Shift with Importance Sampling

In this section, we propose a data selection approach to alleviate the distribution shift issue between expert data and supplementary data. Our approach is inspired by recent works [25, 61], where a discriminator is trained to re-weight samples, and a weighted BC objective is used for policy optimization. Specifically, we define the weighted BC objective as follows:

$$\pi^{ISW-BC} \in \arg\max_\pi \sum_{h=1}^H \sum_{(s,a)\in\mathcal{S}\times\mathcal{A}} \left\{ \widehat{d_h^U}(s,a) \times [w_h(s,a)\log\pi_h(a|s)] \times \mathbb{I}[w_h(s,a) \geq \delta] \right\}, \quad (3)$$

where $\widehat{d_h^U}(s,a)$ is the empirical state-action distribution of the union dataset, and $w_h(s,a) \in [0,\infty)$ is the weight decided by the discriminator. We introduce a hyper-parameter $\delta \in [0,\infty)$ to better select samples.

We propose using the importance sampling technique [46, Chapter 9] to transfer samples in the union dataset to the expert policy distribution, which is the key idea behind our method. This technique helps address the failure mode of NBCU. In an ideal scenario where there are infinite samples (i.e., at the population level), $\widehat{d_h^U}$ would equal $d_h^U$. By setting $w_h(s,a) = d_h^E(s,a)/d_h^U(s,a)$, we obtain $\widehat{d_h^U}(s,a)w_h(s,a) = d_h^E(s,a)$, and the objective (3) enables the learning of a policy as if samples were collected by the expert policy. However, in practice, $d_h^E(s,a)$ and $d_h^U(s,a)$ are unknown, and we only have a finite number of samples from each of these distributions. Therefore, we must estimate the grounded importance sampling ratio $d_h^E(s,a)/d_h^U(s,a)$ from the expert data and union data.

We emphasize that estimating the probability densities of high-dimensional distributions separately for expert and union data and then calculating their quotient can be a challenging task. We take a different

approach. Inspired by [17], we directly train a discriminator to estimate the importance sampling ratio $d_h^{\mathrm{E}}(s,a)/d_h^{\mathrm{U}}(s,a)$. To this end, we introduce time-dependent parameterized discriminators $\{c_h : \mathcal{S} \times \mathcal{A} \to [0,1]\}_{h=1}^{H}$, each of which is optimized according to the objective function

$$\max_{c_h} \sum_{(s,a)\in\mathcal{S}\times\mathcal{A}} \widehat{d_h^{\mathrm{E}}}(s,a)\left[\log\left(c_h(s,a)\right)\right] + \sum_{(s,a)\in\mathcal{S}\times\mathcal{A}} \widehat{d_h^{\mathrm{U}}}(s,a)\left[\log\left(1 - c_h(s,a)\right)\right]. \tag{4}$$

Solving the optimization problem in (4) is equivalent to training a binary classifier that assigns positive labels to expert data and negative labels to union data. We can obtain the optimal discriminator at the population level, from which we can derive the importance sampling ratio formula:

$$c_h^{\star}(s,a) = \frac{d_h^{\mathrm{E}}(s,a)}{d_h^{\mathrm{E}}(s,a) + d_h^{\mathrm{U}}(s,a)}, \quad w_h(s,a) = \frac{c_h^{\star}(s,a)}{1 - c_h^{\star}(s,a)}. \tag{5}$$

Based on the previous discussion, we present the implementation of our proposed method, named ISW-BC (importance-sampling-weighted BC), in Algorithm 1. It is worth noting that ISW-BC employs an unbiased weighting rule since it directly estimates the importance sampling ratio. In contrast, previous approaches such as [25, 61] use regularized weighting rules that may fail to recover the expert policy even with infinite samples. For further details on the differences between our method and previous ones, please refer to Appendix B.

### 5.1 Negative Result of ISW-BC with Tabular Representations of Discriminator

We have not yet specified the representations of the discriminator. One natural choice is to use tabular representations, which correspond to linear function approximation with one-hot features. Tabular representations have a strong representation power since they can span all possible functions. However, surprisingly, we show that tabular representations can fail when considering generalization.

**Proposition 2.** *If the discriminator uses the one-hot feature with $\delta = 0$, we have $\pi^{\mathrm{ISW\text{-}BC}} = \pi^{\mathrm{BC}}$.*

Proposition 2 suggests that even if we have a large number of supplementary samples and use importance sampling, ISW-BC is not guaranteed to outperform BC based on tabular representations. To illustrate, suppose we have a sample $(s,a)$ that is an expert-style sample but only appears in the supplementary dataset, meaning that $d_h^{\mathrm{E}}(s,a) > 0, \widehat{d_h^{\mathrm{E}}}(s,a) = 0$ and $\widehat{d_h^{\mathrm{U}}}(s,a) > 0$. Using tabular representations, we can compute the closed-form solution $c_h^{\star}(s,a) = \widehat{d_h^{\mathrm{E}}}(s,a)/(\widehat{d_h^{\mathrm{E}}}(s,a) + \widehat{d_h^{\mathrm{U}}}(s,a)) = 0$. This implies that the importance sampling ratio $w_h(s,a) = c_h^{\star}(s,a)/(1 - c_h^{\star}(s,a)) = 0$, so this good sample does not contribute to the learning objective (3). The failure of tabular representations is due to their discrete treatment of data, ignoring internal correlations. Consequently, although they work well in minimizing the empirical loss, they are not good at *generalization*. This kind of failure mode is also mentioned in the GAN literature [4].

---

**Algorithm 1** ISW-BC

**Input:** Expert data $\mathcal{D}^{\mathrm{E}}$ and supplementary data $\mathcal{D}^{\mathrm{S}}$.
1: $\mathcal{D}^{\mathrm{U}} \leftarrow \mathcal{D}^{\mathrm{E}} \cup \mathcal{D}^{\mathrm{S}}$.
2: $c \leftarrow$ Train a binary classifier with positive labels for $\mathcal{D}^{\mathrm{E}}$ and negative labels for $\mathcal{D}^{\mathrm{U}}$.
3: $w \leftarrow$ Compute importance sampling ratio by Eq. (5).
4: $\pi^{\mathrm{ISW\text{-}BC}} \leftarrow$ Apply weighted BC to learn a policy by Eq. (3) with $\mathcal{D}^{\mathrm{U}}$ and $w$.
**Output:** Policy $\pi^{\mathrm{ISW\text{-}BC}}$.

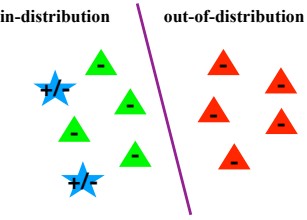

Figure 2: Illustration for ISW-BC. Please refer to the text below Assumption 2 for a detailed explanation.

---

### 5.2 Positive Result of ISW-BC with Function Approximation of Discriminator

In this section, we address the issue raised in the previous section by investigating ISW-BC with a specific function approximation. To avoid the limitations of tabular representations, we consider that the discriminator is parameterized by $c_h(s,a;\theta_h) = \frac{1}{1+\exp(-\langle\phi_h(s,a),\theta_h\rangle)}$, where $\phi_h : \mathcal{S} \times \mathcal{A} \to \mathbb{R}^d$ is a fixed feature mapping and $\theta_h \in \mathbb{R}^d$ is the parameter to be trained. Note that we require $d < |\mathcal{S}||\mathcal{A}|$ to avoid the tabular representations. Let $g(x) = \log(1 + \exp(x))$. Then, the optimization problem of

the discriminator becomes:

$$\min_{\theta_h} \mathcal{L}_h(\theta_h) \triangleq \sum_{(s,a)} \widehat{d_h^{\mathrm{E}}}(s,a)g(-\langle\phi_h(s,a),\theta_h\rangle) + \sum_{(s,a)} \widehat{d_h^{\mathrm{U}}}(s,a)g(\langle\phi_h(s,a),\theta_h\rangle) \qquad (6)$$

Let $\theta^\star = \{\theta_1^\star, \cdots, \theta_H^\star\}$ be the optimal solution obtained from Eq. (6). With the feature vector, samples are no longer treated independently, and the discriminator can perform *structured* estimation. To be consistent with the previous results, the policy is still based on tabular representations.

In the context of general linear function approximation, it is no longer possible to obtain a closed-form solution for $c^\star$ as in Eq. (5). This raises the question: what can we infer about $c^\star$? Our intuition can be described as follows. We can envision the supplementary dataset containing two types of samples: some that were in-expert distribution, and others that were out-of-expert distribution. We expect that $w_h(s,a)$ is large in the former case and small in the latter case. Note that $w_h(s,a)$ is monotonic with respect to the inner product $\langle\phi_h(s,a),\theta\rangle$. Therefore, we conclude that a larger value of $\langle\phi_h(s,a),\theta\rangle$ implies a more significant contribution to the learning objective (3). In the following part, we demonstrate that the aforementioned intuition can be achieved under mild assumptions.

**Assumption 2.** *Let $\mathcal{D}_h^{\mathrm{S}}$ denote the set of state-action pairs in $\mathcal{D}^{\mathrm{S}}$ in $h$. Define $\mathcal{D}_h^{\mathrm{S},1} = \{(s,a) \in \mathcal{D}_h^{\mathrm{S}} : d_h^{\pi^{\mathrm{E}}}(s) > 0, a = \pi_h^{\mathrm{E}}(s)\}$ as the in-expert-distribution dataset in $\mathcal{D}_h^{\mathrm{S}}$ and $\mathcal{D}_h^{\mathrm{S},2} = \mathcal{D}_h^{\mathrm{S}} \setminus \mathcal{D}_h^{\mathrm{S},1}$ as the out-of-expert-distribution dataset. There exists a ground truth parameter $\bar{\theta}_h \in \mathbb{R}^d$, for any $(s,a) \in \mathcal{D}_h^{\mathrm{E}} \cup \mathcal{D}_h^{\mathrm{S},1}$ and $(s',a') \in \mathcal{D}_h^{\mathrm{S},2}$, it holds that $\langle\bar{\theta}_h,\phi_h(s,a)\rangle > 0$ and $\langle\bar{\theta}_h,\phi_h(s',a')\rangle < 0$.*

Readers may realize that Assumption 2 is closely related to the notion of "margin" in the classification problem. Define $\Delta_h(\theta) \triangleq \min_{(s,a)\in\mathcal{D}_h^{\mathrm{E}}\cup\mathcal{D}_h^{\mathrm{S},1}}\langle\theta,\phi_h(s,a)\rangle - \max_{(s',a')\in\mathcal{D}_h^{\mathrm{S},2}}\langle\theta,\phi_h(s',a')\rangle$. From Assumption 2, we have $\Delta_h(\bar{\theta}_h) > 0$. This means that there *exists* a classifier that recognizes samples from both $\mathcal{D}_h^{\mathrm{E}}$ and $\mathcal{D}_h^{\mathrm{S},1}$ as in-expert-distribution samples and samples from $\mathcal{D}_h^{\mathrm{S},2}$ as out-of-expert-distribution samples. Note that such a nice classifier is assumed to exist, which is not identical to what is learned via Eq. (6). Before further discussion, we note that $\bar{\theta}_h$ is not unique if it exists. Without loss of generality, we define $\bar{\theta}_h$ as that can achieve the maximum margin (among all unit vectors).

Let us delve into the technical challenge that arises from Assumption 2. Although we assume two modes in the supplementary dataset, the learner is not aware of them beforehand. To gain a better understanding, refer to Figure 2, where the "star" corresponds to the expert data and the "triangle" corresponds to the supplementary data. The green and red parts of the triangle represent $\mathcal{D}^{\mathrm{S},1}$ and $\mathcal{D}^{\mathrm{S},2}$, respectively. While training the discriminator, we assign positive labels (shown in "+") to the expert data and negative labels (shown in "-") to the union data. Consequently, it becomes challenging to determine the learned decision boundary theoretically. To address this challenge, we develop the landscape properties, Lipschitz continuity and quadratic growth conditions, in Lemma 1 and Lemma 2, respectively. These terminologies are from the optimization literature; see [23, 13]. Incorporating these properties will enable us to infer the learned decision boundary.

**Lemma 1.** *For any $\theta \in \mathbb{R}^d$, the margin function is $L_h$-Lipschitz continuous in the sense that $\Delta_h(\bar{\theta}_h) - \Delta_h(\theta) \leq L_h \left\|\bar{\theta}_h - \theta\right\|$, where $L_h = \left\|\phi_h(s^1,a^1) - \phi_h(s^2,a^2)\right\|$ with $(s^1,a^1) \in \mathrm{argmin}_{(s,a)\in\mathcal{D}_h^{\mathrm{E}}\cup\mathcal{D}_h^{\mathrm{S},1}}\langle\theta,\phi_h(s,a)\rangle$ and $(s^2,a^2) \in \mathrm{argmax}_{(s,a)\in\mathcal{D}_h^{\mathrm{S},2}}\langle\theta,\phi_h(s,a)\rangle$.*

**Lemma 2.** *For any $h$, let $A_h \in \mathbb{R}^{N_{\mathrm{tot}}\times d}$ be the matrix that aggregates the feature vectors of samples in $\mathcal{D}_h^{\mathrm{U}}$. Assume that $\mathrm{rank}(A_h) = d$, then $\mathcal{L}_h$ (defined in Eq. (6)) has a (one-sided) quadratic growth condition. That is, there exists $\tau_h > 0$ such that $\mathcal{L}_h(\bar{\theta}_h) \geq \mathcal{L}_h(\theta_h^\star) + \frac{\tau_h}{2}\left\|\bar{\theta}_h - \theta_h^\star\right\|^2$.*

Using Lemma 1 and Lemma 2, we are ready to obtain the imitation gap bound of ISW-BC.

**Theorem 3.** *Under Assumptions 1 and 2, let $\mu = \max_{(s,h)\in\mathcal{S}\times[H]} d_h^{\pi^{\mathrm{E}}}(s,\pi_h^{\mathrm{E}}(s))/d_h^{\pi^\beta}(s,\pi_h^{\mathrm{E}}(s)) < \infty$, if the feature is designed such that $\sqrt{\frac{2(\mathcal{L}_h(\bar{\theta}_h)-\mathcal{L}_h(\theta_h^\star))}{\tau_h}} < \frac{\Delta_h(\bar{\theta}_h)}{L_h}$ holds, then we have $\Delta_h(\theta_h^\star) > 0$. Furthermore, we have the imitation gap bound $\mathbb{E}[V(\pi^{\mathrm{E}}) - V(\pi^{\mathrm{ISW-BC}})] = \mathcal{O}(\frac{|\mathcal{S}|H^2}{N_{\mathrm{E}}+N_{\mathrm{S}}/\mu})$.*

In order to interpret Theorem 3, it is important to note that $\Delta_h(\theta_h^\star) > 0$ means that there exists a $\delta > 0$ such that $w_h(s,a;\theta_h^\star) > \delta$ for $(s,a) \in \mathcal{D}_h^{\mathrm{E}} \cup \mathcal{D}_h^{\mathrm{S},1}$ and $w_h(s,a;\theta_h^\star) < \delta$ for $(s,a) \in \mathcal{D}_h^{\mathrm{S},2}$. As a result, all samples from $\mathcal{D}_h^{\mathrm{E}}$ and $\mathcal{D}_h^{\mathrm{S},1}$ are assigned with large weights, which allows ISW-BC to make use of additional samples and outperform BC.

We remark that the imitation gap bound of ISW-BC is dependent on the number of expert-style state-action pairs presented in the union of $\mathcal{D}_h^{\mathrm{E}}$ and $\mathcal{D}_h^{\mathrm{S},1}$. This number is represented as $N_{\mathrm{E}} + N_{\mathrm{S}}/\mu$, where $\mu$ is a state-action coverage parameter. It is important to mention that a similar notation is used in the literature of offline RL, as seen in [33, 10]. Additionally, ISW-BC has the ability to eliminate the gap of NBCU, meaning there is no non-vanishing error in Theorem 3. Moreover, ISW-BC can perform well even when $\mathcal{D}_h^{\mathrm{S},2}$ has noisy action labels, a scenario where NBCU may fail.

Although Theorem 3 produces desirable outcomes, it does have some limitations. First, the theoretical analysis necessitates knowledge of $\delta$, which is typically challenging to determine beforehand. However, our empirical findings in Section 6 demonstrate that setting $\delta = 0$ is effective in practice. Second, Theorem 3 mandates the use of good smooth features to ensure the required inequality holds, thereby avoiding the undesirable case presented in Proposition 2. Our paper does not offer a solution for finding such feature representations. Nevertheless, our experiments indicate that neural networks can usually learn suitable features. We present a simple mathematical example corresponding to Theorem 3 in Appendix D.5. We leave more general results of ISW-BC to future work.

## 6 Experiments

To validate the theoretical claims, we perform numerical experiments. We provide a brief overview of the experiment set-up below, and the details can be found in Appendix H.

### 6.1 Robotic Locomotion Control

In this section, we present our experiment on locomotion control, where we train a robot to run like a human in four environments from the Gym MuJoCo suite [14]: `Ant`, `Hopper`, `Halfcheetah`, and `Walker`. We adopt online SAC [19] to train an agent for each environment with 1M steps, and consider the resultant policy as the expert. For each environment, the expert data contains 1 trajectory collected by the expert policy. We consider two types of supplementary datasets:

- `Full Replay` (small distribution shift): the supplementary dataset (1 million samples) is directly sampled from the experience replay buffer of the online SAC agent, which is suggested by [25]. This setting has a small distribution shift as the online agent quickly converges to the expert policy (see Figure 5 in the Appendix), resulting in abundant expert trajectories in the replay buffer.

- `Noisy Expert` (large distribution shift): the supplementary dataset consists of 10 clean expert trajectories and 5 noisy expert trajectories where the action labels are corrupted (i.e., replaced by random actions). This introduces a large state-action distribution shift, as discussed in Remark 1. For further discussion on dataset corruption and distribution shift, please refer to Appendix E.2.

Table 2: Environment return of algorithms on 4 robotic locomotion control tasks. Digits correspond to the mean performance over 5 random seeds and the subscript $\pm$ indicates the standard deviation. "Avg" computes the normalized score over environments. Same as the other tables.

|  |  | Ant | HalfCheetah | Hopper | Walker | Avg |
|---|---|---|---|---|---|---|
|  | Random | $-326$ | $-280$ | $-20$ | $2$ | $0\%$ |
|  | Expert | $5229$ | $11115$ | $3589$ | $5082$ | $100\%$ |
|  | BC | $1759_{\pm287}$ | $931_{\pm273}$ | $2468_{\pm164}$ | $1738_{\pm311}$ | $38\%$ |
| Full Replay | NBCU | $4932_{\pm148}$ | $10566_{\pm86}$ | $3241_{\pm276}$ | $4462_{\pm105}$ | $92\%$ |
|  | DemoDICE | $\mathbf{5000}_{\pm124}$ | $10781_{\pm67}$ | $3394_{\pm93}$ | $\mathbf{4537}_{\pm125}$ | $\mathbf{94\%}$ |
|  | DWBC | $2951_{\pm155}$ | $1485_{\pm377}$ | $2567_{\pm88}$ | $1572_{\pm225}$ | $44\%$ |
|  | ISW-BC | $4933_{\pm110}$ | $\mathbf{10786}_{\pm56}$ | $\mathbf{3434}_{\pm38}$ | $4475_{\pm164}$ | $\mathbf{94\%}$ |
| Noisy Expert | NBCU | $3259_{\pm159}$ | $5561_{\pm539}$ | $558_{\pm23}$ | $518_{\pm56}$ | $35\%$ |
|  | DemoDICE | $2523_{\pm244}$ | $6020_{\pm346}$ | $1990_{\pm90}$ | $1685_{\pm160}$ | $49\%$ |
|  | DWBC | $\mathbf{3270}_{\pm238}$ | $5688_{\pm557}$ | $\mathbf{3317}_{\pm59}$ | $1985_{\pm175}$ | $62\%$ |
|  | ISW-BC | $3075_{\pm268}$ | $\mathbf{9284}_{\pm346}$ | $2624_{\pm249}$ | $\mathbf{2859}_{\pm407}$ | $\mathbf{69\%}$ |

Besides our proposed methods, we also evaluate two state-of-the-art methods in the locomotion control domain: DemoDICE [25] and DWBC [61]. We report the experiment results in Table 2.

We observe that BC suffers since the amount of expert data is limited. In the full replay task, NBCU performs well due to the small distribution shift. However, in the noisy expert task, NBCU's average

performance is inferior to that of BC[5], while ISW-BC outperforms NBCU in average significantly, demonstrating the robustness of ISW-BC to distribution shift. It is worth noting that among all evaluated methods, only our proposed method ISW-BC performs well in both settings. Prior methods such as DemoDICE and DWBC only perform well in one of the two settings.

## 6.2 Atari Video Games

In this section, we evaluate algorithms on Atari games [5], which involve video frames as inputs and discrete controls as outputs. Furthermore, environment transitions are stochastic for these games. We consider 5 games, namely `Alien`, `MsPacman`, `Phoenix`, `Qbert`, and `SpaceInvaders`. We obtain the offline expert data and supplementary data from the replay buffer of an online DQN agent, as released by [2]. We use the expert data from the buffer with the last index, which only contains 50k frames, to create a challenging learning setting. To augment this data, we use earlier replay buffer data to obtain supplementary data with approximately 200k frames. We consider the same baselines as in Section 6.1. All methods build on the classical convolutional neural networks used in DQN.

Similar to Section 6.1, we consider two types of supplementary data. The `full replay` setting involves supplementary data that is close to the expert data, exhibiting a small distribution shift. The `noisy expert` setting has noisy action labels, leading to a large distribution shift. Experiment details can be found in Appendix H.1.2. We report the game scores of the trained policies in Table 3.

Table 3: Environment return of algorithms on 5 Atari video games.

|  |  | Alien | MsPacman | Phoenix | Qbert | SpaceInvaders | Avg |
|---|---|---|---|---|---|---|---|
|  | Random | $-228$ | 307 | 761 | 164 | 148 | 0% |
|  | Expert | 2443 | 3601 | 4869 | 10955 | 1783 | 100% |
|  | BC | $1051_{\pm21}$ | $1799_{\pm27}$ | $1520_{\pm56}$ | $4769_{\pm111}$ | $472_{\pm10}$ | 32% |
| Full Replay | NBCU | $1405_{\pm28}$ | $2089_{\pm48}$ | $\mathbf{2431}_{\pm104}$ | $\mathbf{8065}_{\pm109}$ | $600_{\pm13}$ | **50%** |
|  | DemoDICE | $1401_{\pm16}$ | $2146_{\pm52}$ | $2192_{\pm72}$ | $7820_{\pm206}$ | $558_{\pm29}$ | 48% |
|  | DWBC | $122_{\pm4}$ | $1251_{\pm56}$ | $583_{\pm33}$ | $1078_{\pm50}$ | $287_{\pm6}$ | 7% |
|  | ISW-BC | $\mathbf{1452}_{\pm37}$ | $\mathbf{2162}_{\pm36}$ | $2299_{\pm76}$ | $7848_{\pm237}$ | $\mathbf{613}_{\pm16}$ | **50%** |
| Noisy Expert | NBCU | $944_{\pm22}$ | $1378_{\pm30}$ | $1491_{\pm55}$ | $4366_{\pm458}$ | $418_{\pm14}$ | 27% |
|  | DemoDICE | $1054_{\pm38}$ | $1604_{\pm59}$ | $1448_{\pm112}$ | $\mathbf{5354}_{\pm295}$ | $395_{\pm10}$ | 31% |
|  | DWBC | $643_{\pm18}$ | $656_{\pm16}$ | $1165_{\pm87}$ | $3860_{\pm104}$ | $296_{\pm5}$ | 16% |
|  | ISW-BC | $\mathbf{1122}_{\pm28}$ | $\mathbf{1980}_{\pm51}$ | $\mathbf{1618}_{\pm51}$ | $5247_{\pm328}$ | $\mathbf{497}_{\pm6}$ | **36%** |

Our observations are consistent with those of the previous experiments. NBCU performs well when the distribution shift is small, while only ISW-BC is robust when the distribution shift is large.

## 6.3 Image Classification

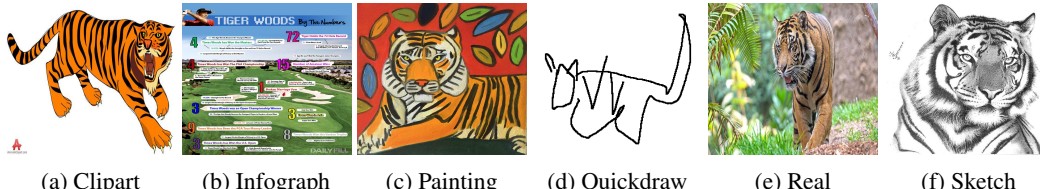

| (a) Clipart | (b) Infograph | (c) Painting | (d) Quickdraw | (e) Real | (f) Sketch |

Figure 3: Samples of `tiger` class from 6 sub-datasets of the DomainNet [36] dataset. Infograph and quickdraw have quite different patterns (i.e., distribution shift) compared with the others.

In our final experiment, we tackle an image classification task. This task is a special type of imitation learning where the planning horizon is 1 and there are no environment transitions. The reward is classification accuracy. Please note that our main purpose here is to use this degraded one-step task to verify the theoretical results.

We use a famous dataset, DomainNet [36], which comprises 6 sub-datasets (`clipart`, `infograph`, `painting`, `quickdraw`, `real`, and `sketch`) that have different feature patterns and hence distribution shifts; see Figure 3 for an illustration. Following [22], our task is to perform 10-class classification (`bird`, `feather`, `headphones`, `ice_cream`, `teapot`, `tiger`, `whale`, `windmill`, `wine_glass`, and

---

[5]We provide an explanation of why NBCU is better than BC on Ant and Walker in this scenario in Appendix.

`zebra`) using 80% of the images for training and 20% for test. Each sub-dataset has roughly 2000-5000 images.

We build the classifier on the pretrained ResNet-18 [20], as directly training ResNet-18 on the DomainNet dataset failed. We then optimize a 2-hidden-layer neural network, where inputs are from the feature representations extracted by the pretrained and fixed ResNet-18. We create 6 sub-tasks, where one of the 6 sub-datasets is used as the expert data while the other 5 sub-datasets are used as the supplementary datasets. We evaluate the classification accuracy on the expert test data. Note that there is no natural extension of DemoDICE for this task. More details can be found in Appendix H.1.3.

The results of our experiment are presented in Table 4. We observe that due to the presence of distribution shifts, NBCU performs even worse than BC, even though NBCU use more data than BC. On the other hand, ISW-BC can improve the performance over BC on 5 out of 6 tasks by re-weighting the supplementary data. At the same time, ISW-BC is more effective than DWBC.

Table 4: Test classification accuracy (%) of algorithms on 6 types of expert and supplementary data.

|  | Clipart | Infograph | Painting | Quickdraw | Real | Sketch | Avg |
|---|---|---|---|---|---|---|---|
| BC | $89.31_{\pm 0.01}$ | $55.80_{\pm 0.01}$ | $90.14_{\pm 0.00}$ | $\mathbf{85.61}_{\pm 0.01}$ | $96.19_{\pm 0.00}$ | $87.58_{\pm 0.01}$ | 84.10 |
| NBCU | $89.16_{\pm 0.02}$ | $56.32_{\pm 0.02}$ | $88.29_{\pm 0.01}$ | $84.78_{\pm 0.03}$ | $95.31_{\pm 0.00}$ | $87.57_{\pm 0.00}$ | 83.57 |
| DWBC | $90.00_{\pm 0.09}$ | $57.44_{\pm 0.06}$ | $90.89_{\pm 0.04}$ | $85.09_{\pm 0.09}$ | $96.35_{\pm 0.01}$ | $88.86_{\pm 0.09}$ | 84.77 |
| ISW-BC | $\mathbf{90.86}_{\pm 0.00}$ | $\mathbf{57.52}_{\pm 0.01}$ | $\mathbf{91.78}_{\pm 0.01}$ | $84.97_{\pm 0.01}$ | $\mathbf{96.56}_{\pm 0.01}$ | $\mathbf{89.63}_{\pm 0.06}$ | $\mathbf{85.22}$ |

## 7    Discussion and Conclusion

This paper introduces a formal mathematical framework for imitation learning with a supplementary yet imperfect dataset, which is designed to address the scarcity of expert data. Within this framework, we present new theoretical insights that illuminate the distribution shift challenge between expert and supplementary data. To deal with this challenge, we devise a new method named ISW-BC, employing the importance sampling technique to select data within the expert distribution. Through both theoretical analysis and empirical evaluations, we show the superiority of the proposed approach.

Our research is closely connected with data-centric artificial intelligence (AI) [39, 54, 64]. Here, the emphasis lies in the quality, availability, and management of data as foundational elements for constructing effective AI models and applications. The importance sampling technique developed in this paper proves valuable for processing imperfect data.

Furthermore, our methods can extend beyond the scope of consideration in this paper. To illustrate, the core concept of data re-weighting and selection can find utility in the realm of large language models. In specific downstream tasks with limited expert data, we can judiciously select a set of pre-training tasks to construct supplementary data and to enhance overall performance of language models; for additional insights, please refer to recent progress in [58, 18]. Other potential avenues for future exploration include the extension to multi-task imitation learning, as well as the unsupervised case where the expert data is not available.[6]

Overall, our findings demonstrate the potential of improving imitation learning performance by leveraging diverse data sources through effective data selection. We aspire for this work to serve as inspiration for future advancements in the field.

## Acknowledgment

We thank Congliang Chen for discussing a technical lemma. The work of Yang Yu is supported by National Key Research and Development Program of China (2020AAA0107200), NSFC (61876077), and Collaborative Innovation Center of Novel Software Technology and Industrialization. The work of Zhi-Quan Luo is supported in part by the National Key Research and Development Project under grant 2022YFA1003900, and in part by the Guangdong Provincial Key Laboratory of Big Data Computing.

---

[6]In this scenario, we can consider using a small fraction of imperfect data to reconstruct a surrogate expert policy and expert data (e.g., via majority voting). Subsequently, we can regard the remaining imperfect data as supplementary and apply the proposed framework.

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

## A Broader Impacts

This study delves into the theoretical aspects of offline imitation learning with supplementary data, and we verify our findings through experiments on established benchmarks. While this paper does not present any immediate, direct social impacts, the potential practical applications of our research could bring about positive change. By expanding the reach of imitation learning algorithms, our work may facilitate the development of more efficient and effective solutions in fields such as robotics, autonomous vehicles, and healthcare. However, we must also acknowledge that the misuse of such technology could have negative consequences, such as the manipulation of information to influence people's behavior. Therefore, it is crucial to remain vigilant in ensuring that the benefits of imitation learning are harnessed in a responsible and ethical manner.

## B Additional Related Work

### B.1 Imitation Learning from Imperfection

The problem considered in this paper is related to IL with a single imperfect dataset [56, 6, 51, 52, 45, 31]. In particular, the supplementary dataset in our set-up can also be viewed as imperfect demonstrations. However, our problem setting differs from IL with imperfect demonstrations in two key aspects. First, in IL with imperfect demonstrations, they either pose strong assumptions [51, 45, 31] or require auxiliary information (e.g., confidence scores on imperfect trajectories) on the imperfect dataset [56, 6]. In contrast, we assume access to a small number of expert trajectories to identify in-expert-distribution data. Second, most works [56, 6, 51, 52] in IL with imperfect demonstrations require online environment interactions while we focus on the offline setting. Additionally, in a related study, [7] employed imperfect data to infer a reward function. Under the assumption of the expert being strictly sub-optimal, [7] demonstrated that it is possible to learn a policy that surpasses the performance of the demonstrator by utilizing the recovered reward function.

Our research also bears relevance to the recent works by [29, 47], which aimed to unify offline Reinforcement Learning (RL) and IL. Different from our approach, these studies adopt a principle akin to adversarial imitation learning [50]. For a more detailed discussion, we refer readers to [29, 47]. We also observe a connection between the objective of importance sampling weighted behavior cloning (ISW-BC) and the reward-weighted regression (RWR) framework introduced in [38, 37]. Specifically, the training objective of ISW-BC can be likened to reward-weighted regression (RWR) with $\gamma = 0$ and a reward function denoted as $r(s,a) = \log \frac{d^{\mathrm{E}}(s,a)}{d^{\mathrm{U}}(s,a)}$. Nevertheless, there exist two distinctions between the two approaches. First, RWR primarily focuses on the online setting, while our work centers around the offline setting. Second, RWR finds its application in the context of RL, where rewards are readily available. Conversely, in our imitation learning setup, the reward (or the importance sampling ratio) needs to be inferred.

### B.2 Difference with DemoDICE and DWBC

Our work builds upon previous research in IL with supplementary data, specifically the algorithms DemoDICE [25] and DWBC [61]. A significant distinction arises between ISW-BC and these two methods in terms of the weighting rule design. While DemoDICE and DWBC employ *regularized* weighting rules, our method directly estimates the importance sampling ratio. This fundamental difference can be critical as regularized weighting rules may struggle to recover the expert policy exactly even with infinite samples. We provide further elaboration on this point below.

First, DemoDICE also uses the weighted BC objective in Eq. (3). But, DemoDICE uses the weighting rule of $\widetilde{w}(s,a) \propto d^\star(s,a)/d^{\mathrm{U}}(s,a)$ (refer to the formula between Equations (19)-(20) in [25]), where $d^\star(s,a)$ is computed by the expert's state-action distribution matching objective regularized by a divergence to the union data distribution (refer to [25, Equations (5)-(7)]):[7]

$$d^\star = \underset{d}{\arg\min}\, D_{\mathrm{KL}}(d\|d^{\mathrm{E}}) + \alpha D_{\mathrm{KL}}(d\|d^{\mathrm{U}})$$
$$\text{s.t.}\quad d(s,a) \geq 0 \quad \forall s,a.$$

---

[7]For a moment, we use the notations in [25] and present their results under the stationary and infinite-horizon MDPs. Same as the discussion of DWBC [61].

$$\sum_a d(s,a) = (1-\gamma)\rho(s) + \gamma \sum_{s',a'} P(s|s',a')d(s',a') \quad \forall s.$$

where $\gamma \in [0,1)$ is the discount factor, $\alpha > 0$ is a hyper-parameter. Due to the regularization term in the objective, it holds that $d^\star(s,a) \neq d^{\pi^E}(s,a)$, resulting in a biased weighting rule $\widetilde{w}(s,a)$.

Second, DWBC considers a different policy learning objective (refer to [61, Equation (17)]):

$$
\begin{aligned}
\min_\pi \quad & \alpha \sum_{(s,a)\in\mathcal{D}^E} [-\log\pi(a|s)] - \sum_{(s,a)\in\mathcal{D}^E}\left[-\log\pi(a|s)\cdot\frac{\lambda}{c(1-c)}\right] \\
& + \sum_{(s,a)\in\mathcal{D}^S}\left[-\log\pi(a|s)\cdot\frac{1}{1-c}\right],
\end{aligned}
\tag{7}
$$

where $\alpha > 0, \lambda > 0$ are hyper-parameters, and $c$ is the output of the discriminator that is jointly trained with $\pi$ (refer to [61, Equation (8)]):

$$
\begin{aligned}
\min_c \quad & \lambda \sum_{(s,a)\in\mathcal{D}^E} [-\log c(s,a,\log\pi(a|s))] + \sum_{(s,a)\in\mathcal{D}^S} [-\log(1-c(s,a,\log\pi(a|s)))] \\
& - \lambda \sum_{(s,a)\in\mathcal{D}^E} [-\log(1-c(s,a,\log\pi(a|s)))].
\end{aligned}
$$

Since its input additionally incorporates $\log\pi$, the discriminator is not guaranteed to estimate the state-action distribution. Thus, the weighting in Eq. (7) loses a connection with the importance sampling ratio.

## C  Proof of Results in Section 4

Recall the objective of BC in Eq. (1):

$$\pi^{BC} \in \arg\max_\pi \sum_{h=1}^H \sum_{(s,a)\in\mathcal{S}\times\mathcal{A}} \widehat{d_h^E}(s,a)\log\pi_h(a|s),$$

where $\widehat{d_h^E}(s,a) = n_h^E(s,a)/N_{\text{tot}}$ is the empirical state-action distribution in the expert dataset, and $n_h^E(s,a)$ is the number of expert trajectories such that their state-action pairs are equal to $(s,a)$ in time step $h$. With the tabular representations, we can obtain a closed-formed solution to the above optimization problem.

$$
\pi_h^{BC}(a|s) = \begin{cases} \frac{n_h^E(s,a)}{n_h^E(s)} & \text{if } n_h^E(s) > 0 \\ \frac{1}{|\mathcal{A}|} & \text{otherwise} \end{cases}
\tag{8}
$$

where $n_h^E(s) \triangleq \sum_{a'} n_h^E(s,a')$. Analogously, we also have a closed-form solution for NBCU in the tabular setting:

$$
\pi_h^{NBCU}(a|s) = \begin{cases} \frac{n_h^U(s,a)}{n_h^U(s)} & \text{if } n_h^U(s) > 0 \\ \frac{1}{|\mathcal{A}|} & \text{otherwise} \end{cases}
\tag{9}
$$

We will discuss the generalization performance of NBCU later.

In the proof, we frequently use the notation $\lesssim$ and $\gtrsim$. In particular, $a(n) \lesssim b(n)$ means that there exist $C, n_0 > 0$ such that $a(n) \leq Cb(n)$ for all $n \geq n_0$. In our context, $n$ usually refers to the number of trajectories. For any two distributions $P$ and $Q$ over a finite set $\mathcal{X}$, we define the total variation distance as

$$\text{TV}(P,Q) = \frac{1}{2}\sum_{x\in\mathcal{X}} |P(x) - Q(x)| = \|P - Q\|_1.$$

## C.1  Proof of Theorem 1

When $|\mathcal{D}^{\mathrm{E}}| \geq 1$, by [42, Theorem 4.2], we have the following imitation gap bound for BC:

$$V(\pi^{\mathrm{E}}) - \mathbb{E}_{\mathcal{D}^{\mathrm{E}}}\left[V(\pi^{\mathrm{BC}})\right] \leq \frac{4|\mathcal{S}|H^2}{9|\mathcal{D}^{\mathrm{E}}|}.$$

When $|\mathcal{D}^{\mathrm{E}}| = 0$, we simply have that

$$V(\pi^{\mathrm{E}}) - \mathbb{E}_{\mathcal{D}^{\mathrm{E}}}\left[V(\pi^{\mathrm{BC}})\right] \leq H.$$

Therefore, we have the following unified bound.

$$V(\pi^{\mathrm{E}}) - \mathbb{E}_{\mathcal{D}^{\mathrm{E}}}\left[V(\pi^{\mathrm{BC}})\right] \leq \frac{|\mathcal{S}|H^2}{\max\{|\mathcal{D}^{\mathrm{E}}|, 1\}} \leq \frac{2|\mathcal{S}|H^2}{|\mathcal{D}^{\mathrm{E}}| + 1}.$$

The last inequality follows that $\max\{x, 1\} \geq (x + 1)/2$ for any $x \geq 0$. Finally, notice that $|\mathcal{D}^{\mathrm{E}}|$ follows a binomial distribution by Assumption 1, i.e., $|\mathcal{D}^{\mathrm{E}}| \sim \mathrm{Bin}(N_{\mathrm{tot}}, \eta)$. By Lemma 3, we have that $\mathbb{E}[1/(|\mathcal{D}|^E + 1)] \leq N_{\mathrm{tot}}\eta$, so

$$V(\pi^{\mathrm{E}}) - \mathbb{E}\left[V(\pi^{\mathrm{BC}})\right] \leq \mathbb{E}\left[\frac{2|\mathcal{S}|H^2}{|\mathcal{D}^{\mathrm{E}}| + 1}\right] \leq \frac{2|\mathcal{S}|H^2}{N_{\mathrm{tot}}\eta} = \frac{2|\mathcal{S}|H^2}{N_{\mathrm{E}}},$$

which completes the proof.

## C.2  Proof of Theorem 2

For analysis, we first define the mixture state-action distribution as follows.

$$d_h^{\mathrm{mix}}(s, a) \triangleq \eta d_h^{\pi^{\mathrm{E}}}(s, a) + (1 - \eta)d_h^{\pi^{\beta}}(s, a),$$
$$d_h^{\mathrm{mix}}(s) \triangleq \sum_{a \in \mathcal{A}} d_h^{\mathrm{mix}}(s, a), \ \forall (s, a) \in \mathcal{S} \times \mathcal{A}, \ \forall h \in [H].$$

By Assumption 1, in the population level, the marginal state-action distribution of union dataset $\mathcal{D}^{\mathrm{U}}$ in time step $h$ is exactly $d_h^{\mathrm{mix}}$. That is, $d_h^{\mathrm{U}}(s, a) = d_h^{\mathrm{mix}}(s, a), \ \forall (s, a, h) \in \mathcal{S} \times \mathcal{A} \times [H]$. Then we define the mixture policy $\pi^{\mathrm{mix}}$ induced by $d^{\mathrm{mix}}$ as follows.

$$\pi_h^{\mathrm{mix}}(a|s) = \begin{cases} \frac{d_h^{\mathrm{mix}}(s,a)}{d_h^{\mathrm{mix}}(s)} & \text{if } d_h^{\mathrm{mix}}(s) > 0, \\ \frac{1}{|\mathcal{A}|} & \text{otherwise.} \end{cases} \quad \forall (s, a) \in \mathcal{S} \times \mathcal{A}, \forall h \in [H]. \tag{10}$$

From the theory of Markov Decision Processes, we know that (see, e.g., [41])

$$\forall h \in [H], \forall (s, a) \in \mathcal{S} \times \mathcal{A}, \quad d_h^{\pi^{\mathrm{mix}}}(s, a) = d_h^{\mathrm{mix}}(s, a).$$

Therefore, we can obtain that the marginal state-action distribution of union dataset $\mathcal{D}^{\mathrm{U}}$ in time step $h$ is exactly $d_h^{\pi^{\mathrm{mix}}}$. Then we have the following decomposition.

$$\begin{aligned} \mathbb{E}\left[V(\pi^{\mathrm{E}}) - V(\pi^{\mathrm{NBCU}})\right] &= \mathbb{E}\left[V(\pi^{\mathrm{E}}) - V(\pi^{\mathrm{mix}}) + V(\pi^{\mathrm{mix}}) - V(\pi^{\mathrm{NBCU}})\right] \\ &= \mathbb{E}\left[V(\pi^{\mathrm{E}}) - V(\pi^{\mathrm{mix}})\right] + \mathbb{E}\left[V(\pi^{\mathrm{mix}}) - V(\pi^{\mathrm{NBCU}})\right] \\ &= V(\pi^{\mathrm{E}}) - V(\pi^{\mathrm{mix}}) + \mathbb{E}\left[V(\pi^{\mathrm{mix}}) - V(\pi^{\mathrm{NBCU}})\right]. \end{aligned}$$

For $V(\pi^{\mathrm{E}}) - V(\pi^{\mathrm{mix}})$, we have that

$$\begin{aligned} V(\pi^{\mathrm{E}}) - V(\pi^{\mathrm{mix}}) &= \sum_{h=1}^{H} \sum_{(s,a) \in \mathcal{S} \times \mathcal{A}} \left(d_h^{\pi^{\mathrm{E}}}(s, a) - d_h^{\pi^{\mathrm{mix}}}(s, a)\right) r_h(s, a) \\ &= \sum_{h=1}^{H} \sum_{(s,a) \in \mathcal{S} \times \mathcal{A}} \left(d_h^{\pi^{\mathrm{E}}}(s, a) - d_h^{\mathrm{mix}}(s, a)\right) r_h(s, a) \\ &= (1 - \eta) \sum_{h=1}^{H} \sum_{(s,a) \in \mathcal{S} \times \mathcal{A}} \left(d_h^{\pi^{\mathrm{E}}}(s, a) - d_h^{\pi^{\beta}}(s, a)\right) r_h(s, a) \\ &= (1 - \eta) \left(V(\pi^{\mathrm{E}}) - V(\pi^{\beta})\right). \end{aligned} \tag{11}$$

The last equation follows the dual formulation of policy value (see, e.g., [41]), i.e., $V(\pi) = \sum_{h=1}^{H} \sum_{(s,a)} d_h^\pi(s,a) r_h(s,a)$ for any policy $\pi$. Besides, notice that $\mathbb{E}\left[V(\pi^{\mathrm{mix}}) - V(\pi^{\mathrm{NBCU}})\right]$ is exactly the imitation gap of BC when regarding $\pi^{\mathrm{mix}}$ and $\mathcal{D}^{\mathrm{U}}$ as the expert policy and expert dataset, respectively. Note that $\pi^{\mathrm{mix}}$ may be a stochastic policy. By [42, Theorem 4.4], we have the following imtiation gap bound

$$\mathbb{E}\left[V(\pi^{\mathrm{mix}}) - V(\pi^{\mathrm{NBCU}})\right] \lesssim \frac{|\mathcal{S}|H^2 \log(N_{\mathrm{tot}})}{N_{\mathrm{tot}}}. \tag{12}$$

Combining Eq. (11) and Eq. (12) yields that

$$\mathbb{E}\left[V(\pi^{\mathrm{E}}) - V(\pi^{\mathrm{NBCU}})\right] \lesssim (1-\eta)\left(V(\pi^{\mathrm{E}}) - V(\pi^\beta)\right) + \frac{|\mathcal{S}|H^2 \log(N_{\mathrm{tot}})}{N_{\mathrm{tot}}}.$$

## C.3 Proof of Proposition 1

The hard instance in Proposition 1 builds on the Standard Imitation MDP proposed in [60]; see Figure 4 for illustration. For this MDP, each state is an absorbing state, i.e., $P_h(s|s,a) = 1$ for any $s$ and $a$. This property is mainly used to facilitate probability calculation and does not change the nature of our analysis. Furthermore, by only taking the action $a^1$ (shown in green), the agent can obtain a reward of $+1$. Otherwise, the agent obtains a reward of $0$ for the other action $a \neq a^1$. The initial state distribution is a uniform distribution, i.e., $\rho(s) = 1/|\mathcal{S}|$ for any $s \in \mathcal{S}$.

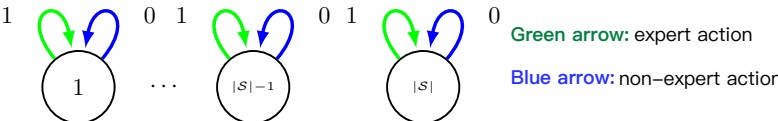

Figure 4: The Standard Imitation MDP in [60] corresponding to prove Proposition 1.

We consider that the expert policy $\pi^{\mathrm{E}}$ always takes the action $a^1$ (shown in green) while the behavioral policy $\pi^\beta$ always takes another action $a^2$ (shown in blue). Formally, $\pi_h^{\mathrm{E}}(a^1|s) = 1$ and $\pi_h^\beta(a^2|s) = 1$ for any $s \in \mathcal{S}$ and $h \in [H]$. It is direct to calculate that $V(\pi^{\mathrm{E}}) = H$ and $V(\pi^\beta) = 0$. The supplementary dataset $\mathcal{D}^{\mathrm{S}}$ and the expert dataset $\mathcal{D}^{\mathrm{E}}$ are collected according to Assumption 1. The mixture state-action distribution (introduced in Appendix C.2) can be calculated as for any $s \in \mathcal{S}$ and $h \in [H]$:

$$d_h^{\mathrm{mix}}(s,a^1) = \eta d_h^{\pi^{\mathrm{E}}}(s,a^1) + (1-\eta) d_h^{\pi^\beta}(s,a^1) = \eta d_h^{\pi^{\mathrm{E}}}(s,a^1) = \eta\rho(s),$$

$$d_h^{\mathrm{mix}}(s,a^2) = \eta d_h^{\pi^{\mathrm{E}}}(s,a^2) + (1-\eta) d_h^{\pi^\beta}(s,a^2) = (1-\eta) d_h^{\pi^\beta}(s,a^2) = (1-\eta)\rho(s).$$

Note that in the population level, the marginal distribution of the union dataset $\mathcal{D}^{\mathrm{U}}$ in time step $h$ is exactly $d_h^{\mathrm{mix}}$. The mixture policy induced by $d^{\mathrm{mix}}$ (introduced in Appendix C.2) can be formulated as

$$\pi_h^{\mathrm{mix}}(a^1|s) = \eta, \pi_h^{\mathrm{mix}}(a^2|s) = 1 - \eta, \forall s \in \mathcal{S}, h \in [H].$$

Just like before, we have $d_h^{\pi^{\mathrm{mix}}}(s,a) = d_h^{\mathrm{mix}}(s,a)$. The policy value of $\pi^{\mathrm{mix}}$ can be calculated as

$$V(\pi^{\mathrm{mix}}) = \sum_{h=1}^{H} \sum_{(s,a)\in\mathcal{S}\times\mathcal{A}} d_h^{\mathrm{mix}}(s,a) r_h(s,a) = \sum_{h=1}^{H} \sum_{s\in\mathcal{S}} d_h^{\mathrm{mix}}(s,a^1) = \eta H.$$

Recall from Eq. (9) that $\pi^{\mathrm{NBCU}}$ can be formulated as

$$\forall h \in [H], \quad \pi_h^{\mathrm{NBCU}}(a|s) = \begin{cases} \frac{n_h^{\mathrm{U}}(s,a)}{\sum_{a'} n_h^{\mathrm{U}}(s,a')} & \text{if } \sum_{a'} n_h^{\mathrm{U}}(s,a') > 0 \\ \frac{1}{|\mathcal{A}|} & \text{otherwise} \end{cases} \tag{13}$$

We can view that the BC's policy learned on the union dataset mimics the mixture policy $\pi^{\mathrm{mix}}$. In the following part, we analyze the lower bound on the imitation gap of $\pi^{\mathrm{NBCU}}$.

$$\mathbb{E}\left[V(\pi^{\mathrm{E}}) - V(\pi^{\mathrm{NBCU}})\right] = V(\pi^{\mathrm{E}}) - V(\pi^{\mathrm{mix}}) + \mathbb{E}\left[V(\pi^{\mathrm{mix}}) - V(\pi^{\mathrm{NBCU}})\right]$$

$$= H - \eta H + \mathbb{E}\left[V(\pi^{\mathrm{mix}}) - V(\pi^{\mathrm{NBCU}})\right]$$

$$= (1 - \eta)(V(\pi^{\mathrm{E}}) - V(\pi^{\beta})) + \mathbb{E}\left[V(\pi^{\mathrm{mix}}) - V(\pi^{\mathrm{NBCU}})\right].$$

Then we consider the term $\mathbb{E}\left[V(\pi^{\mathrm{mix}}) - V(\pi^{\mathrm{NBCU}})\right]$.

$$V(\pi^{\mathrm{mix}}) - V(\pi^{\mathrm{NBCU}})$$

$$= \sum_{h=1}^{H} \sum_{(s,a)\in\mathcal{S}\times\mathcal{A}} \left(d_h^{\pi^{\mathrm{mix}}}(s,a) - d_h^{\pi^{\mathrm{NBCU}}}(s,a)\right) r_h(s,a)$$

$$= \sum_{h=1}^{H} \sum_{(s,a)\in\mathcal{S}\times\mathcal{A}} \rho(s)\left(\pi_h^{\mathrm{mix}}(a|s) - \pi_h^{\mathrm{NBCU}}(a|s)\right) r_h(s,a)$$

$$= \sum_{h=1}^{H} \sum_{(s,a)\in\mathcal{S}\times\mathcal{A}} \rho(s)\left(\pi_h^{\mathrm{mix}}(a|s) - \pi_h^{\mathrm{NBCU}}(a|s)\right) r_h(s,a)\mathbb{I}\{n_h^{\mathrm{U}}(s) > 0\}$$

$$+ \sum_{h=1}^{H} \sum_{(s,a)\in\mathcal{S}\times\mathcal{A}} \rho(s)\left(\pi_h^{\mathrm{mix}}(a|s) - \pi_h^{\mathrm{NBCU}}(a|s)\right) r_h(s,a)\mathbb{I}\{n_h^{\mathrm{U}}(s) = 0\}.$$

We take expectation over the randomness in $\mathcal{D}^{\mathrm{U}}$ on both sides and obtain that

$$\mathbb{E}\left[V(\pi^{\mathrm{mix}}) - V(\pi^{\mathrm{NBCU}})\right] \tag{14}$$

$$= \mathbb{E}\left[\sum_{h=1}^{H} \sum_{(s,a)\in\mathcal{S}\times\mathcal{A}} \rho(s)\left(\pi_h^{\mathrm{mix}}(a|s) - \pi_h^{\mathrm{NBCU}}(a|s)\right) r_h(s,a)\mathbb{I}\{n_h^{\mathrm{U}}(s) > 0\}\right]$$

$$+ \mathbb{E}\left[\sum_{h=1}^{H} \sum_{(s,a)\in\mathcal{S}\times\mathcal{A}} \rho(s)\left(\pi_h^{\mathrm{mix}}(a|s) - \pi_h^{\mathrm{NBCU}}(a|s)\right) r_h(s,a)\mathbb{I}\{n_h^{\mathrm{U}}(s) = 0\}\right]. \tag{15}$$

For the first term in RHS, we have that

$$\mathbb{E}\left[\sum_{h=1}^{H} \sum_{(s,a)\in\mathcal{S}\times\mathcal{A}} \rho(s)\left(\pi_h^{\mathrm{mix}}(a|s) - \pi_h^{\mathrm{NBCU}}(a|s)\right) r_h(s,a)\mathbb{I}\{n_h^{\mathrm{U}}(s) > 0\}\right]$$

$$= \sum_{h=1}^{H} \sum_{(s,a)\in\mathcal{S}\times\mathcal{A}} \rho(s)r_h(s,a)\mathbb{E}\left[\left(\pi_h^{\mathrm{mix}}(a|s) - \pi_h^{\mathrm{NBCU}}(a|s)\right)\mathbb{I}\{n_h^{\mathrm{U}}(s) > 0\}\right]$$

$$= \sum_{h=1}^{H} \sum_{(s,a)\in\mathcal{S}\times\mathcal{A}} \rho(s)r_h(s,a)\mathbb{P}\left(n_h^{\mathrm{U}}(s) > 0\right) \mathbb{E}\left[\pi_h^{\mathrm{mix}}(a|s) - \pi_h^{\mathrm{NBCU}}(a|s) \mid n_h^{\mathrm{U}}(s) > 0\right]$$

$$= 0.$$

The last equation follows the fact that $\pi_h^{\mathrm{NBCU}}(a|s)$ is an unbiased estimation of $\pi_h^{\mathrm{mix}}(a|s)$, so $\mathbb{E}[\pi_h^{\mathrm{mix}}(a|s) - \pi_h^{\mathrm{NBCU}}(a|s) \mid n_h^{\mathrm{U}}(s) > 0]$. For the second term in Eq. (15), we have that

$$\mathbb{E}\left[\sum_{h=1}^{H} \sum_{(s,a)\in\mathcal{S}\times\mathcal{A}} \rho(s)\left(\pi_h^{\mathrm{mix}}(a|s) - \pi_h^{\mathrm{NBCU}}(a|s)\right) r_h(s,a)\mathbb{I}\{n_h^{\mathrm{U}}(s) = 0\}\right]$$

$$= \sum_{h=1}^{H} \sum_{(s,a)\in\mathcal{S}\times\mathcal{A}} \rho(s)r_h(s,a)\mathbb{E}\left[\left(\pi_h^{\mathrm{mix}}(a|s) - \pi_h^{\mathrm{NBCU}}(a|s)\right)\mathbb{I}\{n_h^{\mathrm{U}}(s) = 0\}\right]$$

$$= \sum_{h=1}^{H} \sum_{(s,a)\in\mathcal{S}\times\mathcal{A}} \rho(s)r_h(s,a)\mathbb{P}\left(n_h^{\mathrm{U}}(s) = 0\right) \mathbb{E}\left[\pi_h^{\mathrm{mix}}(a|s) - \pi_h^{\mathrm{NBCU}}(a|s) \mid n_h^{\mathrm{U}}(s) = 0\right]$$

$$= \sum_{h=1}^{H} \sum_{(s,a)\in\mathcal{S}\times\mathcal{A}} \rho(s)r_h(s,a)\mathbb{P}\left(n_h^{\mathrm{U}}(s) = 0\right) \left(\pi_h^{\mathrm{mix}}(a|s) - \frac{1}{|\mathcal{A}|}\right)$$

$$\stackrel{(a)}{=} \sum_{h=1}^{H} \sum_{s \in \mathcal{S}} \rho(s) \mathbb{P}\left(n_h^{\mathrm{U}}(s) = 0\right) \left(\eta - \frac{1}{|\mathcal{A}|}\right)$$

$$\stackrel{(b)}{=} H\left(\eta - \frac{1}{|\mathcal{A}|}\right) \sum_{s \in \mathcal{S}} \rho(s) \mathbb{P}\left(n_1^{\mathrm{U}}(s) = 0\right).$$

In the equation $(a)$, we use the fact that $r_h(s, a^1) = 1$ but $r_h(s, a) = 0$ for any $a \neq a^1$. In the equation $(b)$, since each state is an absorbing state, we have that $\mathbb{P}(n_h^{\mathrm{U}}(s) = 0) = \mathbb{P}(n_1^{\mathrm{U}}(s) = 0)$ for any $h \in [H]$. We consider two cases to address RHS of equation (b). In the first case of $\eta \geq 1/|\mathcal{A}|$, we directly have that

$$\mathbb{E}\left[\sum_{h=1}^{H} \sum_{(s,a) \in \mathcal{S} \times \mathcal{A}} \rho(s) \left(\pi_h^{\mathrm{mix}}(a|s) - \pi_h^{\mathrm{NBCU}}(a|s)\right) r_h(s,a) \mathbb{I}\{n_h^{\mathrm{U}}(s) = 0\}\right] \geq 0.$$

By Eq. (15), we have that

$$\mathbb{E}\left[V(\pi^{\mathrm{mix}}) - V(\pi^{\mathrm{NBCU}})\right] \geq 0,$$

which implies that

$$\mathbb{E}\left[V(\pi^{\mathrm{E}}) - V(\pi^{\mathrm{NBCU}})\right] \geq (1 - \eta)(V(\pi^{\mathrm{E}}) - V(\pi^{\beta})).$$

In the second case of $\eta < 1/|\mathcal{A}|$, we have that

$$H\left(\eta - \frac{1}{|\mathcal{A}|}\right) \sum_{s \in \mathcal{S}} \rho(s) \mathbb{P}\left(n_1^{\mathrm{U}}(s) = 0\right) \stackrel{(a)}{\geq} -\left(\frac{1}{|\mathcal{A}|} - \eta\right) H \exp\left(-\frac{N_{\mathrm{tot}}}{|\mathcal{S}|}\right)$$

$$\geq -(1 - \eta) H \exp\left(-\frac{N_{\mathrm{tot}}}{|\mathcal{S}|}\right)$$

$$\stackrel{(b)}{\geq} -\frac{(1 - \eta)H}{2}.$$

In the inequality $(a)$, we use that

$$\sum_{s \in \mathcal{S}} \rho(s) \mathbb{P}\left(n_1^{\mathrm{U}}(s) = 0\right) = \sum_{s \in \mathcal{S}} \rho(s)(1 - \rho(s))^{N_{\mathrm{tot}}} = \left(1 - \frac{1}{|\mathcal{S}|}\right)^{N_{\mathrm{tot}}} \leq \exp\left(-\frac{N_{\mathrm{tot}}}{|\mathcal{S}|}\right).$$

The inequality $(b)$ holds since we consider the range where $N_{\mathrm{tot}} \geq |\mathcal{S}| \log(2)$. By Eq. (15), we have that

$$\mathbb{E}\left[V(\pi^{\mathrm{mix}}) - V(\pi^{\mathrm{NBCU}})\right] \geq -\frac{(1 - \eta)H}{2}.$$

This implies that

$$\mathbb{E}\left[V(\pi^{\mathrm{E}}) - V(\pi^{\mathrm{NBCU}})\right] \geq (1 - \eta)(V(\pi^{\mathrm{E}}) - V(\pi^{\beta})) - \frac{(1 - \eta)H}{2}$$

$$= \frac{(1 - \eta)}{2}(V(\pi^{\mathrm{E}}) - V(\pi^{\beta})).$$

In both cases, we prove that $\mathbb{E}\left[V(\pi^{\mathrm{E}}) - V(\pi^{\mathrm{NBCU}})\right] \gtrsim (1 - \eta)(V(\pi^{\mathrm{E}}) - V(\pi^{\beta}))$ and thus complete the proof.

## D  Proof of Results in Section 5

### D.1  Proof of Proposition 2

In the tabular case, with the first-order optimality condition, we have $c_h^\star(s, a) = \widehat{d_h^{\mathrm{E}}}(s, a)/(\widehat{d_h^{\mathrm{E}}}(s, a) + \widehat{d_h^{\mathrm{U}}}(s, a))$. By Eq. (5), we have

$$\widehat{d_h^{\mathrm{U}}}(s, a) w_h(s, a) = \widehat{d_h^{\mathrm{U}}}(s, a) \times \frac{\widehat{d_h^{\mathrm{E}}}(s, a)}{\widehat{d_h^{\mathrm{U}}}(s, a)} = \widehat{d_h^{\mathrm{E}}}(s, a).$$

Hence, the learning objective (3) reduces to (1).

## D.2 Proof of Lemma 1

Recall that

$$\Delta_h(\theta) = \min_{(s,a)\in\mathcal{D}_h^{\mathrm{E}}\cup\mathcal{D}_h^{\mathrm{S},1}} \langle\theta, \phi_h(s,a)\rangle - \max_{(s',a')\in\mathcal{D}_h^{\mathrm{S},2}} \langle\theta, \phi_h(s',a')\rangle.$$

Then we have that

$$\Delta_h(\bar{\theta}_h) - \Delta_h(\theta) = \min_{(s,a)\in\mathcal{D}_h^{\mathrm{E}}\cup\mathcal{D}_h^{\mathrm{S},1}} \langle\bar{\theta}_h, \phi_h(s,a)\rangle - \max_{(s',a')\in\mathcal{D}_h^{\mathrm{S},2}} \langle\bar{\theta}_h, \phi_h(s',a')\rangle$$

$$- \min_{(s,a)\in\mathcal{D}_h^{\mathrm{E}}\cup\mathcal{D}_h^{\mathrm{S},1}} \langle\theta, \phi_h(s,a)\rangle + \max_{(s',a')\in\mathcal{D}_h^{\mathrm{S},2}} \langle\theta, \phi_h(s',a')\rangle$$

$$\overset{(a)}{\leq} \langle\bar{\theta}_h, \phi_h(s^1,a^1)\rangle - \langle\bar{\theta}_h, \phi_h(s^2,a^2)\rangle - \langle\theta, \phi_h(s^1,a^1)\rangle + \langle\theta, \phi_h(s^2,a^2)\rangle$$

$$= \langle\bar{\theta}_h - \theta, \phi_h(s^1,a^1) - \phi_h(s^2,a^2)\rangle$$

$$\overset{(b)}{\leq} \left\|\bar{\theta}_h - \theta\right\| \left\|\phi_h(s^1,a^1) - \phi_h(s^2,a^2)\right\|.$$

In inequality $(a)$, we utilize the facts that $(s^1, a^1) \in \operatorname{argmin}_{(s,a)\in\mathcal{D}_h^{\mathrm{E}}\cup\mathcal{D}_h^{\mathrm{S},1}} \langle\theta_h, \phi_h(s,a)\rangle$ and $(s^2, a^2) \in \operatorname{argmax}_{(s,a)\in\mathcal{D}_h^{\mathrm{S},2}} \langle\theta_h, \phi_h(s,a)\rangle$. Inequality $(b)$ follows the Cauchy–Schwarz inequality. Let $L_h = \left\|\phi_h(s^1,a^1) - \phi_h(s^2,a^2)\right\|$ and we finish the proof.

## D.3 Proof of Lemma 2

First, by Taylor's Theorem, there exists $\theta_h' \in \{\theta \in \mathbb{R}^d : \theta^t = \theta_h^\star + t(\bar{\theta}_h - \theta_h^\star), \ \forall t \in [0,1]\}$ such that

$$\mathcal{L}_h(\bar{\theta}_h) = \mathcal{L}_h(\theta_h^\star) + \langle\nabla\mathcal{L}_h(\theta_h^\star), \bar{\theta}_h - \theta_h^\star\rangle + \frac{1}{2}\left(\bar{\theta}_h - \theta_h^\star\right)^\top \nabla^2\mathcal{L}_h(\theta_h')\left(\bar{\theta}_h - \theta_h^\star\right)$$

$$= \mathcal{L}_h(\theta_h^\star) + \frac{1}{2}\left(\bar{\theta}_h - \theta_h^\star\right)^\top \nabla^2\mathcal{L}_h(\theta_h')\left(\bar{\theta}_h - \theta_h^\star\right). \tag{16}$$

The last equality follows the optimality condition that $\nabla\mathcal{L}_h(\theta_h^\star) = 0$. Then, our strategy is to prove that the smallest eigenvalue of the Hessian matrix $\nabla^2\mathcal{L}_h(\theta_h')$ is positive, i.e., $\lambda_{\min}(\nabla^2\mathcal{L}_h(\theta_h')) > 0$. We first calculate the Hessian matrix $\nabla^2\mathcal{L}_h(\theta_h')$. Given $\mathcal{D}^{\mathrm{E}}$ and $\mathcal{D}^{\mathrm{U}}$, we define the function $G : \mathbb{R}^{(|\mathcal{D}^{\mathrm{E}}|+|\mathcal{D}^{\mathrm{U}}|)} \to \mathbb{R}$ as

$$G(v) \triangleq \frac{1}{|\mathcal{D}^{\mathrm{E}}|}\sum_{i=1}^{|\mathcal{D}^{\mathrm{E}}|} g(v_i) + \frac{1}{|\mathcal{D}^{\mathrm{U}}|}\sum_{j=1}^{|\mathcal{D}^{\mathrm{U}}|} g(v_j),$$

where $v_i$ is the $i$-th element in the vector $v \in \mathbb{R}^{(|\mathcal{D}^{\mathrm{E}}|+|\mathcal{D}^{\mathrm{U}}|)}$ and $g(x) = \log\left(1 + \exp(x)\right)$ is a real-valued function. Besides, we use $B_h \in \mathbb{R}^{(|\mathcal{D}^{\mathrm{E}}|+|\mathcal{D}^{\mathrm{U}}|)\times d}$ to denote the matrix whose $i$-th row $B_{h,i} = -y_i\phi_h(s^i,a^i)^\top$, and $y_i = 1$ if $(s^i,a^i) \in \mathcal{D}_h^{\mathrm{E}}$, $y_i = -1$ if $(s^i,a^i) \notin \mathcal{D}_h^{\mathrm{E}}$. Then the objective function can be reformulated as

$$\mathcal{L}_h(\theta_h)$$

$$= \sum_{(s,a)} \widehat{d_h^{\mathrm{E}}}(s,a)\left[\log\left(1 + \exp\left(-\langle\phi_h(s,a),\theta_h\rangle\right)\right)\right] + \sum_{(s,a)} \widehat{d_h^{\mathrm{U}}}(s,a)\left[\log\left(1 + \exp\left(\langle\phi_h(s,a),\theta_h\rangle\right)\right)\right]$$

$$= \frac{1}{|\mathcal{D}^{\mathrm{E}}|}\sum_{(s,a)\in\mathcal{D}^{\mathrm{E}}} \log\left(1 + \exp\left(-\langle\phi_h(s,a),\theta_h\rangle\right)\right) + \frac{1}{|\mathcal{D}^{\mathrm{U}}|}\sum_{(s,a)\in\mathcal{D}^{\mathrm{U}}} \log\left(1 + \exp\left(\langle\phi_h(s,a),\theta_h\rangle\right)\right)$$

$$= G(B_h\theta_h).$$

Then we have that $\nabla^2\mathcal{L}_h(\theta_h) = B_h^\top \nabla^2 G(B_h\theta_h)B_h$, where

$$\nabla^2 G(B_h\theta_h)$$

$$= \mathbf{diag}\left(\frac{g''((B_h\theta_h)_1)}{|\mathcal{D}^{\mathrm{E}}|}, \dots, \frac{g''((B_h\theta_h)_{|\mathcal{D}^{\mathrm{E}}|})}{|\mathcal{D}^{\mathrm{E}}|}, \frac{g''((B_h\theta_h)_{|\mathcal{D}^{\mathrm{E}}|+1})}{|\mathcal{D}^{\mathrm{E}}| + |\mathcal{D}^{\mathrm{U}}|}, \dots, \frac{g''((B_h\theta_h)_{|\mathcal{D}^{\mathrm{E}}|+|\mathcal{D}^{\mathrm{U}}|})}{|\mathcal{D}^{\mathrm{E}}| + |\mathcal{D}^{\mathrm{U}}|}\right).$$

Here $g''(x) = \sigma(x)(1 - \sigma(x))$, where $\sigma(x) = 1/(1 + \exp(-x))$ is the sigmoid function. The eigenvalues of $\nabla^2 G(B_h \theta_h)$ are

$$\left\{ \frac{g''((B_h\theta_h)_1)}{|\mathcal{D}^{\mathrm{E}}|}, \ldots, \frac{g''((B_h\theta_h)_{|\mathcal{D}^{\mathrm{E}}|})}{|\mathcal{D}^{\mathrm{E}}|}, \frac{g''((B_h\theta_h)_{|\mathcal{D}^{\mathrm{E}}|+1})}{|\mathcal{D}^{\mathrm{E}}| + |\mathcal{D}^{\mathrm{U}}|}, \ldots, \frac{g''((B_h\theta_h)_{|\mathcal{D}^{\mathrm{E}}|+|\mathcal{D}^{\mathrm{U}}|})}{|\mathcal{D}^{\mathrm{E}}| + |\mathcal{D}^{\mathrm{U}}|} \right\}.$$

Notice that $\theta'_h \in \{\theta \in \mathbb{R}^d : \theta^t = \theta_h^\star + t(\bar{\theta}_h - \theta_h^\star), \forall t \in [0, 1]\}$. For a matrix $A$, we use $\lambda_{\min}(A)$ to denote the minimal eigenvalue of $A$. Here we claim that the minimum of the minimal eigenvalues of $\nabla^2 G(B_h \theta^t)$ over $t \in [0, 1]$ is achieved at $t = 0$ or $t = 1$. That is,

$$\min\{\lambda_{\min}(\nabla^2 G(B_h \theta^t)) : \forall t \in [0, 1]\} = \min\{\lambda_{\min}(\nabla^2 G(B_h \theta^0)), \lambda_{\min}(\nabla^2 G(B_h \theta^1))\}.$$

We prove this claim as follows. For any $t \in [0, 1]$, we use $\{\lambda_1(t), \ldots, \lambda_{|\mathcal{D}^{\mathrm{E}}|+|\mathcal{D}^{\mathrm{U}}|}(t)\}$ to denote the eigenvalues of $\nabla^2 G(B_h \theta^t)$. For each $i \in [|\mathcal{D}^{\mathrm{E}}| + |\mathcal{D}^{\mathrm{U}}|]$, we consider $\lambda_i(t) : [0, 1] \to \mathbb{R}$ as a function of $t$. Specifically,

$$\lambda_i(t) = \begin{cases} \frac{g''((B_h\theta_h^\star)_i + t(B_h(\bar{\theta}_h - \theta_h^\star))_i)}{|\mathcal{D}^{\mathrm{E}}|}, & \text{if } i \in [|\mathcal{D}^{\mathrm{E}}|] \\ \frac{g''((B_h\theta_h^\star)_i + t(B_h(\bar{\theta}_h - \theta_h^\star))_i)}{|\mathcal{D}^{\mathrm{E}}| + |\mathcal{D}^{\mathrm{U}}|}, & \text{otherwise.} \end{cases}$$

We observe that $g'''(x) = \sigma(x)(1 - \sigma(x))(1 - 2\sigma(x))$ which satisfies that $\forall x \leq 0$, $g'''(x) \geq 0$, and $\forall x \geq 0$, $g'''(x) \leq 0$. Therefore, we have that the minimum of $\lambda_i(t)$ over $t \in [0, 1]$ must be achieved at $t = 0$ or $t = 1$. That is,

$$\min_{t \in [0,1]} \lambda_i(t) = \min\{\lambda_i(0), \lambda_i(1)\}. \tag{17}$$

For any $t \in [0, 1]$, we define $i^t \in [|\mathcal{D}^{\mathrm{E}}| + |\mathcal{D}^{\mathrm{U}}|]$ as the index of the minimal eigenvalue of $\nabla^2 G(B_h \theta^t)$, i.e., $\lambda_{i^t}(t) = \lambda_{\min}(\nabla^2 G(B_h \theta^t))$. Then we have that

$$\min\{\lambda_{\min}(\nabla^2 G(B_h \theta^t)) : \forall t \in [0, 1]\} = \min\{\lambda_{i^t}(t) : \forall t \in [0, 1]\}$$

$$\overset{(a)}{=} \min\{\min\{\lambda_{i^t}(0), \lambda_{i^t}(1)\} : \forall t \in [0, 1]\}$$

$$= \min\{\lambda_{i^0}(0), \lambda_{i^1}(1)\}$$

$$\overset{(b)}{=} \min\{\lambda_{\min}(\nabla^2 G(B_h \theta^0)), \lambda_{\min}(\nabla^2 G(B_h \theta^1))\}$$

Equality $(a)$ follows (17) and equality $(b)$ follows that $\lambda_{i^0}(0)$ and $\lambda_{i^1}(1)$ are the minimal eigenvalues of $\nabla^2 G(B_h \theta^0)$ and $\nabla^2 G(B_h \theta^1)$, respectively.

In summary, we derive that

$$\min\{\lambda_{\min}(\nabla^2 G(B_h \theta^t)) : \forall t \in [0, 1]\} = \min\{\lambda_{\min}(\nabla^2 G(B_h \theta^0)), \lambda_{\min}(\nabla^2 G(B_h \theta^1))\}, \tag{18}$$

which proves the previous claim.

Further, we consider $\lambda_{\min}\left(\nabla^2 \mathcal{L}_h(\theta_h)\right)$.

$$\lambda_{\min}\left(\nabla^2 \mathcal{L}_h(\theta_h)\right) = \inf_{x \in \mathbb{R}^d : \|x\| = 1} x^\top \nabla^2 \mathcal{L}_h(\theta_h) x$$

$$= \inf_{x \in \mathbb{R}^d : \|x\| = 1} (B_h x)^\top \nabla^2 G(B_h \theta_h)(B_h x)$$

$$= \inf_{z \in \mathrm{Im}(B_h)} z^\top \nabla^2 G(B_h \theta_h) z$$

$$= \left( \inf_{z \in \mathrm{Im}(B_h)} \|z\| \right)^2 \lambda_{\min}(\nabla^2 G(B_h \theta_h))$$

$$\geq \left( \inf_{z \in \mathrm{Im}(B_h)} \|z\| \right)^2 \min\{\lambda_{\min}(\nabla^2 G(B_h \theta^0)), \lambda_{\min}(\nabla^2 G(B_h \theta^1))\}.$$

Here $\mathrm{Im}(B_h) = \{z \in \mathbb{R}^d : z = B_h x, \|x\| = 1\}$. The last inequality follows Eq. (18).

Recall we assume that $\mathbf{rank}(A_h) = d$, so we have that $\mathbf{rank}(B_h) = d$. Thus, $\mathrm{Im}(B_h)$ is a set of vectors with positive norms, i.e., $\inf_{z \in \mathrm{Im}(B_h)} \|z\| > 0$. Besides, since $g''(x) = \sigma(x)(1 - \sigma(x)) > 0$, we also have that

$$\min\{\lambda_{\min}(\nabla^2 G(B_h \theta^0)), \lambda_{\min}(\nabla^2 G(B_h \theta^1))\} > 0.$$

In summary, we obtain that

$$\lambda_{\min}\left(\nabla^2 \mathcal{L}_h(\theta_h)\right) \geq \left(\inf_{z \in \text{Im}(B_h)} \|z\|\right)^2 \min\{\lambda_{\min}(\nabla^2 G(B_h \theta^0)), \lambda_{\min}(\nabla^2 G(B_h \theta^1))\} > 0.$$

Then, with Eq. (16), there exists

$$\tau_h = \left(\inf_{z \in \text{Im}(B_h)} \|z\|\right)^2 \min\{\lambda_{\min}(\nabla^2 G(B_h \theta^0)), \lambda_{\min}(\nabla^2 G(B_h \theta^1))\} > 0$$

such that

$$\mathcal{L}_h(\bar{\theta}_h) \geq \mathcal{L}_h(\theta_h^\star) + \frac{\tau_h}{2} \left\|\bar{\theta}_h - \theta_h^\star\right\|^2,$$

which completes the proof.

### D.4 Proof of Theorem 3

First, invoking Lemma 1 with $\theta = \theta_h^\star$ yields that

$$\Delta_h(\theta_h^\star) \geq \Delta_h(\bar{\theta}_h) - L_h \left\|\bar{\theta}_h - \theta_h^\star\right\|.$$

Here $L_h = \|\phi_h(s,a) - \phi_h(s',a')\|$ with $(s,a) \in \text{argmin}_{(s,a) \in \mathcal{D}_h^{\text{E}} \cup \mathcal{D}_h^{\text{S},1}} \langle \theta_h^\star, \phi_h(s,a) \rangle$ and $(s',a') \in \text{argmax}_{(s,a) \in \mathcal{D}_h^{\text{S},2}} \langle \theta_h^\star, \phi_h(s,a) \rangle$. Then, by Lemma 2, there exists $\tau_h > 0$ such that

$$\mathcal{L}_h(\bar{\theta}_h) \geq \mathcal{L}_h(\theta_h^\star) + \frac{\tau_h}{2} \left\|\bar{\theta}_h - \theta_h^\star\right\|^2.$$

This directly implies an upper bound of the distance between $\bar{\theta}_h$ and $\theta_h^\star$.

$$\left\|\bar{\theta}_h - \theta_h^\star\right\| \leq \sqrt{\frac{2\left(\mathcal{L}_h(\bar{\theta}_h) - \mathcal{L}_h(\theta_h^\star)\right)}{\tau_h}}.$$

If the feature is designed such that $\sqrt{\frac{2\left(\mathcal{L}_h(\bar{\theta}_h) - \mathcal{L}_h(\theta_h^\star)\right)}{\tau_h}} < \frac{\Delta_h(\bar{\theta}_h)}{L_h}$ holds, we further have that $\left\|\bar{\theta}_h - \theta_h^\star\right\| < \Delta_h(\bar{\theta}_h)/L_h$. Then we get that

$$\Delta_h(\theta_h^\star) \geq \Delta_h(\bar{\theta}_h) - L_h \left\|\bar{\theta}_h - \theta_h^\star\right\| > 0,$$

which completes the proof of the first statement.

Then we proceed to prove the imitation gap bound. We first identify the property of $\pi^{\text{ISW-BC}}$. Recall the objective of WBCU.

$$\pi^{\text{ISW-BC}} \in \text{argmax}_\pi \sum_{h=1}^{H} \sum_{(s,a) \in \mathcal{S} \times \mathcal{A}} \left\{\widehat{d_h^{\text{U}}}(s,a) \times [w_h(s,a) \log \pi_h(a|s)] \times \mathbb{I}\left[w_h(s,a) \geq \delta\right]\right\}.$$

For any state $s$ with $\sum_{a \in \mathcal{A}} \widehat{d_h^{\text{U}}}(s,a) w_h(s,a) \mathbb{I}\left[w_h(s,a) \geq \delta\right] > 0$, with the first-order optimality condition, we have

$$\pi_h^{\text{ISW-BC}}(a|s) = \frac{\widehat{d_h^{\text{U}}}(s,a) w_h(s,a) \mathbb{I}\left[w_h(s,a) \geq \delta\right]}{\sum_{a \in \mathcal{A}} \widehat{d_h^{\text{U}}}(s,a) w_h(s,a) \mathbb{I}\left[w_h(s,a) \geq \delta\right]}.$$

For an expert state $s$ with $d_h^{\pi^{\text{E}}}(s) > 0$, if $(s, \pi_h^{\text{E}}(s)) \in \mathcal{D}_h^{\text{E}} \cup \mathcal{D}_h^{\text{S},1}$, we have that

$$\langle \theta_h^\star, \phi_h(s, \pi_h^{\text{E}}(s)) \rangle > \langle \theta_h^\star, \phi_h(s,a) \rangle, \quad \forall (s,a) \in \mathcal{D}_h^{\text{S},2}.$$

This is due to the first statement that $\Delta_h(\theta_h^\star) > 0$ in this theorem. Recall that

$$c_h(s,a;\theta_h^\star) = \frac{1}{1 + \exp(-\langle \phi_h(s,a), \theta_h^\star \rangle)} \quad \text{and} \quad w_h(s,a) = \frac{c_h(s,a;\theta_h^\star)}{1 - c_h(s,a;\theta_h^\star)}.$$

We can further obtain that $w_h(s, \pi_h^{\text{E}}(s)) > w_h(s,a)$ for any $(s,a) \in \mathcal{D}_h^{\text{S},2}$. This implies that we can find a $\delta$ such that $\mathbb{I}\left[w_h(s, \pi_h^{\text{E}}(s)) \geq \delta\right] = 1$ for any $(s, \pi_h^{\text{E}}(s)) \in \mathcal{D}_h^{\text{E}} \cup \mathcal{D}_h^{\text{S},1}$ and $\mathbb{I}\left[w_h(s,a) \geq \delta\right] = 0$ for any $(s,a) \in \mathcal{D}_h^{\text{S},2}$. Based on the above analytical form of $\pi^{\text{ISW-BC}}$,

we have that $\pi_h^{\text{ISW-BC}}(\pi_h^{\text{E}}(s)|s) = 1$ for any $(s, \pi_h^{\text{E}}(s)) \in \mathcal{D}_h^{\text{E}} \cup \mathcal{D}_h^{\text{S},1}$. In summary, for any state $s$ with $(s, \pi_h^{\text{E}}(s)) \in \mathcal{D}_h^{\text{E}} \cup \mathcal{D}_h^{\text{S},1}$, we have that $\pi_h^{\text{ISW-BC}}(\pi_h^{\text{E}}(s)|s) = 1$.

With the above property of $\pi^{\text{ISW-BC}}$, we proceed to analyze the policy value gap. According to [42, Lemma 4.3], we have

$$V(\pi^{\text{E}}) - V(\pi^{\text{ISW-BC}}) \leq H \sum_{h=1}^{H} \mathbb{E}_{s \sim d_h^{\pi^{\text{E}}}(\cdot)} \left[ \text{TV}\left(\pi_h^{\text{E}}(\cdot|s), \pi_h^{\text{ISW-BC}}(\cdot|s)\right)\right].$$

Since $\pi^{\text{E}}$ is assumed to be deterministic, we have

$$V(\pi^{\text{E}}) - V(\pi^{\text{ISW-BC}}) \leq H \sum_{h=1}^{H} \mathbb{E}_{s \sim d_h^{\pi^{\text{E}}}(\cdot)} \left[ \mathbb{E}_{a \sim \pi_h^{\text{ISW-BC}}(\cdot|s)} \left[ \mathbb{I}\left\{a \neq \pi_h^{\text{E}}(s)\right\}\right]\right]$$

$$\overset{(a)}{\leq} H \sum_{h=1}^{H} \mathbb{E}_{s \sim d_h^{\pi^{\text{E}}}(\cdot)} \left[ \mathbb{I}\left\{(s, \pi_h^{\text{E}}(s)) \notin \mathcal{D}_h^{\text{E}} \cup \mathcal{D}_h^{\text{S},1}\right\}\right]$$

$$\overset{(b)}{=} H \sum_{h=1}^{H} \mathbb{E}_{s \sim d_h^{\pi^{\text{E}}}(\cdot)} \left[ \mathbb{I}\left\{(s, \pi_h^{\text{E}}(s)) \notin \mathcal{D}_h^{\text{U}}\right\}\right].$$

Inequality $(a)$ follows the property of $\pi^{\text{ISW-BC}}$ derived above. In particular, for any state $s$ with $(s, \pi_h^{\text{E}}(s)) \in \mathcal{D}_h^{\text{E}} \cup \mathcal{D}_h^{\text{S},1}$, we have that $\pi_h^{\text{ISW-BC}}(\pi_h^{\text{E}}(s)|s) = 1$. Equation $(b)$ holds due to the Assumption 2. In particular, for an expert state $s$ that $d_h^{\pi^{\text{E}}}(s) > 0$, the events of $(s, \pi_h^{\text{E}}(s)) \notin \mathcal{D}_h^{\text{E}} \cup \mathcal{D}_h^{\text{S},1}$ and $(s, \pi_h^{\text{E}}(s)) \notin \mathcal{D}_h^{\text{U}}$ are equivalent.

Moreover, we take the expectation over $\mathcal{D}^{\text{U}}$ on both sides and obtain that

$$\mathbb{E}\left[V(\pi^{\text{E}}) - V(\pi^{\text{ISW-BC}})\right] \leq H \sum_{h=1}^{H} \mathbb{E}_{s \sim d_h^{\pi^{\text{E}}}(\cdot)} \left[ \mathbb{P}\left((s, \pi_h^{\text{E}}(s)) \notin \mathcal{D}_h^{\text{U}}\right)\right]$$

$$= H \sum_{h=1}^{H} \sum_{s \in \mathcal{S}} d_h^{\pi^{\text{E}}}(s) \mathbb{P}\left((s, \pi_h^{\text{E}}(s)) \notin \mathcal{D}_h^{\text{U}}\right).$$

According to Assumption 1, we have that

$$d_h^{\text{U}}(s, \pi_h^{\text{E}}(s)) = \eta d_h^{\pi^{\text{E}}}(s, \pi_h^{\text{E}}(s)) + (1 - \eta) d_h^{\pi^{\beta}}(s, \pi_h^{\text{E}}(s))$$

$$\overset{(a)}{\geq} \eta d_h^{\pi^{\text{E}}}(s, \pi_h^{\text{E}}(s)) + \frac{(1-\eta)}{\mu} d_h^{\pi^{\text{E}}}(s, \pi_h^{\text{E}}(s))$$

$$= \left(\eta + \frac{(1-\eta)}{\mu}\right) d_h^{\pi^{\text{E}}}(s, \pi_h^{\text{E}}(s)).$$

Inequality $(a)$ follows the definition of $\mu$ in Theorem 3: for any $(s, h) \in \mathcal{S} \times [H]$, we have $d_h^{\pi^{\text{E}}}(s, \pi_h^{\text{E}}(s))/d_h^{\pi^{\beta}}(s, \pi_h^{\text{E}}(s)) \leq \mu$. Then we obtain that

$$\mathbb{E}\left[V(\pi^{\text{E}}) - V(\pi^{\text{ISW-BC}})\right] \leq H \sum_{h=1}^{H} \sum_{s \in \mathcal{S}} d_h^{\pi^{\text{E}}}(s)(1 - d_h^{\text{U}}(s, \pi_h^{\text{E}}(s)))^{N_{\text{tot}}}$$

$$\leq \left(\frac{1}{\eta + (1-\eta)/\mu}\right) H \sum_{h=1}^{H} \sum_{s \in \mathcal{S}} d_h^{\text{U}}(s, \pi_h^{\text{E}}(s)) \mathbb{P}\left((s, \pi_h^{\text{E}}(s)) \notin \mathcal{D}_h^{\text{U}}\right).$$

For each $(s, h) \in \mathcal{S} \times [H]$, we observe that

$$d_h^{\text{U}}(s, \pi_h^{\text{E}}(s)) \mathbb{P}\left((s, \pi_h^{\text{E}}(s)) \notin \mathcal{D}_h^{\text{U}}\right) = d_h^{\text{U}}(s, \pi_h^{\text{E}}(s))\left(1 - d_h^{\text{U}}(s, \pi_h^{\text{E}}(s))\right)^{N_{\text{tot}}} \leq \frac{4}{9 N_{\text{tot}}}.$$

Here the last inequality follows Lemma 5. Consequently, we can derive that

$$\sum_{h=1}^{H} \sum_{s \in \mathcal{S}} d_h^{\text{U}}(s, \pi_h^{\text{E}}(s)) \mathbb{P}\left((s, \pi_h^{\text{E}}(s)) \notin \mathcal{D}_h^{\text{U}}\right) \leq \frac{4H|\mathcal{S}|}{9 N_{\text{tot}}},$$

which further implies that

$$\mathbb{E}\left[V(\pi^{\mathrm{E}}) - V(\pi^{\mathrm{ISW\text{-}BC}})\right] \leq \left(\frac{1}{\eta + (1-\eta)/\mu}\right)\frac{4H^2|\mathcal{S}|}{9N_{\mathrm{tot}}} = \frac{4H^2|\mathcal{S}|}{9\left(N_{\mathrm{E}} + N_{\mathrm{S}}/\mu\right)}.$$

We complete the proof.

### D.5   An Example Corresponding to Theorem 3

In this section, we provide an example that illustrates the required feature design in Theorem 3 can hold.

**Example 1.** *To illustrate Theorem 3, we consider an example in the feature space $\mathbb{R}^2$. In particular, for time step $h \in [H]$, we have the expert dataset and supplementary dataset as follows.*

$$\mathcal{D}_h^{\mathrm{E}} = \left\{\left(s^{(1)}, a^{(1)}\right), \left(s^{(4)}, a^{(4)}\right)\right\}, \ \mathcal{D}_h^{\mathrm{S}} = \left\{\left(s^{(2)}, a^{(2)}\right), \left(s^{(3)}, a^{(3)}\right)\right\},$$

$$\mathcal{D}_h^{\mathrm{S},1} = \left\{\left(s^{(2)}, a^{(2)}\right)\right\}, \ \mathcal{D}_h^{\mathrm{S},2} = \left\{\left(s^{(3)}, a^{(3)}\right)\right\}.$$

*The corresponding features are*

$$\phi_h\left(s^{(1)}, a^{(1)}\right) = (0,1)^\top, \ \phi_h\left(s^{(2)}, a^{(2)}\right) = \left(-\frac{1}{2}, 0\right)^\top,$$

$$\phi_h\left(s^{(3)}, a^{(3)}\right) = \left(0, -\frac{1}{2}\right)^\top, \ \phi_h\left(s^{(4)}, a^{(4)}\right) = (-1,0)^\top.$$

*Notice that the set of expert-style samples is $\mathcal{D}_h^{\mathrm{E}} \cup \mathcal{D}_h^{\mathrm{S},1} = \{(s^{(1)}, a^{(1)}), (s^{(2)}, a^{(2)}), (s^{(4)}, a^{(4)})\}$ and the set of non-expert-style samples is $\mathcal{D}_h^{\mathrm{S},2} = \{(s^{(3)}, a^{(3)})\}$. It is direct to calculate that the ground-truth parameter that achieves the maximum margin among unit vectors is $\bar{\theta}_h = (-\sqrt{2}/2, \sqrt{2}/2)^\top$ and the maximum margin is $\Delta_h(\bar{\theta}_h) = \sqrt{2}/2$. According to Eq. (6), for $\theta_h = (\theta_{h,1}, \theta_{h,2})^\top$, the optimization objective is*

$$\mathcal{L}_h(\theta_h)$$
$$= \sum_{(s,a)} \widehat{d_h^{\mathrm{E}}}(s,a)\left[\log\left(1 + \exp\left(-\langle\phi_h(s,a), \theta_h\rangle\right)\right)\right] + \sum_{(s,a)} \widehat{d_h^{\mathrm{U}}}(s,a)\left[\log\left(1 + \exp\left(\langle\phi_h(s,a), \theta_h\rangle\right)\right)\right]$$

$$= \frac{1}{2}\left(\log\left(1 + \exp\left(-\theta_{h,2}\right)\right) + \log\left(1 + \exp\left(\theta_{h,1}\right)\right)\right)$$

$$+ \frac{1}{4}\left(\log\left(1 + \exp\left(\theta_{h,2}\right)\right) + \log\left(1 + \exp\left(-\frac{1}{2}\theta_{h,1}\right)\right)\right)$$

$$+ \frac{1}{4}\left(\log\left(1 + \exp\left(-\frac{1}{2}\theta_{h,2}\right)\right) + \log\left(1 + \exp\left(-\theta_{h,1}\right)\right)\right).$$

*We apply CVXPY [12] to calculate the optimal solution $\theta_h^\star \approx (-0.310, 0.993)^\top$ and the objective values $\mathcal{L}_h(\theta_h^\star) \approx 1.287$, $\mathcal{L}_h(\bar{\theta}_h) \approx 1.309$. Furthermore, we calculate the Lipschitz coefficient $L_h$ appears in Lemma 1.*

$$(s^{(2)}, a^{(2)}) = \underset{(s,a)\in\mathcal{D}_h^{\mathrm{E}}\cup\mathcal{D}_h^{\mathrm{S},1}}{\mathrm{argmin}} \langle\theta_h^\star, \phi_h(s,a)\rangle, \ (s^{(3)}, a^{(3)}) \in \underset{(s,a)\in\mathcal{D}_h^{\mathrm{S},2}}{\mathrm{argmax}} \langle\theta_h^\star, \phi_h(s,a)\rangle,$$

$$L_h = \left\|\phi_h(s^{(2)}, a^{(2)}) - \phi_h(s^{(3)}, a^{(3)})\right\| = \frac{\sqrt{2}}{2}.$$

*Then we calculate the parameter of strong convexity $\tau_h$ appears in Lemma 2. Based on the proof of Lemma 2, our strategy is to calculate the minimal eigenvalue of the Hessian matrix.*

*First, for $\theta_h = (\theta_{h,1}, \theta_{h,2})^\top$, the gradient of $\mathcal{L}_h(\theta_h)$ is*

$$\nabla\mathcal{L}_h(\theta_h)$$
$$= -\sum_{(s,a)\in\mathcal{S}\times\mathcal{A}} \widehat{d_h^{\mathrm{E}}}(s,a)\sigma(-\langle\phi_h(s,a), \theta_h\rangle) + \sum_{(s,a)\in\mathcal{S}\times\mathcal{A}} \widehat{d_h^{\mathrm{U}}}(s,a)\sigma\left(\langle\phi_h(s,a), \theta_h\rangle\right)$$

$$= \left( \frac{1}{2}\sigma(\theta_{h,1}) - \frac{1}{4}\sigma(-\theta_{h,1}) - \frac{1}{8}\sigma(-\frac{1}{2}\theta_{h,1}), \frac{1}{4}\sigma(\theta_{h,2}) - \frac{1}{2}\sigma(-\theta_{h,2}) - \frac{1}{8}\sigma(-\frac{1}{2}\theta_{h,2}) \right)^{\top}.$$

*Here $\sigma(x) = 1/(1 + \exp(-x))$ for $x \in \mathbb{R}$ is the sigmoid function. Then the Hessian matrix at $\theta_h$ is*

$$\nabla^2 \mathcal{L}_h(\theta_h) = \begin{pmatrix} \frac{3}{4}f(\theta_{h,1}) + \frac{1}{16}f\left(\frac{1}{2}\theta_{h,1}\right) & 0 \\ 0 & \frac{3}{4}f(\theta_{h,2}) + \frac{1}{16}f\left(\frac{1}{2}\theta_{h,2}\right) \end{pmatrix},$$

*where $f(x) = \sigma(x)(1 - \sigma(x))$ and $f(x) = f(-x)$. For any $t \in [0,1]$, the eigenvalues of the Hessian matrix at $\theta_h^t = \bar{\theta}_h + t(\theta_h^{\star} - \bar{\theta}_h)$ are*

$$\frac{3}{4}f(\theta_{h,1}^t) + \frac{1}{16}f\left(\frac{1}{2}\theta_{h,1}^t\right), \ \frac{3}{4}f(\theta_{h,2}^t) + \frac{1}{16}f\left(\frac{1}{2}\theta_{h,2}^t\right).$$

*Now, we calculate the minimal eigenvalues of $\nabla^2 \mathcal{L}_h(\theta_h^t)$. We consider the function*

$$g(x) = \frac{3}{4}f(x) + \frac{1}{16}f\left(\frac{1}{2}x\right), \ \forall x \in [a,b].$$

*The gradient is*

$$g'(x) = \frac{3}{4}\sigma(x)(1 - \sigma(x))(1 - 2\sigma(x)) + \frac{1}{32}\sigma\left(\frac{1}{2}x\right)\left(1 - \sigma\left(\frac{1}{2}x\right)\right)\left(1 - 2\sigma\left(\frac{1}{2}x\right)\right).$$

*We observe that $\forall x \leq 0$, $g'(x) \geq 0$, and $\forall x \geq 0$, $g'(x) \leq 0$. Thus, we have that the minimum of $g(x)$ must be achieved at $x = a$ or $x = b$. Besides, we have that $g(x) = g(-x)$. With the above arguments, we know that the minimal eigenvalue is $g(0.993) \approx 0.163$ and $\tau_h \approx 0.163$. Then we can calculate that*

$$\sqrt{\frac{2\left(\mathcal{L}_h(\bar{\theta}_h) - \mathcal{L}_h(\theta_h^{\star})\right)}{\tau_h}} \approx 0.520, \ \frac{\Delta_h(\bar{\theta}_h)}{L_h} = 1.$$

*The inequality in Theorem 3 holds.*

# E  Discussion

In the main text, we focus on the tabular representations for policies. Furthermore, we consider a trajectory sampling procedure for behavior policy in collecting the supplementary dataset. We present two possible extensions in this section.

## E.1  Function Approximation of Policies

In the main text, the theoretical analysis for BC-based algorithms considers the tabular setting in policy learning where a table function represents the policy. Here we provide an analysis of BC with *general function approximation* in policy learning. Notice that the algorithms considered in this paper (i.e., BC, NBCU and ISW-BC) can be unified under the framework of maximum likelihood estimation (MLE)[8]. Therefore, the theoretical results in the main text can also be extended to the setting of general function approximation by a similar analysis.

Assume that the learner is access to a finite function class $\Pi = \{\pi = (\pi_1, \pi_2, \ldots, \pi_h)\}$, where $\pi_h : \mathcal{S} \to \Delta(\mathcal{A})$ could be any function (e.g., neural networks). For simplicity of analysis, we assume that $\Pi$ is a finite class. Notice that the algorithms considered in this paper are BC and its variants, which all take the principle of maximum likelihood estimation (MLE). The theoretical analysis of these algorithms is based on the following inequality:

$$V(\pi^{\mathrm{E}}) - V(\pi) \leq H \sum_{h=1}^{H} \mathbb{E}_{s \sim d_h^{\pi^{\mathrm{E}}}(\cdot)} \left[ \mathrm{TV}\left(\pi_h^{\mathrm{E}}(\cdot|s), \pi_h(\cdot|s)\right) \right].$$

Therefore, the key is to upper bound the TV distance. Take BC as an example (i.e., $\pi = \pi^{\mathrm{BC}}$). By using the concentration inequality in [1, Theorem 21], we obtain that for any $\delta \in (0,1)$, when

---

[8]Among these algorithms, the main difference is the weight function in the MLE objective; see Equations (1), (2) and (7).

$|\mathcal{D}^{\mathrm{E}}| \geq 1$, with probability at least $1 - \delta$ over the randomness within $\mathcal{D}^{\mathrm{E}}$,

$$\mathbb{E}_{s \sim d_h^{\pi^{\mathrm{E}}}(\cdot)} \left[ \mathrm{TV}^2 \left( \pi_h^{\mathrm{E}}(\cdot|s), \pi_h^{\mathrm{BC}}(\cdot|s) \right) \right] \leq 2 \frac{\log(|\Pi|/\delta)}{|\mathcal{D}^{\mathrm{E}}|}. \tag{19}$$

With additional efforts (by using union bound and Jensen's inequality), we have the following result.

**Theorem 4** (BC with Function Approximation). *Under Assumption 1. In the general function approximation setting, additionally assume that $\pi^{\mathrm{E}} \in \Pi$. If we apply BC on the expert data, we have*

$$\mathbb{E} \left[ V(\pi^{\mathrm{E}}) - V(\pi^{\mathrm{BC}}) \right] = \mathcal{O} \left( H^2 \sqrt{\frac{\log(|\Pi| H N_{\mathrm{E}})}{N_{\mathrm{E}}}} \right),$$

*where the expectation is taken over the randomness in the dataset collection.*

The detailed proof is deferred to Appendix F. Compared with Theorem 1, we notice that the change in theoretical bound is that $\mathcal{O}(|\mathcal{S}|/N_{\mathrm{E}})$ is replaced by $\mathcal{O}(\sqrt{\log(|\Pi| H N_{\mathrm{E}})/N_{\mathrm{E}}})$. Such a change is expected for other algorithms (e.g., NBCU and WBCU), so our theoretical implications still hold. We leave the detailed analysis for future works.

## E.2 Supplementary Data with Corruption

In the main text, we consider the trajectory sampling procedure in Assumption 1. However, in some cases, the supplementary data can be poisoned and corrupted by an adversary. For example, although the human expert demonstrates an optimal trajectory, the recorder or the recording system possibly corrupts the data by accident or on purpose. Data corruption is one of the main security threats to imitation learning methods [31]. Therefore, it is valuable to investigate the robustness of the presented algorithms in this poison setting. Supplementary data with corruption is partially investigated in our experiments under the noisy expert setting, which we argue have a large state-action distribution shift.

**Assumption 3** (Poison Setting). *The supplementary dataset $\mathcal{D}^{\mathrm{S}}$ and expert dataset $\mathcal{D}^{\mathrm{E}}$ are collected in the following way: each time, with probability $\eta$, we rollout the expert policy to collect a trajectory. With probability $1 - \eta$, we still rollout the expert policy to collect a trajectory but with probability $1 - \eta'$, the actions along the sampled trajectory are replaced with actions uniformly sampled from the action space. Such an experiment is independent and identically conducted by $N_{\mathrm{tot}}$ times.*

**Theorem 5** (NBCU in the Poison Setting). *Under Assumption 3. In the tabular case, for any $\eta \in (0, 1]$, we have*

$$\mathbb{E} \left[ V(\pi^{\mathrm{E}}) - V(\pi^{\mathrm{NBCU}}) \right] = \mathcal{O} \left( (1 - \eta)(1 - \eta') H^2 \left( 1 - \frac{1}{|\mathcal{A}|} \right) + H^2 \sqrt{\frac{|\mathcal{S}||\mathcal{A}|}{N_{\mathrm{tot}}}} \right),$$

*where the expectation is taken over the randomness in the dataset collection.*

**Theorem 6** (ISW-BC in the Poison Setting). *Under Assumptions 2 and 3, if the feature is designed such that $\sqrt{\frac{2(\mathcal{L}_h(\bar{\theta}_h) - \mathcal{L}_h(\theta_h^\star))}{\tau_h}} < \frac{\Delta_h(\bar{\theta}_h)}{L_h}$ holds, we have the imitation gap bound*

$$\mathbb{E}[V(\pi^{\mathrm{E}}) - V(\pi^{\mathrm{ISW-BC}})] = \mathcal{O} \left( \frac{H^2 |\mathcal{S}|}{N_{\mathrm{E}} + N_{\mathrm{S}} \eta'} \right).$$

Proofs of Theorem 5 and Theorem 6 can be found in Appendix F. Compared with the imitation gap of NBCU, there is no non-vanishing gap due to the corrupted actions in the imitation gap of ISW-BC. This means that ISW-BC is still robust in this setting.

# F Proof of Results in Section E

## F.1 Proof of Theorem 4

According to [42, Lemma 4.3], we have

$$V(\pi^{\mathrm{E}}) - V(\pi^{\mathrm{BC}}) \leq H \sum_{h=1}^{H} \mathbb{E}_{s \sim d_h^{\pi^{\mathrm{E}}}(\cdot)} \left[ \mathrm{TV} \left( \pi_h^{\mathrm{E}}(\cdot|s), \pi_h^{\mathrm{BC}}(\cdot|s) \right) \right].$$

With [1, Theorem 21], when $|\mathcal{D}^{\mathrm{E}}| \geq 1$, for any $\delta \in (0,1)$, with probability at least $1 - \delta$ over the randomness within $\mathcal{D}^{\mathrm{E}}$, we have that

$$\mathbb{E}_{s \sim d_h^{\pi^{\mathrm{E}}}(\cdot)} \left[ \mathrm{TV}^2 \left( \pi_h^{\mathrm{E}}(\cdot|s), \pi_h^{\mathrm{BC}}(\cdot|s) \right) \right] \leq 2 \frac{\log(|\Pi|/\delta)}{|\mathcal{D}^{\mathrm{E}}|}.$$

With union bound, with probability at least $1 - \delta$, for all $h \in [H]$, it holds that

$$\mathbb{E}_{s \sim d_h^{\pi^{\mathrm{E}}}(\cdot)} \left[ \mathrm{TV}^2 \left( \pi_h^{\mathrm{E}}(\cdot|s), \pi_h^{\mathrm{BC}}(\cdot|s) \right) \right] \leq 2 \frac{\log(|\Pi|H/\delta)}{|\mathcal{D}^{\mathrm{E}}|},$$

which implies that

$$\begin{aligned}
V(\pi^{\mathrm{E}}) - V(\pi^{\mathrm{BC}}) &\leq H \sum_{h=1}^{H} \mathbb{E}_{s \sim d_h^{\pi^{\mathrm{E}}}(\cdot)} \left[ \mathrm{TV} \left( \pi_h^{\mathrm{E}}(\cdot|s), \pi_h^{\mathrm{BC}}(\cdot|s) \right) \right] \\
&\overset{(a)}{\leq} H \sum_{h=1}^{H} \sqrt{\mathbb{E}_{s \sim d_h^{\pi^{\mathrm{E}}}(\cdot)} \left[ \mathrm{TV}^2 \left( \pi_h^{\mathrm{E}}(\cdot|s), \pi_h^{\mathrm{BC}}(\cdot|s) \right) \right]} \\
&\leq \sqrt{2} H^2 \sqrt{\frac{\log(|\Pi|H/\delta)}{|\mathcal{D}^{\mathrm{E}}|}}.
\end{aligned}$$

Inequality $(a)$ follows Jensen's inequality. Taking expectation over the randomness within $\mathcal{D}^{\mathrm{E}}$ yields that

$$\begin{aligned}
\mathbb{E}_{\mathcal{D}^{\mathrm{E}}} \left[ V(\pi^{\mathrm{E}}) - V(\pi^{\mathrm{BC}}) \right] &\leq \delta H + (1 - \delta) \sqrt{2} H^2 \sqrt{\frac{\log(|\Pi|H/\delta)}{|\mathcal{D}^{\mathrm{E}}|}} \\
&\overset{(a)}{=} \frac{H}{2|\mathcal{D}^{\mathrm{E}}|} + \left( 1 - \frac{1}{2|\mathcal{D}^{\mathrm{E}}|} \right) \sqrt{2} H^2 \sqrt{\frac{\log(2|\Pi|H|\mathcal{D}^{\mathrm{E}}|)}{|\mathcal{D}^{\mathrm{E}}|}} \\
&\leq \left( \sqrt{2} + 1 \right) H^2 \sqrt{\frac{\log(2|\Pi|H|\mathcal{D}^{\mathrm{E}}|)}{|\mathcal{D}^{\mathrm{E}}|}} \\
&\leq 4 H^2 \sqrt{\frac{\log(4|\Pi|H|\mathcal{D}^{\mathrm{E}}|)}{|\mathcal{D}^{\mathrm{E}}|}}.
\end{aligned}$$

Equation $(a)$ holds due to the choice that $\delta = 1/(2|\mathcal{D}^{\mathrm{E}}|)$. For $|\mathcal{D}^{\mathrm{E}}| = 0$, we directly have that

$$\mathbb{E}_{\mathcal{D}^{\mathrm{E}}} \left[ V(\pi^{\mathrm{E}}) - V(\pi^{\mathrm{BC}}) \right] \leq H.$$

Therefore, for any $|\mathcal{D}^{\mathrm{E}}| \geq 0$, we have that

$$\mathbb{E}_{\mathcal{D}^{\mathrm{E}}} \left[ V(\pi^{\mathrm{E}}) - V(\pi^{\mathrm{BC}}) \right] \leq 4 H^2 \sqrt{\frac{\log(4|\Pi|H \max\{|\mathcal{D}^{\mathrm{E}}|, 1\})}{\max\{|\mathcal{D}^{\mathrm{E}}|, 1\}}}.$$

We consider a real-valued function $f(x) = \log(cx)/x$ for $x \geq 1$, where $c = 4|\Pi|H > 4$. Its gradient function is $f'(x) = (1 - \log(cx))/x^2 \leq 0$ when $x \geq 1$. Then we know that $f(x)$ is decreasing as $x$ increases. Furthermore, we have that $\max\{|\mathcal{D}^{\mathrm{E}}|, 1\} \geq (|\mathcal{D}^{\mathrm{E}}| + 1)/2$ when $|\mathcal{D}^{\mathrm{E}}| \geq 0$. Then we obtain

$$\begin{aligned}
\mathbb{E}_{\mathcal{D}^{\mathrm{E}}} \left[ V(\pi^{\mathrm{E}}) - V(\pi^{\mathrm{BC}}) \right] &\leq 4 H^2 \sqrt{\frac{\log(4|\Pi|H \max\{|\mathcal{D}^{\mathrm{E}}|, 1\})}{\max\{|\mathcal{D}^{\mathrm{E}}|, 1\}}} \\
&\leq 4 H^2 \sqrt{\frac{2 \log(4|\Pi|H(|\mathcal{D}^{\mathrm{E}}| + 1))}{|\mathcal{D}^{\mathrm{E}}| + 1}}.
\end{aligned}$$

Taking expectation over the random variable $|\mathcal{D}^{\mathrm{E}}| \sim \mathrm{Bin}(N_{\mathrm{tot}}, \eta)$ yields that

$$\mathbb{E} \left[ V(\pi^{\mathrm{E}}) - V(\pi^{\mathrm{BC}}) \right] \leq 4 H^2 \mathbb{E} \left[ \sqrt{\frac{2 \log(4|\Pi|H(|\mathcal{D}^{\mathrm{E}}| + 1))}{|\mathcal{D}^{\mathrm{E}}| + 1}} \right]$$

$$\overset{(a)}{\leq} 4H^2 \sqrt{\mathbb{E}\left[\frac{2\log(4|\Pi|H(|\mathcal{D}^{\mathrm{E}}|+1))}{|\mathcal{D}^{\mathrm{E}}|+1}\right]}.$$

Inequality $(a)$ follows Jensen's inequality. We consider the function $g(x) = -x\log(x/c)$ for $x \in (0,1]$, where $c = 4|\Pi|H$.

$$g'(x) = -(\log(x/c)+1) \geq 0, \ g''(x) = -\frac{1}{x} \leq 0, \quad \forall x \in (0,1].$$

Thus, $g(x)$ is a concave function. By Jensen's inequality, we have that $\mathbb{E}[g(x)] \leq g(\mathbb{E}[x])$. Then we can derive that

$$\mathbb{E}\left[V(\pi^{\mathrm{E}}) - V(\pi^{\mathrm{BC}})\right] \leq 4H^2 \sqrt{\mathbb{E}\left[\frac{2\log(4|\Pi|H(|\mathcal{D}^{\mathrm{E}}|+1))}{|\mathcal{D}^{\mathrm{E}}|+1}\right]}$$

$$= 4\sqrt{2}H^2 \sqrt{\mathbb{E}\left[g\left(\frac{1}{|\mathcal{D}^{\mathrm{E}}|+1}\right)\right]}$$

$$\leq 4\sqrt{2}H^2 \sqrt{g\left(\mathbb{E}\left[\frac{1}{|\mathcal{D}^{\mathrm{E}}|+1}\right]\right)}$$

$$\overset{(a)}{\leq} 4\sqrt{2}H^2 \sqrt{g\left(\frac{1}{N_{\mathrm{E}}}\right)}$$

$$\leq 4\sqrt{2}H^2 \sqrt{\frac{\log(4|\Pi|HN_{\mathrm{E}})}{N_{\mathrm{E}}}}.$$

In inequality $(a)$, we use the facts that $g'(x) \geq 0$ and $\mathbb{E}\left[1/(|\mathcal{D}^{\mathrm{E}}|+1)\right] \leq 1/N_{\mathrm{E}}$ from Lemma 3. We complete the proof.

### F.2 Proof of Theorem 5

We first analyze the data distribution in $\mathcal{D}^{\mathrm{U}}$. According to Assumption 3, we summarize the sampling procedure of trajectories in $\mathcal{D}^{\mathrm{U}}$ as follows. Each time, we rollout the expert policy to collect a trajectory. Furthermore, with the probability of $(1-\eta)(1-\eta')$, the actions along the sampled expert trajectory are replaced with actions uniformly sampled from the action space. Then we put this poisoned expert trajectory into $\mathcal{D}^{\mathrm{U}}$. Otherwise, with the probability of $1-(1-\eta)(1-\eta')$, we directly put the original expert trajectory into $\mathcal{D}^{\mathrm{U}}$. Therefore, we can formulate the marginal distribution of the state-action pairs in time step $h$ in $\mathcal{D}^{\mathrm{U}}$. For each $(s,a,h) \in \mathcal{S} \times \mathcal{A} \times [H]$,

$$d_h^{\mathrm{U}}(s,a) = (1-(1-\eta)(1-\eta'))\, d_h^{\pi^{\mathrm{E}}}(s,a) + (1-\eta)(1-\eta')d_h^{\pi^{\mathrm{E}}}(s)\frac{1}{|\mathcal{A}|},$$

$$d_h^{\mathrm{U}}(s) = \sum_{a \in \mathcal{A}} d_h^{\mathrm{U}}(s,a) = d_h^{\pi^{\mathrm{E}}}(s).$$

Then we proceed to analyze the imitation gap. Similar to the proof of Theorem 2, according to [42, Lemma 4.3], we have

$$V(\pi^{\mathrm{E}}) - V(\pi^{\mathrm{NBCU}}) \leq H \sum_{h=1}^{H} \mathbb{E}_{s \sim d_h^{\pi^{\mathrm{E}}}(\cdot)} \left[\mathrm{TV}\left(\pi_h^{\mathrm{E}}(\cdot|s), \pi_h^{\mathrm{NBCU}}(\cdot|s)\right)\right].$$

Again, we introduce the definition of the policy $\pi^{\mathrm{mix}}$.

$$\forall (s,a) \in \mathcal{S} \times \mathcal{A}, \forall h \in [H], \ \pi_h^{\mathrm{mix}}(a|s) = \begin{cases} \frac{d_h^{\mathrm{U}}(s,a)}{d_h^{\mathrm{U}}(s)} & \text{if } d_h^{\mathrm{U}}(s) = d_h^{\pi^{\mathrm{E}}}(s) > 0, \\ \frac{1}{|\mathcal{A}|} & \text{otherwise.} \end{cases}$$

In particular, if $d_h^{\mathrm{U}}(s) > 0$, we have that

$$\pi_h^{\mathrm{mix}}(a|s) = \frac{d_h^{\mathrm{U}}(s,a)}{d_h^{\mathrm{U}}(s)} = (1-(1-\eta)(1-\eta'))\, \pi_h^{\mathrm{E}}(a|s) + (1-\eta)(1-\eta')\frac{1}{|\mathcal{A}|}.$$

Then we decompose the imitation gap into two parts.

$$V(\pi^{\mathrm{E}}) - V(\pi^{\mathrm{NBCU}})$$

$$\leq H \sum_{h=1}^{H} \mathbb{E}_{s \sim d_h^{\pi^{\mathrm{E}}}(\cdot)} \left[ \mathrm{TV} \left( \pi_h^{\mathrm{E}}(\cdot|s), \pi_h^{\mathrm{NBCU}}(\cdot|s) \right) \right]$$

$$\leq H \sum_{h=1}^{H} \mathbb{E}_{s \sim d_h^{\pi^{\mathrm{E}}}(\cdot)} \left[ \mathrm{TV} \left( \pi_h^{\mathrm{E}}(\cdot|s), \pi_h^{\mathrm{mix}}(\cdot|s) \right) \right] + H \sum_{h=1}^{H} \mathbb{E}_{s \sim d_h^{\pi^{\mathrm{E}}}(\cdot)} \left[ \mathrm{TV} \left( \pi_h^{\mathrm{mix}}(\cdot|s), \pi_h^{\mathrm{NBCU}}(\cdot|s) \right) \right].$$

We first analyze the first term in RHS. For certain $(s, h)$ such $d_h^{\mathrm{U}}(s) = d_h^{\pi^{\mathrm{E}}}(s) > 0$, we have that

$$\mathrm{TV} \left( \pi_h^{\mathrm{E}}(\cdot|s), \pi_h^{\mathrm{mix}}(\cdot|s) \right) = \sum_{a \neq \pi_h^{\mathrm{E}}(s)} \pi_h^{\mathrm{mix}}(a|s)$$

$$= \sum_{a \neq \pi_h^{\mathrm{E}}(s)} \left( 1 - (1-\eta)(1-\eta') \right) \pi_h^{\mathrm{E}}(a|s) + (1-\eta)(1-\eta') \frac{1}{|\mathcal{A}|}$$

$$= (1-\eta)(1-\eta') \left( 1 - \frac{1}{|\mathcal{A}|} \right).$$

Therefore, we can derive that

$$H \sum_{h=1}^{H} \mathbb{E}_{s \sim d_h^{\pi^{\mathrm{E}}}(\cdot)} \left[ \mathrm{TV} \left( \pi_h^{\mathrm{E}}(\cdot|s), \pi_h^{\mathrm{mix}}(\cdot|s) \right) \right] \leq (1-\eta)(1-\eta') H^2 \left( 1 - \frac{1}{|\mathcal{A}|} \right).$$

Now we analyze the second term of

$$H \sum_{h=1}^{H} \mathbb{E}_{s \sim d_h^{\pi^{\mathrm{E}}}(\cdot)} \left[ \mathrm{TV} \left( \pi_h^{\mathrm{mix}}(\cdot|s), \pi_h^{\mathrm{NBCU}}(\cdot|s) \right) \right].$$

Recall the formula of $\pi^{\mathrm{NBCU}}$.

$$\pi_h^{\mathrm{NBCU}}(a|s) = \begin{cases} \frac{n_h^{\mathrm{U}}(s,a)}{n_h^{\mathrm{U}}(s)} & \text{if } n_h^{\mathrm{U}}(s) > 0 \\ \frac{1}{|\mathcal{A}|} & \text{otherwise} \end{cases}$$

Notice that $\pi^{\mathrm{NBCU}}$ is the maximum likelihood estimation of $\pi^{\mathrm{mix}}$. According to the concentration inequality of total variation [53], for each $(s, h) \in \mathcal{S} \times [H]$, for any fixed $\delta \in (0, 1)$, when $n_h^{\mathrm{U}}(s) > 0$, with probability at least $1 - \delta$, we have

$$\mathrm{TV} \left( \pi_h^{\mathrm{mix}}(\cdot|s), \pi_h^{\mathrm{NBCU}}(\cdot|s) \right) \leq \sqrt{\frac{|\mathcal{A}| \log(3/\delta)}{n_h^{\mathrm{U}}(s)}}.$$

When $n_h^{\mathrm{U}}(s) = 0$, we have that

$$\mathrm{TV} \left( \pi_h^{\mathrm{mix}}(\cdot|s), \pi_h^{\mathrm{NBCU}}(\cdot|s) \right) \leq 1 \leq \sqrt{|\mathcal{A}| \log(3/\delta)}.$$

By combining the above two inequalities, for each $(s, h) \in \mathcal{S} \times [H]$, with probability at least $1 - \delta$, we have

$$\mathrm{TV} \left( \pi_h^{\mathrm{mix}}(\cdot|s), \pi_h^{\mathrm{NBCU}}(\cdot|s) \right) \leq \sqrt{\frac{|\mathcal{A}| \log(3/\delta)}{\max\{n_h^{\mathrm{U}}(s), 1\}}}.$$

Applying union bound yields that with probability at least $1 - \delta/2$, for all $(s, h) \in \mathcal{S} \times [H]$,

$$\mathrm{TV} \left( \pi_h^{\mathrm{mix}}(\cdot|s), \pi_h^{\mathrm{NBCU}}(\cdot|s) \right) \leq \sqrt{\frac{|\mathcal{A}| \log(6|\mathcal{S}|H/\delta)}{\max\{n_h^{\mathrm{U}}(s), 1\}}}.$$

Then we have that

$$H \sum_{h=1}^{H} \mathbb{E}_{s \sim d_h^{\pi^{\mathrm{E}}}(\cdot)} \left[ \mathrm{TV} \left( \pi_h^{\mathrm{mix}}(\cdot|s), \pi_h^{\mathrm{NBCU}}(\cdot|s) \right) \right]$$

$$\leq H \sum_{h=1}^{H} \mathbb{E}_{s \sim d_h^{\pi^{\mathrm{E}}}(\cdot)} \left[ \sqrt{\frac{|\mathcal{A}| \log(6|\mathcal{S}|H/\delta)}{\max\{n_h^{\mathrm{U}}(s), 1\}}} \right]$$

$$= H \sqrt{|\mathcal{A}| \log(6|\mathcal{S}|H/\delta)} \sum_{h=1}^{H} \mathbb{E}_{s \sim d_h^{\pi^{\mathrm{E}}}(\cdot)} \left[ \sqrt{\frac{1}{\max\{n_h^{\mathrm{U}}(s), 1\}}} \right]$$

$$= H \sqrt{|\mathcal{A}| \log(6|\mathcal{S}|H/\delta)} \sum_{h=1}^{H} \sum_{s \in \mathcal{S}} \sqrt{d_h^{\pi^{\mathrm{E}}}(s)} \sqrt{\frac{d_h^{\pi^{\mathrm{E}}}(s)}{\max\{n_h^{\mathrm{U}}(s), 1\}}}$$

$$\leq H \sqrt{|\mathcal{A}| \log(6|\mathcal{S}|H/\delta)} \sum_{h=1}^{H} \sqrt{\sum_{s \in \mathcal{S}} \frac{d_h^{\pi^{\mathrm{E}}}(s)}{\max\{n_h^{\mathrm{U}}(s), 1\}}}.$$

Here the last inequality follows Cauchy-Swartz inequality. Notice that $n_h^{\mathrm{U}}(s)$ is the number of times that the state $s$ appears in $\mathcal{D}^{\mathrm{U}}$ in time step $h$ and thus follows the Binomial distribution of $\mathrm{Bin}(N_{\mathrm{tot}}, d_h^{\pi^{\mathrm{E}}}(s))$. By applying Lemma 4, for each $(s, h)$, with probability at least $1 - \delta$, we have

$$\frac{d_h^{\pi^{\mathrm{E}}}(s)}{\max\{n_h^{\mathrm{U}}(s), 1\}} \leq \frac{8 \log(1/\delta)}{N_{\mathrm{tot}}}.$$

By union bound, with probability at least $1 - \delta/2$, for all $(s, h) \in \mathcal{S} \times [H]$,

$$\frac{d_h^{\pi^{\mathrm{E}}}(s)}{\max\{n_h^{\mathrm{U}}(s), 1\}} \leq \frac{8 \log(2|\mathcal{S}|H/\delta)}{N_{\mathrm{tot}}}.$$

Then, with probability at least $1 - \delta$, we have

$$H \sum_{h=1}^{H} \mathbb{E}_{s \sim d_h^{\pi^{\mathrm{E}}}(\cdot)} \left[ \mathrm{TV}\left( \pi_h^{\mathrm{mix}}(\cdot|s), \pi_h^{\mathrm{NBCU}}(\cdot|s) \right) \right] \leq H^2 \sqrt{\frac{8|\mathcal{S}||\mathcal{A}| \log^2(6|\mathcal{S}|H/\delta)}{N_{\mathrm{tot}}}}.$$

Finally, we upper bound the imitation gap. With probability at least $1 - \delta$, we have

$$V(\pi^{\mathrm{E}}) - V(\pi^{\mathrm{NBCU}}) \leq (1-\eta)(1-\eta')\left(1 - \frac{1}{|\mathcal{A}|}\right) + H^2 \sqrt{\frac{8|\mathcal{S}||\mathcal{A}| \log^2(6|\mathcal{S}|H/\delta)}{N_{\mathrm{tot}}}}.$$

We set $\delta = H/N_{\mathrm{tot}}$ and obtain that

$$\mathbb{E}\left[ V(\pi^{\mathrm{E}}) - V(\pi^{\mathrm{NBCU}}) \right]$$

$$\leq \delta H + (1-\delta)\left( (1-\eta)(1-\eta')\left(1 - \frac{1}{|\mathcal{A}|}\right) + H^2 \sqrt{\frac{8|\mathcal{S}||\mathcal{A}| \log^2(6|\mathcal{S}|H/\delta)}{N_{\mathrm{tot}}}} \right)$$

$$\leq \frac{H^2}{N_{\mathrm{tot}}} + (1-\eta)(1-\eta')\left(1 - \frac{1}{|\mathcal{A}|}\right) + H^2 \sqrt{\frac{8|\mathcal{S}||\mathcal{A}| \log^2(6|\mathcal{A}|N_{\mathrm{tot}})}{N_{\mathrm{tot}}}}$$

$$\leq (1-\eta)(1-\eta')\left(1 - \frac{1}{|\mathcal{A}|}\right) + 4H^2 \sqrt{\frac{2|\mathcal{S}||\mathcal{A}| \log^2(6|\mathcal{A}|N_{\mathrm{tot}})}{N_{\mathrm{tot}}}}.$$

On the other hand, we directly have $\mathbb{E}[V(\pi^{\mathrm{E}}) - V(\pi^{\mathrm{NBCU}})] \leq H$. We complete the proof.

### F.3 Proof of Theorem 6

In the poison setting, we can conduct the same analysis as in the proof of Theorem 3 and demonstrate that $\pi^{\mathrm{ISW\text{-}BC}}(\pi_h^{\mathrm{E}}(s)|s) = 1$, $\forall (s, \pi_h^{\mathrm{E}}(s)) \in \mathcal{D}_h^{\mathrm{E}} \cup \mathcal{D}_h^{\mathrm{S},1}$, where $\mathcal{D}_h^{\mathrm{E}}$ is the set of state-action pairs in $\mathcal{D}^{\mathrm{E}}$ in time step $h$ and $\mathcal{D}_h^{\mathrm{S},1} = \{(s, a) \in \mathcal{D}_h^{\mathrm{S}} : d_h^{\pi^{\mathrm{E}}}(s) > 0, a = \pi_h^{\mathrm{E}}(s)\}$. According to [42, Lemma 4.3], we have

$$V(\pi^{\mathrm{E}}) - V(\pi^{\mathrm{ISW\text{-}BC}}) \leq H \sum_{h=1}^{H} \mathbb{E}_{s \sim d_h^{\pi^{\mathrm{E}}}(\cdot)} \left[ \mathrm{TV}\left( \pi_h^{\mathrm{E}}(\cdot|s), \pi_h^{\mathrm{ISW\text{-}BC}}(\cdot|s) \right) \right].$$

Since the expert policy is assumed to be deterministic, we can obtain

$$V(\pi^{\mathrm{E}}) - V(\pi^{\mathrm{ISW\text{-}BC}}) \le H \sum_{h=1}^{H} \mathbb{E}_{s \sim d_h^{\pi^{\mathrm{E}}}(\cdot)} \left[ \mathbb{E}_{a \sim \pi_h^{\mathrm{ISW\text{-}BC}}(\cdot|s)} \left[ \mathbb{I}\left\{ a \ne \pi_h^{\mathrm{E}}(s) \right\} \right] \right]$$

$$\le H \sum_{h=1}^{H} \mathbb{E}_{s \sim d_h^{\pi^{\mathrm{E}}}(\cdot)} \left[ \mathbb{I}\left\{ (s, \pi_h^{\mathrm{E}}(s)) \notin \mathcal{D}_h^{\mathrm{E}} \cup \mathcal{D}_h^{\mathrm{S},1} \right\} \right].$$

Let $\mathcal{D}^{\mathrm{S,clean}}$ denote the non-corrupted dataset in $\mathcal{D}^{\mathrm{S}}$. Then we can obtain that

$$V(\pi^{\mathrm{E}}) - V(\pi^{\mathrm{ISW\text{-}BC}}) \overset{(a)}{\le} H \sum_{h=1}^{H} \mathbb{E}_{s \sim d_h^{\pi^{\mathrm{E}}}(\cdot)} \left[ \mathbb{I}\left\{ (s, \pi_h^{\mathrm{E}}(s)) \notin \mathcal{D}_h^{\mathrm{E}} \cup \mathcal{D}_h^{\mathrm{S,clean}} \right\} \right]$$

$$= H \sum_{h=1}^{H} \sum_{s \in \mathcal{S}} d_h^{\pi^{\mathrm{E}}}(s) \mathbb{I}\left\{ (s, \pi_h^{\mathrm{E}}(s)) \notin \mathcal{D}_h^{\mathrm{E}} \cup \mathcal{D}_h^{\mathrm{S,clean}} \right\},$$

where $\mathcal{D}_h^{\mathrm{S,clean}}$ denotes the set of state-action pairs in $\mathcal{D}^{\mathrm{S,clean}}$ in time step $h$. Inequality $(a)$ follows that $\mathcal{D}_h^{\mathrm{S,clean}} \subseteq \mathcal{D}_h^{\mathrm{S},1}$ since $\mathcal{D}^{\mathrm{S,clean}}$ is collected by the expert policy. Taking expectation over the randomness in $\mathcal{D}^{\mathrm{E}}$ and $\mathcal{D}^{\mathrm{S,clean}}$ on both sides yields that

$$\mathbb{E}_{\mathcal{D}^{\mathrm{E}}, \mathcal{D}^{\mathrm{S,clean}}} \left[ V(\pi^{\mathrm{E}}) - V(\pi^{\mathrm{ISW\text{-}BC}}) \right] \le H \sum_{h=1}^{H} \sum_{s \in \mathcal{S}} d_h^{\pi^{\mathrm{E}}}(s) \mathbb{P}\left( (s, \pi_h^{\mathrm{E}}(s)) \notin \mathcal{D}_h^{\mathrm{E}} \cup \mathcal{D}_h^{\mathrm{S,clean}} \right).$$

Notice that both $\mathcal{D}^{\mathrm{E}}$ and $\mathcal{D}^{\mathrm{S,clean}}$ are collected by the expert policy. Then if $|\mathcal{D}^{\mathrm{E}}| + |\mathcal{D}^{\mathrm{S,clean}}| \ge 1$, we can calculate that for each $(s, h) \in \mathcal{S} \times [H]$,

$$d_h^{\pi^{\mathrm{E}}}(s) \mathbb{P}\left( (s, \pi_h^{\mathrm{E}}(s)) \notin \mathcal{D}_h^{\mathrm{E}} \cup \mathcal{D}_h^{\mathrm{S,clean}} \right) = d_h^{\pi^{\mathrm{E}}}(s) \left( 1 - d_h^{\pi^{\mathrm{E}}}(s) \right)^{|\mathcal{D}^{\mathrm{E}}| + |\mathcal{D}^{\mathrm{S,clean}}|}$$

$$\le \frac{4}{9(|\mathcal{D}^{\mathrm{E}}| + |\mathcal{D}^{\mathrm{S,clean}}|)},$$

where the last inequality follows Lemma 5. If $|\mathcal{D}^{\mathrm{E}}| + |\mathcal{D}^{\mathrm{S,clean}}| = 0$, we directly have that

$$d_h^{\pi^{\mathrm{E}}}(s) \mathbb{P}\left( (s, \pi_h^{\mathrm{E}}(s)) \notin \mathcal{D}_h^{\mathrm{E}} \cup \mathcal{D}_h^{\mathrm{S,clean}} \right) \le 1 = \frac{1}{\max\{|\mathcal{D}^{\mathrm{E}}| + |\mathcal{D}^{\mathrm{S,clean}}|, 1\}}.$$

We unify the above two inequalities and get that

$$d_h^{\pi^{\mathrm{E}}}(s) \mathbb{P}\left( (s, \pi_h^{\mathrm{E}}(s)) \notin \mathcal{D}_h^{\mathrm{E}} \cup \mathcal{D}_h^{\mathrm{S,clean}} \right) \le \frac{1}{\max\{|\mathcal{D}^{\mathrm{E}}| + |\mathcal{D}^{\mathrm{S,clean}}|, 1\}}.$$

Now we proceed to upper bound the imitation gap.

$$\mathbb{E}_{\mathcal{D}^{\mathrm{E}}, \mathcal{D}^{\mathrm{S,clean}}} \left[ V(\pi^{\mathrm{E}}) - V(\pi^{\mathrm{ISW\text{-}BC}}) \right] \le H \sum_{h=1}^{H} \sum_{s \in \mathcal{S}} d_h^{\pi^{\mathrm{E}}}(s) \mathbb{P}\left( (s, \pi_h^{\mathrm{E}}(s)) \notin \mathcal{D}_h^{\mathrm{E}} \cup \mathcal{D}_h^{\mathrm{S,clean}} \right)$$

$$\le \frac{|\mathcal{S}|H^2}{\max\{|\mathcal{D}^{\mathrm{E}}| + |\mathcal{D}^{\mathrm{S,clean}}|, 1\}}.$$

Note that $|\mathcal{D}^{\mathrm{E}}| + |\mathcal{D}^{\mathrm{S,clean}}| \sim \mathrm{Bin}(N_{\mathrm{tot}}, \eta + (1 - \eta)\eta')$. Taking expectation with respect to $|\mathcal{D}^{\mathrm{E}}| + |\mathcal{D}^{\mathrm{S,clean}}|$ yields that

$$\mathbb{E}\left[ V(\pi^{\mathrm{E}}) - V(\pi^{\mathrm{ISW\text{-}BC}}) \right] \le \mathbb{E}\left[ \frac{|\mathcal{S}|H^2}{\max\{|\mathcal{D}^{\mathrm{E}}| + |\mathcal{D}^{\mathrm{S,clean}}|, 1\}} \right]$$

$$\le \mathbb{E}\left[ \frac{2|\mathcal{S}|H^2}{|\mathcal{D}^{\mathrm{E}}| + |\mathcal{D}^{\mathrm{S,clean}}| + 1} \right]$$

$$\overset{(a)}{\le} \frac{2|\mathcal{S}|H^2}{N_{\mathrm{tot}}(\eta + (1 - \eta)\eta')}$$

$$= \frac{2|\mathcal{S}|H^2}{N_{\mathrm{E}} + \eta' N_{\mathrm{S}}}.$$

Inequality $(a)$ follows Lemma 3. We finish the proof.

## G    Technical Lemmas

**Lemma 3.** *For any $N \in \mathbb{N}_+$ and $p \in (0, 1)$, if the random variable $X$ follows the binomial distribution, i.e., $X \sim \text{Bin}(N, p)$, then we have that*

$$\mathbb{E}\left[\frac{1}{X+1}\right] \leq \frac{1}{Np}.$$

*Proof.*

$$
\begin{aligned}
\mathbb{E}\left[\frac{1}{X+1}\right] &= \sum_{x=0}^{N} \left(\frac{1}{x+1}\right) \frac{N!}{x!(N-x)!} p^x (1-p)^{N-x} \\
&= \frac{1}{(N+1)p} \sum_{x=1}^{N+1} \left(\frac{(N+1)!}{x!(N+1-x)!}\right) p^x (1-p)^{N+1-x} \\
&= \frac{1}{(N+1)p} \left(1 - (1-p)^{N+1}\right) \leq \frac{1}{Np}.
\end{aligned}
$$

$\square$

**Lemma 4** (Binomial concentration (Lemma A.1 in [57])). *For any $N \in \mathbb{N}_+$ and $p \in (0, 1)$, suppose $X \sim \text{Bin}(N, p)$. Then with probability at least $1 - \delta$, we have*

$$\frac{p}{\max\{X, 1\}} \leq \frac{8 \log(1/\delta)}{N}.$$

**Lemma 5.** *For any $N \in \mathbb{N}_+$ and $x \in [0, 1]$, consider the function $f(x) := x(1-x)^N$, then we have*

$$\forall x \in [0, 1], \ f(x) \leq \frac{4}{9N}.$$

*Proof.* We calculate that $f'(x) = (1-x)^{N-1}(1 - (N+1)x)$. It is direct to have that $f(x)$ achieves its maximum at $x^\star = 1/(N+1)$. Furthermore, we have

$$f\left(\frac{1}{N+1}\right) = \frac{1}{N}\left(1 - \frac{1}{N+1}\right)^{N+1} \overset{(a)}{\leq} \frac{1}{eN} \leq \frac{4}{9N}.$$

Inequality $(a)$ follows that $(1 + x/N)^N \leq \exp(x), \ \forall N \geq 1, |x| \leq N$. We complete the proof. $\square$

## H    Experiments Details and Additional Results

### H.1    Experiment Details

In this section, we present the experiment details to facilitate the replication of our results. The experiments are conducted on a machine comprising 48 CPU cores and 4 V100 GPU cores. We repeat each experiment 5 times using different random seeds (2021, 2022, 2023, 2024, and 2025).

#### H.1.1    Robotic Locomotion Control

In this study, we evaluate the performance of various imitation learning algorithms on four locomotion control tasks from the MuJoCo suite: `Ant-v2`, `HalfCheetah-v2`, `Hopper-v2`, and `Walker2d-v2`. These tasks are widely used in the literature and are considered challenging benchmarks.

To train the expert policy, we use the online Soft Actor-Critic (SAC) algorithm [19] with 1 million training steps. We implement the algorithm using the rlkit codebase, which is available at `https://github.com/rail-berkeley/rlkit`. The training curves of the online SAC agent are shown in Figure 5. We treat the resulting policy as the expert policy and use it to generate expert trajectories.

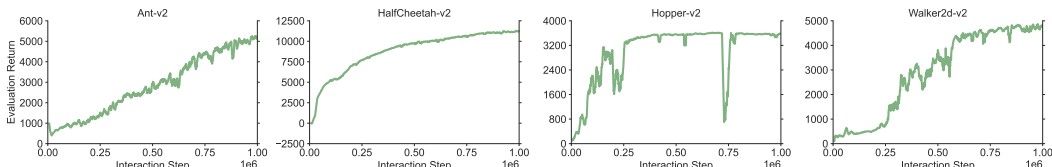

Figure 5: Training curves of online SAC on 4 locomotion control environments.

In our experimental setup, we utilize an expert dataset comprising of 1 expert trajectory collected by the trained SAC agent. Additionally, all algorithms are provided with a supplementary dataset. There are two setting of the supplementary data.

- `Full Replay`. The supplementary dataset is obtained from the replay buffer of the online SAC agent, which has over one million samples, equivalent to 1000+ trajectories. The rapid convergence of online SAC, as illustrated in Figure 5, implies that the replay buffer is enriched with a substantial number of expert-level trajectories. As a result, we expect that utilizing the supplementary data without any modification may lead to desirable results.

- `Noisy Expert`. The supplementary dataset comprises of 10 clean expert trajectories and 5 noisy expert trajectories. In this case, we replace the action labels in the noisy trajectories with random actions drawn from $[-1, 1]$. This replacement creates noisy action labels for the expert states, leading to a significant distribution shift at the state-action level, as noted in Remark 1. The high degree of distribution shift makes it challenging for using the supplementary data.

We use a 2-hidden-layer multi-layer perceptron (MLP) with hidden size 256 and ReLU activation for all algorithms, as the state information in locomotion control tasks is informative by design. The codebase of DemoDICE is based on the original authors' work, which can be accessed at https://github.com/KAIST-AILab/imitation-dice. For DWBC, we also use the authors' codebase, which is available at https://github.com/ryanxhr/DWBC. We experimented with different hyper-parameters for both algorithms but found that the default parameters provided by the authors work well. We normalize state observations in the dataset before training all algorithms, following [25]. This is crucial for achieving satisfactory performance.

In training the discriminator of ISW-BC, we use the gradient penalty (GP) regularization, as recommended by [25]. We add the following loss to the original loss (4) to enforce 1-Lipschitz continuity:

$$\min_{\theta} \sum_{(s,a) \in \mathcal{B}} \left( \|g(s, a; \theta)\| - 1 \right)^2,$$

where $g$ is the gradient of the discriminator $c(s, a; \theta)$, and $\mathcal{B}$ is a mini-batch. This promotes the learning of smooth features and can improve generalization performance.

In our implementation of ISW-BC, we employ 2-hidden-layer MLPs with 256 hidden units and ReLU activation for both the discriminator and policy networks. We use a batch size of 256 and Adam optimizer with a learning rate of 0.0003 for training both networks. The training objective is to maximize the log-likelihood. We set $\delta$ to 0 and use a gradient penalty coefficient of 8 by default, unless otherwise stated. The training process is carried out for 1 million iterations. We evaluate the performance every 10k iterations with 10 episodes. The normalized score in the last column of Table 2 is computed in the following way:

$$\text{Normalized score} = \frac{\text{Expert performance} - \text{Agent performance}}{\text{Expert performance} - \text{Random policy performance}}. \tag{20}$$

### H.1.2 Atari Video Games

We evaluate algorithms on a set of 5 Atari video games from the standard benchmark: `Alien`, `MsPacman`, `Phoenix`, `Qbert`, and `SpaceInvaders`. We preprocess the game environments using a standard set of procedures, including sticky actions with a probability of 0.25, grayscaling, downsampling to an image size of [84, 84], and stacking frames of 4. These procedures follow the instructions provided by the dopamine codebase, which is available at https://github.com/google/dopamine/blob/master/dopamine/discrete_domains/atari_lib.py. The final image inputs are of shape (84, 84, 4).

We use the replay buffer data from an online DQN agent, which is publicly available at `https://console.cloud.google.com/storage/browser/atari-replay-datasets`, thanks to the work of [2]. The dataset consists of 200 million frames, divided into 50 indexed buckets (ranging from 0 to 49). However, using the entire dataset is computationally infeasible[9] and unnecessary for our task. Therefore, we select specific buffer buckets for imitation learning.

We choose the expert data from bucket index 49, using only the first 400K frames for training. This makes the task challenging (we find that BC performs well with 1M frames of expert data). For the full replay setting, we select supplementary data from buffer indices 45 to 48, using the first 400K frames from each bucket. This yields a supplementary dataset that is 4 times larger than the expert data. In the noisy task setting, we follow the same procedure for selecting supplementary data, but replace the action labels with random labels on buffer index 45.

All agents employ the same convolutional neural network (CNN) architecture as the DQN agent, consisting of three convolutional blocks. The first block applies a filter size of 8, a stride of 4, and has a channel size of 32. The second block uses a filter size of 4, a stride of 4, and a channel size of 64, while the third block applies a filter size of 3, a stride of 4, and has a channel size of 64. All blocks use the ReLU activation function. The feature representations are flattened to a vector, on which a 1-hidden-layer MLP with a hidden size of 512 and ReLU activation function is applied. Finally, the outputs are passed through a softmax function to obtain a probability distribution.

Atari games are not considered in [25, 61] and public implementations of DemoDICE and DWBC for Atari games are not available. To use these methods in the Atari environment, we extend their original implementation by replacing the MLP used in locomotion control with the CNN described earlier. Implementing ISW-BC is a little more complicated. We use the same CNN policy network as in the other methods, but find that directly training the discriminator from scratch is less effective. This is because the discriminator tends to focus on irrelevant background information instead of the decision-centric part. To overcome this issue, we build the discriminator upon the feature extractor of the policy network, leveraging its ability to extract useful information. The discriminator is an MLP with ReLU activation and a hidden size of 1024: the image feature representation has a dimension 512 and the action feature representation also has a dimension 512 (we randomly project one-hot discrete actions to a 512-dimension space). We find that the depth of the MLP is crucial for performance, using a depth of 1 for the full replay setting and 3 for the noisy expert setting. We clip the importance sampling ratio for numerical stability, using a minimum value of 0 and a maximum value of 5 for the full replay setting, and a minimum value of 0.2 and a maximum value of 5 for the noisy expert setting. We provide ablation studies of these hyperparameters in Appendix H.2.2.

All methods were optimized using the Adam optimizer with a learning rate of 0.00025 and a batch size of 256. The training objective is to maximize the log-likelihood. The training process consisted of 200K gradient steps. Every 2K gradient steps, the algorithms were evaluated by running 10 episodes and computing the raw game scores. The normalized score in the last column of Table 3 is computed by Eq. (20).

### H.1.3 Image Classification

We utilize the publicly available DomainNet dataset [36] for our experiments, which can be accessed at `http://csr.bu.edu/ftp/visda/2019/multi-source`. This dataset comprises six sub-datasets: `clipart`, `infograph`, `painting`, `quickdraw`, `real`, and `sketch`, with 2103, 2626, 2472, 4000, 4864, and 2213 images, respectively. Our task involves recognizing objects from 10 different classes: `bird`, `feather`, `headphones`, `ice_cream`, `teapot`, `tiger`, `whale`, `windmill`, `wine_glass`, and `zebra`. We divided the images into training and test sets, with 80% for training and 20% for testing.

We employ a 2-hidden-layer neural network with a hidden size of 512 and ReLU activation as the classifier. To extract features from images, we utilize the pretrained ResNet-18 model (trained on ImageNet), which has a feature dimension of 512. The ResNet-18 model can be accessed at `https://pytorch.org/vision/main/models/generated/torchvision.models.resnet18.html`. We opted for this approach as training such a large convolutional neural network directly on the DomainNet dataset proved to be ineffective. The training objective is to minimize the cross-entropy loss. To optimize the network parameters, we use the stochastic gradient descent (SGD) optimizer with a learning rate of 0.01 and momentum of 0.9. Additionally, we apply

---

[9]Loading 200M frames requires over 500GB memory.

weight decay with a coefficient of 0.0005. The models are trained for 100 epochs with a batch size of 100, following the standard practice.

The discriminators used in ISW-BC and DWBC are implemented as 2-hidden-layer neural networks with ReLU activation. It's important to note that these discriminators take both the image and label as inputs. The image input is processed by the pre-trained and fixed ResNet-18, while the label input is projected to the same dimension (512) by a random projection matrix. The hidden size for the discriminator is set to 1024 for ISW-BC and 1025 for DWBC, as the discriminator in DWBC also takes the log-likelihood as an input. For ISW-BC, the discriminator is trained independently for 100 epochs with the same optimization configuration as the classifier. Afterward, the discriminator is fixed, and its output is used to compute the importance sampling ratio, which is then used to train the classifier.

## H.2 Additional Results

### H.2.1 Training Curves

The training curves on the robotic locomotion control tasks are displayed in Figure 6, Figure 7 and Figure 8. The training curves on Atari video games are displayed in Figure 9 and Figure 10. The training curves on the image classification task are displayed in Figure 11.

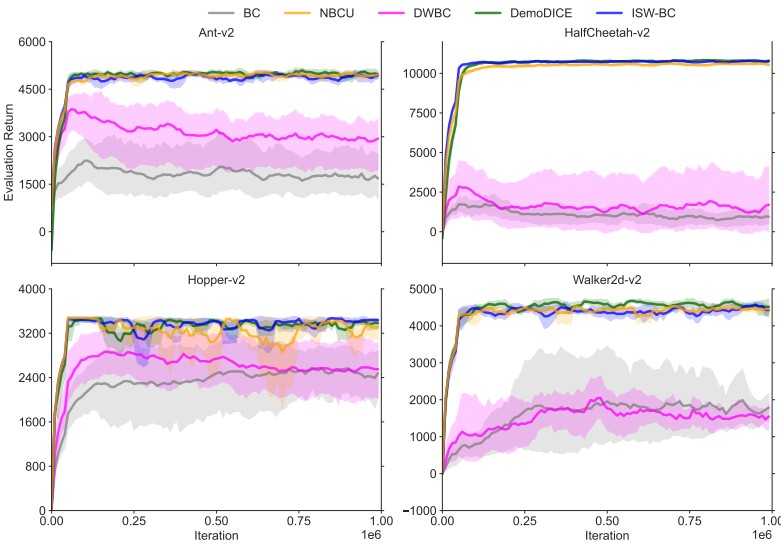

Figure 6: Training curves of algorithms on the locomotion control task in the full replay setting. Solid lines correspond to the mean performance and shaded regions correspond to the 95% confidence interval. Same as other figures.

### H.2.2 Ablation Study

In this section, we present ablation studies conducted on Atari games, aiming to provide insights into the underlying working scheme of our method. We specifically emphasize Atari games due to their high-dimensional image inputs, making these tasks particularly challenging. In contrast, the other two tasks, locomotion control and image classification, involve informative vector inputs, setting them apart from the unique characteristics of Atari games.

**Ablation Study on Feature Representations of Discriminator Network.** Our study reveals that employing a separate CNN for the discriminator yields inferior results compared to utilizing the feature extractor of the policy network. Please refer to Figure 12. Our conjecture is that training the discriminator independently may cause it to fit noise information (e.g., background). In contrast, the policy CNN network is capable of learning decision-centric information, enabling an effective approach to building the discriminator network through the feature extractor of the policy network.

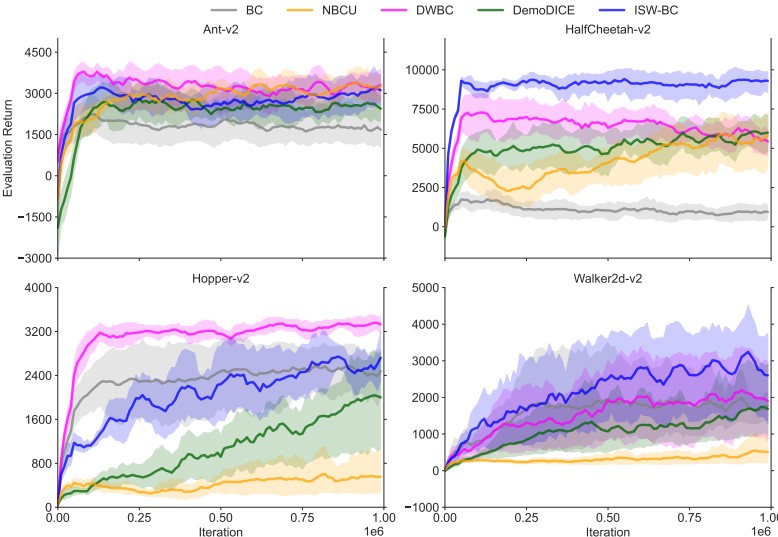

Figure 7: Training curves of algorithms on the locomotion control task in the noisy expert setting.

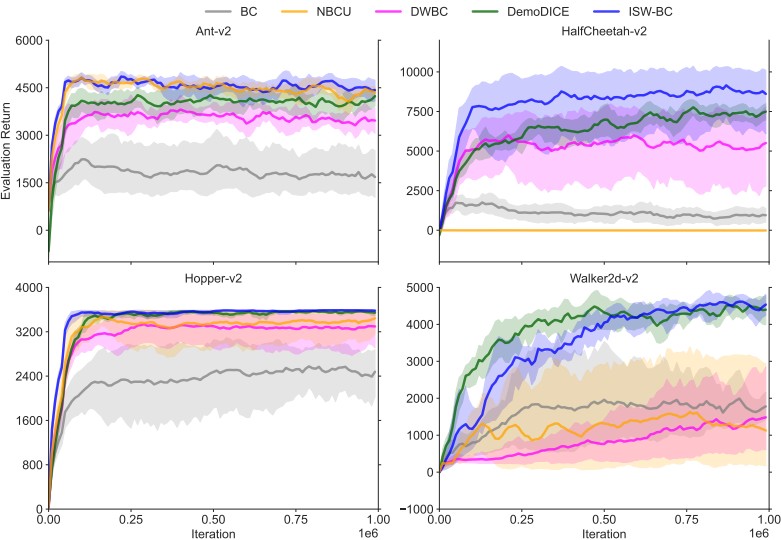

Figure 8: Training curves of algorithms on the locomotion control task in the expert&random setting.

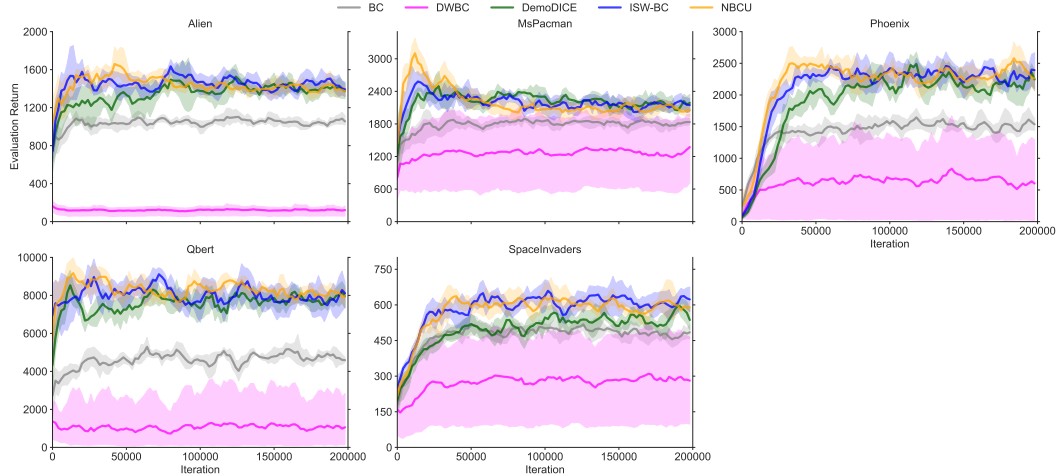

Figure 9: Training curves of algorithms on the Atari games in the full replay setting.

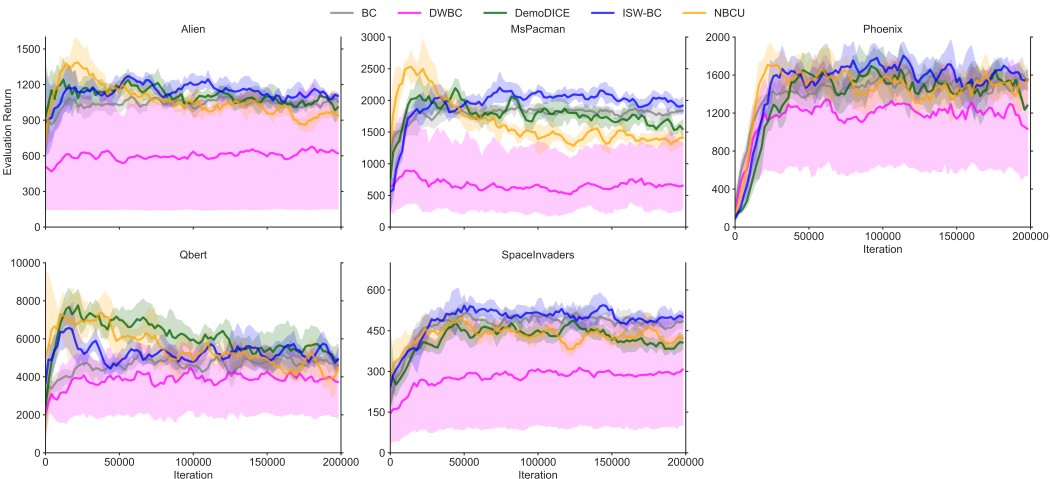

Figure 10: Training curves of algorithms on the Atari games in the noisy expert setting.

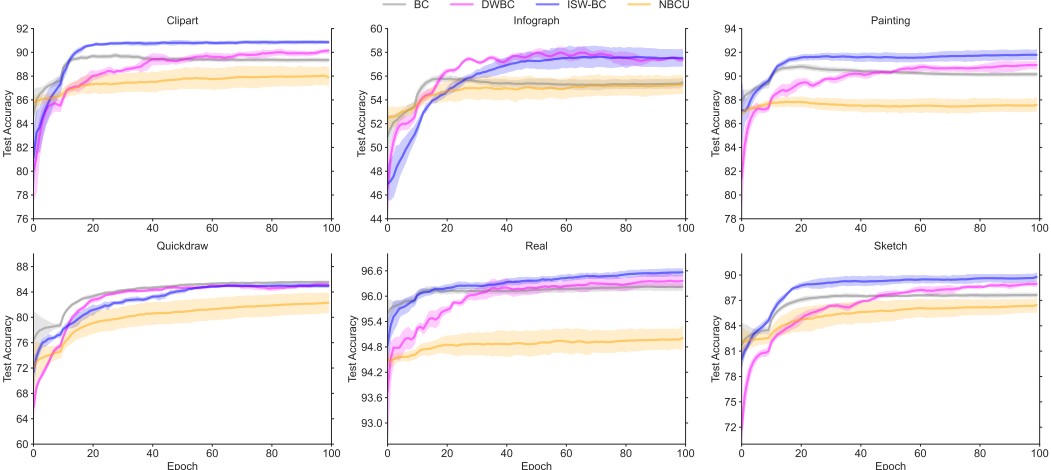

Figure 11: Training curves of algorithms on the image classification task using the DomainNet dataset.

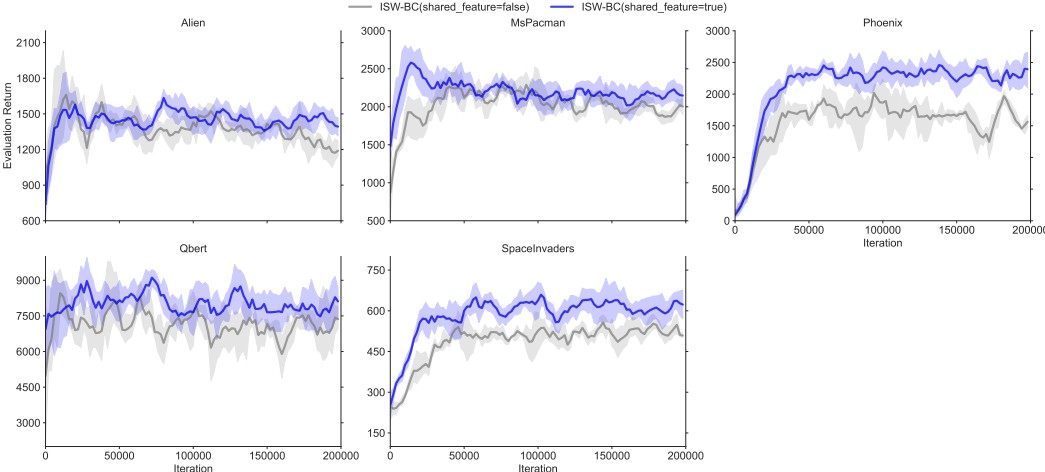

Figure 12: Training curves of ISW-BC on the Atari games in the full replay setting. We test the performance with different feature extractors of the discriminator.

**Ablation Study on Depth of Discriminator Network.** We have discovered that the number of discriminator layers plays a crucial role in the performance of Atari games. The training curves, depicted in both Figure 13 and Figure 14, illustrate the performance variation based on the number of layers in the discriminator network. Notably, a 1-hidden-layer neural network yields the best results for the full replay setting, while a 3-hidden-layer neural network performs optimally in the noisy expert setting. It is important to note that this phenomenon is specific to Atari games. We do not have a good explanation yet. We believe this deserves further investigation in the future work.

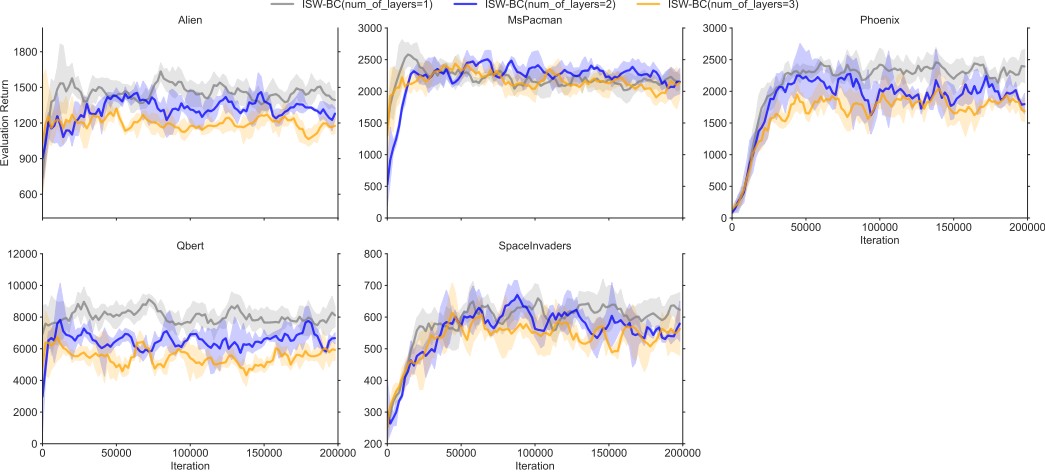

Figure 13: Training curves of ISW-BC on the Atari games in the full replay setting. We test the performance with different number of layers for the discriminator network.

### H.2.3  Results on the Mix of Expert and Random Data

In the main text, we consider two types of supplementary datasets: full replay and noisy expert. Here we consider another type of supplementary datasets which consists of expert trajectories and trajectories generated by a random policy [25, 47]. We evaluate all methods on this new type of supplementary datasets on the robotic locomotion control tasks. In particular, the supplementary dataset contains 10 expert trajectories and 10 random trajectories. We report the experimental results on Table 5. The results show that ISW-BC outperforms NBCU significantly, demonstrating

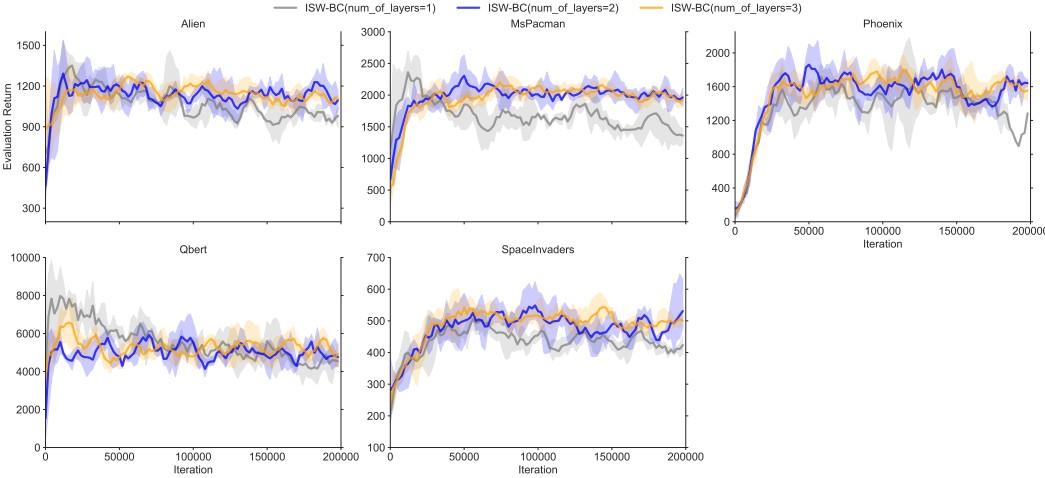

Figure 14: Training curves of ISW-BC on the Atari games with the noisy expert setting. We test the performance with different number of layers for the discriminator network.

the robustness of ISW-BC to distribution shift. Furthermore, ISW-BC also performs better than DemoDICE and DWBC.

Table 5: Environment return of algorithms on the mixture of expert and random dataset on 4 locomotion control tasks

|  | Ant | HalfCheetah | Hopper | Walker | Avg |
|---|---|---|---|---|---|
| Random | $-326$ | $-280$ | $-20$ | $2$ | $0\%$ |
| Expert | $5229$ | $11115$ | $3589$ | $5082$ | $100\%$ |
| BC | $1759_{\pm287}$ | $931_{\pm273}$ | $2468_{\pm164}$ | $1738_{\pm311}$ | $38\%$ |
| NBCU | $4316_{\pm215}$ | $-14_{\pm0}$ | $3401_{\pm68}$ | $1189_{\pm100}$ | $51\%$ |
| DemoDICE | $4097_{\pm237}$ | $7401_{\pm356}$ | $3556_{\pm27}$ | $4451_{\pm141}$ | $83\%$ |
| DWBC | $3447_{\pm226}$ | $5307_{\pm319}$ | $3286_{\pm32}$ | $1451_{\pm118}$ | $59\%$ |
| ISW-BC | $\mathbf{4467}_{\pm134}$ | $\mathbf{8697}_{\pm365}$ | $\mathbf{3584}_{\pm6}$ | $\mathbf{4528}_{\pm226}$ | $\mathbf{88}\%$ |

