# OpenReview forum: "Imitation Learning from Imperfection: Theoretical Justifications and Algorithms"
_NeurIPS.cc/2023/Conference — NeurIPS 2023 spotlight_

### Official Review · Reviewer_FJAh · 2023-06-27

**Soundness:** 3 good
**Presentation:** 3 good
**Contribution:** 3 good
**Rating:** 6
**Confidence:** 3

**Summary:**

This paper considers the problem of offline imitation learning with supplementary data with optimality not guaranteed. The paper gives theoretical analysis on the performance gap bound between expert policy and learner's policy for behavior cloning (BC) on expert data only and naively using BC over the union of expert and supplementary data. Based on the analysis, the paper proposes a provably better method than BC, which is ISW-BC. ISW-BC uses importance sampling (with a lower threshold) between state-action pairs to "correct" the learning from non-expert state-action occupancy onto expert state-action occupancy. In several mujoco and Atari environments, ISW-BC works comparably well or better than the SOTA methods, DemoDICE and DWBC.

**Strengths:**

**1. Good writing that is easy to follow and clearly conveys the idea.** The paper is a math-heavy one with many pages of theoretical proofs; however, the core result is well-summarized by Tab. 1, the theorems, and Eq. 3-5. Besides, for readers that are interested in theoretical results with function approximators, the theorems are kindly summarized in Sec. E in the appendix, leaving the long proof details in the next section. The limitations, broader impact and computational resource are all well-discussed.

**2. Simple but effective idea.** The idea of ISW-BC proposed by the paper is very simple; it only requires separate training of a discriminator and an actor, which is quite easy to implement. However, such method is theoretically guaranteed to be better than BC and is indeed better than many baselines in multiple environments.

**3. Solid theoretical and practical results.** The theoretical results clearly shows that how bad could it be when we are treating non-expert data as expert data in behavior cloning (BC), and how the quality (value function) of the non-expert data affects the result. Based on this, the author proposes ISW-BC, a method that is theoretically proved to be better, and indeed achieves superior performance over multiple baselines (DWBC, DemoDICE) on multiple environments (Atari, mujoco, and even non-RL tasks).









**Weaknesses:**

**1. The proposed method, ISW-BC, might be non-robust to stochastic environment.** Consider a simple tabular MDP with five states $s_{begin}, s_1, s_2, s_{success}, s_{fail}$;

the agent always begin at $s_{begin}$, and there is only one action for $s_{begin}$, which has 50% probability of leading to $s_1$ and 50% probability of leading to $s_2$;

there is only one action for $s_1$ that 100% leads to $s_{success}$, and two actions for $s_2$ that 100% leads to $s_{success}$ and $s_{fail}$ respectively;

 $s_{success}$ is the success state that terminates the episode with $+1$ reward; $s_{fail}$ terminates the episode with $-1$ reward.

Now, consider the scenario where we only have one expert trajectory (1-shot is a common case) that goes from $s_{begin}$ to $s_1$ and finally $s_{success}$. The supplementary data acts uniformly random. By definition, the discriminator now gives a close-to-zero ratio for $d_h^E/d_h^U$ with any history that involves $s_2$ (let us ignore the numerical stability issue for now because there are engineering solutions). DICE methods have an equivalence of value function (which is the Lagrange dual function), which can guide the agent back from $s_2$ to $s_{success}$. ISW-BC, however, append no or very little weight to supplementary data on $s_2$, and, because expert has never experienced $s_2$, does not know what to do on $s_2$.

**2. It strikes me as a little strange how we "justify" the use of weight in imitation with imperfect data from theoretical analysis**, because the story of the paper seems to be improve over NBCU (later proved to be empirically better than DICE/DWBC), but the NBCU analyzed in the paper is too unintuitive to ever work; it is natural, rather than with theoretical analysis for one to know that (s)he cannot treat non-expert data that is arbitrarily bad as expert ones. **(Despite of this, I am still convinced that the theoretical analysis on NBCU is a notable contribution.)**

**3. Other minor problems:**

a) Besides the empirical advance on offline IL mentioned in the paper, there are more theoretical advances in offline IL (more specifically, the unification of offline IL and RL), which are MAHALO [1] and ReCOIL [2]. I encourage the author to briefly discuss them in the related work section.

b) The color of the curve for each method should be unified throughout the paper. For example, the color of ISW-BC in Fig.9 and Fig.10 in the appendix are not unified and might mislead the readers.



**References:**

[1] A. Li et al. MAHALO: Unifying Offline Reinforcement Learning and Imitation Learning from Observations. In ICML, 2023.

[2] H. S. Sikchi et al. Imitation from Arbitrary Experience: A Dual Unification of Reinforcement and Imitation Learning Methods. In ArXiv, 2023.

**Questions:**

I have one question: can we actually comprehend ISW-BC as reward-weighted regression [1, 2] with $\gamma=0$ and reward $r(s,a)=\log\frac{d^E(s,a)}{d^U(s,a)}$? If so, then I think more discussion is needed on the relationship between ISW-BC and RWR [1] / AWR [2].

The suggestions for the authors are listed as follows. I am open to increase my score if the authors can improve their paper as suggested:

1. Modify the paper as suggested in the weaknesses (discussion of point 1, modification with point 2, 3) and limitation section;

2. It would be great for the authors to try the supplementary data consists of some expert trajectories and many random trajectories or medium-level trajectories, like those tested in [3]. The component of the supplementary data can largely affect the performance of the algorithm [3];

3. I strongly recommend the authors to open-source their code upon acceptance (or next submission).

**References:**

[1] Jan Peters and Stefan Schaal. Reinforcement learning by reward-weighted regression for operational space control. In ICML, 2007.

[2] Advantage-Weighted Regression: Simple and Scalable Off-Policy Reinforcement Learning. X. Peng et al. in ArXiv, 2019.

[3] H. S. Sikchi et al. Imitation from Arbitrary Experience: A Dual Unification of Reinforcement and Imitation Learning Methods. In ArXiv, 2023.

**Limitations:**

**Limitations:** The authors have discussed the theoretical limitations in line 280 and line 746-750. I think, however, there are more limitations that the authors could consider to add, and concentrated in a single limitation section:

1) non-robustness to stochasticity (elaborated in point 1 of the weakness section);

2) the assumption that $d^U$ covers $d^E$, which is also a weakness that DICE possesses but still a concern in practical use.

Note these limitations do not necessarily mean that the work is not valuable enough for the conference; however, they do pose concern for readers who want to apply ISW-BC in the future.

**Potential Negative Societal Impact:** The paper does a good job in discussing the broader impact at the beginning of the supplementary material.

---

> ### Author Rebuttal · Authors · 2023-08-09
>
> Thank you for taking the time to review our paper, and for your constructive comments.
>
> **Comment 1:** The proposed method, ISW-BC, might be non-robust to stochastic environment.
>
> **Response 1:** Thank you for providing the example and initiating the discussion. However, we would like to clarify that in theory, ISW-BC can work in stochastic environments (we do not pose a deterministic transition assumption). Moreover, in practice, we have considered non-deterministic tasks and showed that ISW-BC works well.
>
> Regarding your concern with the provided example, we clarify that this failure mode is associated with Proposition 2, where the main issue lies in the tabular representations rather than the stochasticity itself. However, we have shown that when feature design is well-crafted, ISW-BC can effectively avoid this failure mode, as demonstrated in our Theorem 3.
>
> Regarding your mention of DICE methods, we would appreciate further clarification on which specific DICE method you are referring to and how it manages to work with just one expert trajectory. If you are referring to the DemoDICE method, we would like to emphasize that it cannot recover the expert policy even in the population level (see Appendix B for more details).
>
> **Comment 2:** It strikes me as a little strange how we "justify" the use of weight in imitation with imperfect data from theoretical analysis.
>
> **Response 2:** We appreciate your feedback, and we would like to provide further clarification to address your concern. Our theoretical analysis indeed sheds new light on the justification for using weights in imitation with imperfect data. While it is intuitive to believe that treating non-expert data, which can be arbitrarily bad, as expert data would not work in the worst-case scenario, our theory reveals that this naive idea may still perform well in certain cases. Our theory introduces the concept of state-action/policy distribution shifts, as discussed in Remark 1, and provides a characterization of when the naive approach fails and when it can be effective. We use experiments to illustrate both the good and bad cases for NBCU. Then our paper attempts to introduce the use of importance sampling to improve NBCU in both cases.
>
> **Comment 3:** Besides the empirical advance on offline IL mentioned in the paper, there are more theoretical advances in offline IL, which are MAHALO [1] and ReCOIL [2].
>
> **Response 3:** Thanks for pointing out these references. We provide a short discussion below, which will be involved in the revised paper. Similar to MILO (citied as [7] in our paper), MAHALO [1] analyzes the suboptimality of MAHALO which follows the pessimism principle in offline RL, while we study BC and its variants. ReCOIL[2] optimizes the state-action distribution matching objective with new duality techniques and presents a new energy-based model viewpoint. In contrast, ISW-BC leverages the importance sampling technique, and we provide a new analysis for it.
>
> **Comment 4:** The color of the curve for each method should be unified throughout the paper.
>
> **Response 4:** We will ensure that the color format is unified to avoid any confusion in the revised version.
>
> **Comment 5:** I have one question: can we actually comprehend ISW-BC as reward-weighted regression [1, 2] with $\gamma=0$ and reward $r(s, a) = \log \frac{d^{\operatorname{E}}(s, a)}{ d^{\operatorname{U}}(s, a)}$.
>
> **Response 5**: Yes, the training objective of ISW-BC can be comprehended as reward-weighted regression (RWR) with $\gamma=0$ and reward $r(s, a) = \log \frac{d^{\operatorname{E}}(s, a)}{ d^{\operatorname{U}}(s, a)}$. This viewpoint is intriguing, but we would like to highlight the following differences:
>
> - Papers on RWR mainly consider the online setting, while we focus on the offline setting.
> - Papers on RWR are applicable in the RL setting where the reward is readily available, whereas in our imitation learning setting, we need to infer the reward (or the importance sampling ratio).
>
> **Comment 6:** Modify the paper as suggested in the weaknesses and limitation section.
>
> **Response 6:** Your suggestions are very helpful and we will revise the paper as suggested.
>
> **Comment 7:** It would be great for the authors to try the supplementary data consists of some expert trajectories and many random trajectories or medium-level trajectories, like those tested in [3].
>
> **Response 7:**  Thanks for pointing out that supplementary data distribution is important for algorithm performance, which is consistent with our claim in discussing NBCU. We have experimented that supplementary data consists of 10 expert trajectories and 10 random trajectories in the locomotion control benchmark. The results show that ISW-BC outperforms BC by a wide margin, demonstrating the robustness of ISW-BC to distribution shift. Furthermore, ISW-BC also performs better than other baselines DemoDICE and DWBC. The detailed results are available in Table 1 in a separated pdf file. We are also planning to conduct more experiments with medium-level trajectories or in Atari games and replenish these results in the future revision.
>
> **Comment 8:** I strongly recommend the authors to open-source their code upon acceptance (or next submission).
>
> **Response 8:** As promised in the Appendix, we will make our code and datasets available for public access upon acceptance. Currently, in accordance with the NeurIPS instructions, we have provided the code to the area chair via an anonymized link.
>
> **Comment 9:** The assumption that $d^{\operatorname{U}}$ covers $d^{\operatorname{E}}$, which is also a weakness that DICE posses but still a concern in practical use.
>
> **Response 9:** We appreciate your feedback, but we would like to clarify that this assumption directly holds because the union data includes the expert data.
>
> ---
>
> We hope that the above response can address your concerns adequately. We would greatly appreciate it if you could re-evaluate our paper based on the above responses.

---

> > ### Comment · Reviewer_FJAh · 2023-08-11
> > **Response to Rebuttal**
> >
> > Thanks for your detailed response. Generally, I think most of my question raised are well-addressed. However, I have two follow-up comments to the response:
> >
> > 1. Regarding response 1: while I agree that DICE methods such as DemoDICE cannot retrieve the policy with minimal occupancy divergence to the expert policy, my argument is that at the policy retrieval step, DemoDICE uses the exponential of (scaled) advantage, which will give more weight toward the action leading to $s_{success}$ as long as the value function (dual variable) learned for $s_{success}$ is higher than that for $s_{fail}$. Thus, though DemoDICE does not accurately retreive expert policy, with higher probability it will make the right choice on $s_2$.
> >
> > 2. Regarding response 5: There are works that are similar to RWR/AWR but considers offline scenario, e.g. MARWIL [1]. The use of advantage / return-based weighted regression is common in the RL/IL community [2, 3].
> >
> > **References:**
> >
> > [1] Q. Wang et al. Exponentially Weighted Imitation Learning for Batched Historical Data. In NeurIPS, 2018.
> >
> > [2] Abdolmaleki, A., Springenberg, J. T., Tassa, Y., Munos, R., Heess, N., & Riedmiller, M. (2018). Maximum a posteriori policy optimisation. arXiv preprint arXiv:1806.06920.
> >
> > [3] Wang, Z., Novikov, A., Zolna, K., Merel, J. S., Springenberg, J. T., Reed, S. E., ... & de Freitas, N. (2020). Critic regularized regression. Advances in Neural Information Processing Systems, 33, 7768-7778.

---

> > > ### Author Response · Authors · 2023-08-12
> > > **Thanks for Your Helpful Comments**
> > >
> > > We appreciate your prompt response!
> > >
> > > **Response 1:** Your insightful discussion is greatly appreciated. We agree with your viewpoint that the DICE method could indeed make a correct decision for $s_2$, if we ignore the bias concern within DemoDICE. We intend to incorporate this valuable discourse into our revised paper.
> > >
> > > **Response 2:** We extend our gratitude for highlighting these references to us. It is important to note that the methods outlined in the cited works necessitate access to accurate environment rewards, thus rendering them unsuitable for direct application within the context of our paper. Nonetheless, we acknowledge the significance of these works and intend to include a comprehensive discussion of them in our revised paper.

---

> > > > ### Comment · Reviewer_FJAh · 2023-08-14
> > > > **Further Response to the Authors**
> > > >
> > > > Thanks a lot for your response! I have no other follow-up comments and have changed my score from 5 to 6.

---

> > > > > ### Author Response · Authors · 2023-08-19
> > > > >
> > > > > We sincerely appreciate your constructive feedback throughout the review process. Your insights have been instrumental, and we are committed to incorporating your suggestions as we revise the paper. We are pleased to hear that our responses addressed your concerns, and we are thankful for your reconsideration of the score.

---

### Official Review · Reviewer_Y1mx · 2023-07-04

**Soundness:** 3 good
**Presentation:** 3 good
**Contribution:** 2 fair
**Rating:** 7
**Confidence:** 3

**Summary:**

The paper provides derivations of the imitation gap, the gap in performance between the trained agent and expert who provided the data, when traditional behavioural cloning (BC) is used. The specific setting assumes there is plentiful of supplementary data to train the BC agent on, but since this supplementary data may (and probably is) of poor quality (i.e., not as good as expert demonstrations), traditional BC produces less than optimal agents. The paper proposes a importance-sampling correction by training a discriminator on the dataset (to distinguish the high quality and low quality demonstrations), and then train BC using the importance-sampling correction. The results indicate the proposed method improves over BC and existing methods, while not reducing original BC performance in settings with high amount of expert data.

**Strengths:**

- Theoretical backing and derivation of the method.
- Both theoretical and empirical improvements over baselines, and proposed method is more applicable than baselines (e.g., no natural extension of DemoDICE to image recognition task, as it lacks rewards).
- Empirical results in three different settings (MuJoCo, Atari and object recognition).
- Method is simple to implement. With a good baseline code base shared, I could see other people adapting this method and trying it out.

**Weaknesses:**

- Proposed method has rather small/noisy improvements in terms of metrics in the experiments.
	- In "noisy expert" setting, the proposed method is clearly better than the baselines. However this setting seems rather unrealistic (proper trajectories from a policy but actions are random). A more realistic scenario would be rollouts from a poorly trained policy, or a random agent.
- Not a very novel setting (as evident by the number of baselines) and the solution, while well executed, is a combination of existing works in somewhat simple way.
- Training discriminators may be problematic (which is a shared difficulty with baselines). Authors note this in the Appendix for the Atari experiments.
- (Minor) No code available, but paper lists references to libraries and datasets used. Nevertheless, replicating the results as presented in the paper will be near-impossible, given the earlier works in ML field. I urge authors to share the code, even if it is "messy", so that others can build on the contributions of this work.

**Questions:**

1) What is the "BC" method shown in Tables 2, 3 and 4? Is this just "BC" row trained on the expert dataset only? If so, please clarify in main paper text.
2) The "one expert trajectory in expert dataset" setup throughout experiments seems overly restrictive. Have you experiment with other settings, like with 5 or 10 expert trajectories in the expert dataset? Was it ensured that the "one expert trajectory" in the expert dataset was indeed a good demonstration? Even with a good policy, some trajectories may end up being bad.
3)  In the object recognition task, the supplementary data is data with valid labels but with a different style. However, the focus is on "sub-optimal policy" data, and in other environments the supplementary data consists of different policies ("full dataset" setting) and outright wrong labels ("noisy expert"), where some actions are replaced with random ones. Following this, it seems like a more natural setup for object recognition task would be to do everything in single domain (e.g., "Real"), but randomly replace the sample label with a wrong label in the supplementary dataset. What is the reason behind using different sub-sets as supplementary data?

### Comments

Note: I do not have the expertise to comment on the mathematical derivations. I work on the assumption these are correct/throughout, and comment on the experimental setup and results.

- Footnotes come after commas
- Term "imitation gap" is used early on (e.g., lines 39 and 66), but not formally or informally defined. A quick definition after first use of the term would help readability.
- Not required, but in future, please use same colors consistently between figures (e.g., same method always gets the same color). This improves paper readability quite considerably.
- Section 3 header sounds like it is missing something (e.g., should it be "Preliminaries" or "Problem setup"?)
Numbers in the tables are in bit different format; some (Random, Expert) seem to be in normal text mode, while rest are in math mode. Formatting should be consistent.

**Limitations:**

Authors have broader impact section, and correctly report the limited societal impact. Authors list some of the limitations of the method in Appendix, e.g., difficulties regarding training discriminator in the Atari domain.

## Rebuttal acknowledgement

I have read authors' rebuttal and new results which did address my concerns, and I updated my score from 6 to 7 (before discussion period closed).

---

> ### Author Rebuttal · Authors · 2023-08-09
>
> Thank you for taking the time to review our paper and providing us with your valuable feedback.
>
> **Comment 1:** In "noisy expert" setting, the proposed method is clearly better than the baselines. However this setting seems rather unrealistic (proper trajectories from a policy but actions are random). A more realistic scenario would be rollouts from a poorly trained policy, or a random agent.
>
> **Response 1:**  We appreciate your concern and would like to provide two solutions to address it. On the one hand, we believe that the considered noisy expert setting is practical in some applications with potential data corruption. For example, although the human expert demonstrates an optimal trajectory, the recorder or the recording system possibly corrupts the data by accident or on purpose. Motivated by such applications, the study on the robustness to data corruption has drawn much research interest in imitation learning [R1, R2].
>
> To further address your concern about the rollouts from non-expert policy, we have experimented the case where the supplementary dataset contains expert trajectories and trajectories collected by a random agent in the locomotion control benchmark. The experimental results show that ISW-BC outperforms NBCU significantly, implying the robustness of ISW-BC to distribution shift. Besides, ISW-BC also performs better than other baselines DemoDICE and DWBC. The detailed results are available in Table 1 in a separated pdf file. We are also planning to conduct more experiments in Atari games and include these results in the future version.
>
> References:
>
> [R1] Liu Liu, et al. “Robust Imitation Learning from Corrupted Demonstrations.” arXiv: 2201.12594
>
> [R2] Fumihiro Sasaki, Ryota Yamashina. “Behavioral Cloning from Noisy Demonstrations." ICLR 2021.
>
> **Comment 2:** No code available.
>
> **Response 2:** As promised in the Appendix, we will make our code and datasets available for public access upon acceptance. Currently, in accordance with the NeurIPS instructions, we have provided the code to the area chair via an anonymized link.
>
> **Question 3:** What is the "BC" method shown in Tables 2, 3 and 4? Is this just "BC" row trained on the expert dataset only?
>
> **Response 3:** Yes, you are correct. The "BC" method shown in Tables 2, 3, and 4 refers to the row labeled "BC," which is trained solely on the expert dataset. We apologize for any confusion, and we will make sure to clarify this point in the later revision of the paper.
>
> **Question 4:** The "one expert trajectory in expert dataset" setup throughout experiments seems overly restrictive. Have you experiment with other settings, like with 5 or 10 expert trajectories in the expert dataset? Was it ensured that the "one expert trajectory" in the expert dataset was indeed a good demonstration? Even with a good policy, some trajectories may end up being bad.
>
> **Response 4:**  We appreciate your concern. Indeed, for the locomotion control tasks, we have conducted experiments with 5 expert trajectories and observed that BC (Behavioral Cloning) trained solely on the expert dataset performs exceptionally well. Consequently, in this case, there appears to be no significant advantage in using supplementary data. Note that one expert trajectory is a common choice in the existing literature. As for your concern that some trajectories may end up being bad, we notice that the collected expert trajectories are all in high quality for this benchmark. For the other tasks (e.g., Atari games and object recognization), we do consider the set-up of more than single demonstration.
>
> **Question 5:** In the object recognition task, the supplementary data is data with valid labels but with a different style. However, the focus is on "sub-optimal policy" data, and in other environments the supplementary data consists of different policies ("full dataset" setting) and outright wrong labels ("noisy expert"), where some actions are replaced with random ones. Following this, it seems like a more natural setup for object recognition task would be to do everything in single domain (e.g., "Real"), but randomly replace the sample label with a wrong label in the supplementary dataset. What is the reason behind using different sub-sets as supplementary data?
>
> **Response 5:** We utilize distinct subsets as supplementary data, as this approach is a common practice within this benchmark for examining the robustness of the learning method against distribution shifts. To establish a connection between this setup and our formulation, one might consider that each domain corresponds to a unique sub-state space.
>
> Nonetheless, in line with your insights, we have conducted a new experiment. In this experiment, each supplementary dataset is situated within the same domain as the expert dataset, and certain labels within the supplementary dataset have been deliberately subjected to noise injection. For comprehensive details, please refer to Table 2 in the separate PDF file. Empirical results demonstrate that ISW-BC outperforms baseline methods and maintains its robustness even within this particular scenario.
>
> **Comment 6:** Footnotes come after commas. Term "imitation gap" is used early on (e.g., lines 39 and 66), but not formally or informally defined.  Please use same colors consistently between figures.  Section 3 header sounds like it is missing something (e.g., should it be "Preliminaries" or "Problem setup"?) Numbers in the tables are in bit different format.
>
> **Response 6:** Thank you for bringing these issues to our attention. We will make the corresponding corrections and improvements based on your suggestions.
>
> ---
>
> We hope that the above answers can address your concerns satisfactorily and improve the clarity of our contribution. We would be grateful if you could re-evaluate our work based on the above responses.

---

> > ### Comment · Reviewer_Y1mx · 2023-08-14
> >
> > Thank you for your replies and additional experiments. Indeed the results look promising for the additional experiments you ran. It is reassuring to see that the method works with random agent data as well, which is easy to come by, compared to expert trajectories with random noise labels.
> >
> > Given the replies, I am happy to increase my score from 6 to 7. I encourage authors to take care to open-source the exact code to replicate results; not only it is good research to do so, but it will drive up the attention this work will get, as people are able to base their results on your code and use it as baselines in the future.

---

> > > ### Author Response · Authors · 2023-08-19
> > >
> > > Thanks a lot for your insightful comments and feedback. We will revise the paper as suggested and open-source the code and datasets. We are delighted to learn that our responses have addressed your concerns, and we express our deep appreciation for your reconsideration of the score.

---

### Official Review · Reviewer_LjZi · 2023-07-04

**Soundness:** 3 good
**Presentation:** 3 good
**Contribution:** 3 good
**Rating:** 7
**Confidence:** 4

**Summary:**

This paper studies the problem of offline imitation learning (IL) with a supplementary dataset, which can address the scarce expert data issue in pure IL. In this setting, the challenge is that the supplementary dataset may have out-of-distribution samples. This paper considers the classical method Behavioral Cloning (BC) and its variants, and proves their imitation gap bounds in offline IL with a supplementary dataset. The theoretical results show that the naïve BC on union dataset (NBCU) method suffers a non-vanishing gap, and thus may be worse than BC which only learns from the expert dataset. To address this issue, the authors propose the method Importance-sampling weighted BC (ISW-BC), which can select in-distribution samples in supplementary dataset. They prove that ISW-BC can eliminate the gap in NBCU. The experimental results also show that ISW-BC outperforms existing methods on a variety of tasks.

**Strengths:**

1. This paper conducts a systematic theoretical study of offline IL with a supplementary dataset. The developed theory closes the gap between theory and practice and lays a foundation for further studies of this problem.

2. This paper proposes a simple and effective method ISW-BC. The authors validate that ISW-BC can address the distribution shift issue in both theory and practice, which makes advances over existing methods.

**Weaknesses:**

1. This paper is a bit dense to read. I believe that this paper would benefit from providing more intuitions and proof sketch for the theoretical results. Besides, the authors should give more analysis of the experimental results, which can give the reader an intuitive idea about how and where the proposed algorithm improves upon existing methods.

**Questions:**

The theoretical results for ISW-BC in lines 260-270 are quite complicated and difficult to understand. Can the authors give more explanations and proof sketch for these results?

**Limitations:**

The authors have discussed the limitations and broader impacts of this paper in the conclusion part and appendix.

---

> ### Author Rebuttal · Authors · 2023-08-09
>
> We appreciate your time to review and provide positive feedback for our work.
>
> **Comment 1:** This paper is a bit dense to read. I believe that this paper would benefit from providing more intuitions and proof sketch for the theoretical results. Besides, the authors should give more analysis of the experimental results, which can give the reader an intuitive idea about how and where the proposed algorithm improves upon existing methods.
>
> **Response 1:** Thanks for your helpful suggestions. We will revise our paper based on your comments.
>
> **Question 2:** The theoretical results for ISW-BC in lines 260-270 are quite complicated and difficult to understand. Can the authors give more explanations and proof sketch for these results?
>
> **Response 2:** Thanks for your helpful comments. We provide a proof sketch below for your reference and we will put it in the revised paper.
>
> First, based on a classical reduction lemma, we can upper bound the imitation gap by the divergence between the expert policy and the learned policy distributions.
> $$
> V\left(\pi^{\mathrm{E}}\right) - V\left(\pi^{\mathrm{ISW}-\mathrm{BC}}\right) \leq H \sum\_{h=1}^H \mathbb{E}\_{s \sim d\_h^{\mathrm{E}}(\cdot)}\left[\mathrm{TV}\left(\pi\_h^{\mathrm{E}}(\cdot \mid s), \pi\_h^{\mathrm{ISW}-\mathrm{BC}}(\cdot \mid s)\right)\right]
> $$
> Then our target is to analyze the properties of the learned policy distribution. To achieve this goal, we derive the closed-form solution for the learned policy, on which the learned weights of samples play a crucial role.
> $$
> \pi\_h^{\mathrm{ISW}-\mathrm{BC}}(a \mid s)=\frac{\widehat{d\_h^{\mathrm{U}}}(s, a) w\_h(s, a) \mathbb{I}\left[w\_h(s, a) \geq \delta\right]}{\sum\_{a \in \mathcal{A}} \widehat{d\_h^{\mathrm{U}}}(s, a) w\_h(s, a) \mathbb{I}\left[w\_h(s, a) \geq \delta\right]}
> $$
> Then we continue to analyze the properties of the learned classifier which induces the weights of samples. Through a landscape-based analysis (Lemma 1 and Lemma 2), we prove that the learned classifier induces the consistent margin with the ground-truth classifier ($\Delta\_h (\theta^\star\_h) > 0$) and thus can distinguish between in-expert-distribution samples $\mathcal{D}^{\text{E}}\_h \cup \mathcal{D}^{\text{S}, 1}\_h$ and out-expert-distribution samples $\mathcal{D}^{\text{S}, 2}\_h$. With this result, we further show that the learned policy matches the expert policy on states within in-expert-distribution samples $\mathcal{D}^{\text{E}}\_h \cup \mathcal{D}^{\text{S}, 1}\_h$. Finally, we can obtain the improved imitation gap bound.
>
> Please let us know if you have further concerns about this point.

---

### Official Review · Reviewer_Yb9D · 2023-07-05

**Soundness:** 3 good
**Presentation:** 3 good
**Contribution:** 2 fair
**Rating:** 6
**Confidence:** 3

**Summary:**

In the paper, the authors focus on imitation learning (IL) when working with supplementary imperfect data. They conduct a thorough theoretical analysis to understand the limitations of IL under various dataset compositions. The authors' theoretical analysis provides insights into the bounds and constraints of IL when dealing with different types of datasets.

To address this problem, they propose a novel method called importance-sampling behavior cloning ISW-BC. The proposed method is designed to mitigate the issues associated with imperfect data in IL. This technique leverages importance sampling to assign appropriate weights to different samples, thereby effectively reducing the impact of imperfections in the training data.

To validate the effectiveness of their approach, the authors conduct extensive evaluations on a diverse set of tasks. The results indicate that the proposed method outperforms the current state-of-the-art techniques in most cases. This suggests that the importance-sampling behavior cloning method is a promising solution for tackling the problem of imitation learning with supplementary imperfect data.

**Strengths:**

One strength of the paper is the authors' meticulous exploration of the various theoretical bounds that arise when working with imperfect data within the framework of BC. By dissecting these limitations, they provide a deep understanding of the challenges faced in practice, enabling researchers and practitioners to make more informed decisions when applying BC to real-world datasets.

Furthermore, the authors introduce a novel method based on importance sampling, which offers a clear and intuitive approach for addressing the imperfections in the data.

In addition to their theoretical contributions, the authors demonstrate the practical relevance of their proposed method by conducting a thorough analysis on diverse tasks. This empirical evaluation validates the effectiveness of their approach across various application domains, further strengthening the paper's findings.

**Weaknesses:**

The authors' analysis lacks consideration of alternative methods that can effectively learn from imperfect data. [1]

In order for the method to be applied, a dataset of labeled expert demonstrations is required. In many practical applications, we do not have access to this information.

[1]Better-than-Demonstrator Imitation Learning via Automatically-Ranked Demonstrations,Daniel S. Brown, Wonjoon Goo and Scott Niekum, CoRL 2019

**Questions:**

In line 310, authors claim that NBCU performs worse than BC. This would be the expected results, however in Table 2 for Ant and HalfCheetah NBCU with Noisy Expert data performs significantly better than BC. Can the authors explain this in more detail?

**Limitations:**

See Weaknesses

---

> ### Author Rebuttal · Authors · 2023-08-09
>
> Thank you for taking the time to review and check our paper, and for your insightful comments.
>
> **Comment 1:** The authors' analysis lacks consideration of alternative methods that can effectively learn from imperfect data. [1]
>
> **Response 1:** Thank you for bringing up the work [1], which utilizes automatically generated ranked demonstrations to learn a reward function for imitation. However, unlike our analysis, [1] primarily focuses on the online setting, and its main theorem contributes significantly by enabling the learning of a policy better than the demonstrator using the recovered reward function. It is worth noting that [1] requires a different assumption, namely that the expert policy is sufficiently sub-optimal.
>
> We will address this point in the revised paper by including the above discussion.
>
> **Comment 2:** In order for the method to be applied, a dataset of labeled expert demonstrations is required. In many practical applications, we do not have access to this information.
>
> **Response 2:** We appreciate your concern and would like to address it with two potential solutions. First, while obtaining a *complete* dataset of labeled expert demonstrations may be challenging in some practical scenarios, we believe it can be feasible and cost-effective to acquire *a few* labeled expert demonstrations (e.g., at least one expert trajectory in the locomotion control tasks considered in our experiments).
>
> Alternatively, if obtaining labeled expert demonstrations proves truly unfeasible, we propose a two-stage solution. Let us consider a mixed dataset containing both expert trajectories and non-expert trajectories. We can divide this dataset into two equal parts, denoted as $D_1$ and $D_2$. In the first stage, we infer a representative policy $\widehat{\pi}^{\operatorname{E}}$ from $D_1$ and use this policy to generate labels for the corresponding trajectories in $D_1$. Subsequently, in the second stage, we apply our framework, considering $D_2$ as supplementary data. Our intuition here is that expert trajectories likely constitute the majority of the entire dataset, so the policy $\widehat{\pi}^{\operatorname{E}}$ recovered from the noisy data $D_1$ can effectively act as a proxy for the expert policy.
>
> **Question 3:** In line 310, authors claim that NBCU performs worse than BC. This would be the expected results, however in Table 2 for Ant and HalfCheetah NBCU with Noisy Expert data performs significantly better than BC. Can the authors explain this in more detail?
>
> **Response 3:** Thanks for the nice observation and we have carefully examined this case. Through visualization (via PCA), we found that the state coverage (between the expert data and noisy non-expert data) is relatively nice for Ant and HalfCheetah tasks (compared with the other two tasks). Therefore, NBCU performs relatively well on these two tasks. For your reference, we have included the visualization plots in Figure 1 of a separate PDF file.
>
> ---
>
> We hope that the above answers can address your concerns satisfactorily. We would be grateful if you could re-evaluate our paper based on the above responses. We look forward to receiving your further feedback.

---

> > ### Comment · Reviewer_Yb9D · 2023-08-16
> >
> > Thanks for the responses. The provided responses adequately address my concerns and answer my questions, therefore I will increase my score to Weak Accept.
> >
> > The response that the authors provided to *Question 3* should be included in the main paper, and the strong claim that the authors make in line 310 should be adjusted accordingly.

---

> > > ### Author Response · Authors · 2023-08-19
> > >
> > > Your valuable comments and feedback are deeply appreciated. We are pleased to know that our responses addressed your concerns, and we extend our gratitude for your kind reconsideration of the score. We are committed to integrating your suggestions as we revise the paper.

---

### Author Rebuttal · Authors · 2023-08-09

We thank all reviewers for their expertise and efforts in reviewing our paper. We have responsed to each review seperately. We hope that our response can address the concerns well. Furthermore, we look forward to any additional comments or suggestions for improvement.

Please take note that we have attached a separate PDF file containing new experimental results aimed at addressing the concerns from Reviewers Yb9D, Y1mx, and FJAh.

Best,

The Authors

---

### Decision · Program_Chairs · 2023-09-21

**Decision:**

Accept (spotlight)

**Comment:**

All reviewers agree that this is a submission above the bar for acceptance. I have gone through the reviews and paper myself, and I find the paper to overall be a strong submission all-around, combining strong theoretical results with extensive experiments and clear writing/presentation.